# Quantum Gutzwiller approach
# for the two-component Bose-Hubbard model

**Victor E. Colussi[1]⋆, Fabio Caleffi[2], Chiara Menotti[1] and Alessio Recati[1,3]**

**1** INO-CNR BEC Center and Dipartimento di Fisica, Università di Trento,
Via Sommarive 14, 38123 Povo, Trento, Italy
**2** International School for Advanced Studies (SISSA),
Via Bonomea 265, I-34136 Trieste, Italy
**3** Trento Institute for Fundamental Physics and Applications,
INFN, Via Sommarive 14, 38123 Povo, Trento, Italy

⋆ colussiv@gmail.com

## Abstract

We study the effects of quantum fluctuations in the two-component Bose-Hubbard model generalizing to mixtures the quantum Gutzwiller approach introduced recently in [Phys. Rev. Research 2, 033276 (2020)]. As a basis for our study, we analyze the mean-field ground-state phase diagram and spectrum of elementary excitations, with particular emphasis on the quantum phase transitions of the model. Within the quantum critical regimes, we address both the superfluid transport properties and the linear response dynamics to density and spin probes of direct experimental relevance. Crucially, we find that quantum fluctuations have a dramatic effect on the drag between the superfluid species of the system, particularly in the vicinity of the paired and antipaired phases absent in the usual one-component Bose-Hubbard model. Additionally, we analyse the contributions of quantum corrections to the one-body coherence and density/spin fluctuations from the perspective of the collective modes of the system, providing results for the few-body correlations in all the regimes of the phase diagram.



# 1 Introduction

The precision control available in ultracold atomic gas experiments has opened up a platform where models of condensed matter physics can be simulated in a relatively defect-free environment. In particular, ultracold atoms trapped in optical lattices are described by Hubbard models [1, 2], which are of central importance to the study of solid-state materials [3]. The reduction of three-body losses in lattices makes possible the study of strongly-correlated regimes absent in the continuum, such as the paradigmatic Mott insulator (MI) to superfluid (SF) quantum phase transition for single-component bosons [4]. In the two-component Bose-Hubbard (BH) model a richer phase diagram emerges, including the additional possibility of pair (PSF) and counterflow (CFSF) superfluids, supersolidity, charge-density quasiorder, and peculiar magnetic states [5–10]. Such coupled superfluids can undergo also mutual dissipationless transport with an induced entrainment or counterflow of one component due to a nonzero superfluid velocity of the other. This phenomenon, better known as superfluid drag, was first discussed by Andreev and Bashkin in the context of three-fluid hydrodynamics [11], but is of universal relevance to systems ranging from neutron-star matter [12–16] to multicomponent superconductors [17–19] and ultracold atomic mixtures [9, 20–28]. Direct measurement of this effect has however remained elusive, due in part to the low miscibility of superfluid $^3$He and $^4$He and recombination heating in strongly-interacting ultracold atomic mixtures. Recently, the PSF and CFSF phase transitions of the two-component BH model have emerged as promising candidates where the drag can saturate at its maximum value [25, 26]. Still, a

deeper understanding of the fundamental role played by quantum fluctuations is needed to gain insight into the physics of such strongly-correlated quantum critical regimes of Hubbard models at zero temperature.

The Gutzwiller approach provides a simple solution of the BH model able to interpolate between a coherent-state description of a dilute Bose-Einstein condensate (BEC) and a MI with integer occupation of each lattice site [29–31]. Its generalization to the two-component BH model is based on the wave function ansatz [32, 33]

$$|\Psi_G\rangle = \bigotimes_{\mathbf{r}} \sum_{n_1, n_2=0}^{\infty} c_{n_1, n_2}(\mathbf{r}) |n_1, n_2, \mathbf{r}\rangle, \tag{1}$$

which is a site-factorized product of local Fock states with occupation of $n_i$ atoms in the $i^{\text{th}}$ component at site $\mathbf{r}$ weighted by normalized complex amplitudes $c_{n_1, n_2}(\mathbf{r})$, which can capture at the mean-field level paired and antipaired ground states and superfluidity [32]. The Gutzwiller description is however inadequate in strongly-interacting and critical regimes, where quantum fluctuations are enhanced. Of course, also the basic assumptions of the celebrated Bogoliubov theory of quantum fluctuations about a coherent-state BEC break down in the vicinity of the phase transitions [34, 35]. In this spirit, a canonical quantization procedure $c \to \hat{c}$ was proposed in Ref. [36], which enables an order-by-order layering of quantum fluctuations on top of the mean-field Gutzwiller ground state, remaining valid in the whole phase diagram. Predictions of the resultant quantum Gutzwiller theory (QGW) featured a successful comparison across the MI-SF phase transition for single-component bosons to the available quantum Monte Carlo (QMC) data, while requiring only modest computational costs and providing a valuable description of the role of quantum fluctuations through a semi-analytical formalism. Moreover, the QGW theory was used recently to study the non-Markovian spin decoherence of a two-level impurity embedded in a two-dimensional BH model [37]. It is important to note that, at the level of Gaussian fluctuations, the QGW approach is analogous to slave boson methods (c.f Refs. [38–40]). However, in the two-component BH model comparatively little is known about time-dependent phenomena and the role of quantum fluctuations in quantum critical regimes, motivating the establishment of the QGW theory for bosonic mixtures.

In this work, we study a homogeneous system of two-component bosons on a square lattice considering short-range intra and interspecies interactions, which can be realized in optical lattices loaded with atoms of two different species or internal states [41–43]. Although the derivations presented in this work are completely general, our numerical findings are specific to two dimensions where existing QMC results [25] make quantitative comparisons with our predictions for the drag possible, and where there exists a strong motivating analogy between the fermionic version of the problem and high-temperature superconductivity [44]. We first study the rich phase diagram of the model for both repulsive and attractive interspecies interactions, finding counterflow and paired superfluid phases in addition to the MI and SF phases which carry over from single-component bosonic systems. We show how, in the case of mixtures, the QGW approach provides a straightforward way to calculate linear response and correlation functions to a desired order in the quantum fluctuations. This permits a systematic study of the role of quantum corrections, which we first investigate by considering the linear response dynamics of the system to density and spin perturbations. In this respect, we highlight the experimentally relevant signatures of the onset of the PSF and CFSF phases in the dynamical structure factor. We then focus on superfluid transport in quantum critical regimes, finding in particular a large interspecies drag comparable in magnitude with the superfluid density in the vicinity of the CFSF and PSF phases. Furthermore, we address the one and two-body correlation functions, focusing on the strongly-interacting regime, where quantum fluctuations play a crucial role. Specifically, we find that the PSF and CFSF transitions behave

as channel-selective MI transitions with respect to the spin and density degrees of freedom, respectively.

We have structured the present work in a relatively self-contained manner, which we hope may prove useful also as a starting point and reference for future studies. In Sec. 2, we first outline the time-dependent Gutzwiller method for the two-component BH model, and then study the ground state and elementary excitations of the system for both repulsive and attractive interactions, expanding on Ref. [32] where possible. Subsequently, to include the effects of quantum fluctuations, we introduce the QGW approach, which generalizes the formalism of Ref. [36] to mixtures. Section 3 is focused on applying the QGW method to study the effects of beyond mean-field local and non-local quantum correlations in a range of experimentally relevant observables. Here, general semi-analytical formulas are derived for the susceptibility, compressibility, density and spin sound speeds, superfluid components (including the drag), one-body coherence function, and spin and density pair correlation functions. In particular, the order-by-order contributions of quantum fluctuations are made always explicitly clear. We conclude in Sec. 4 by providing also an outlook for future studies. The appendices contain additional details on the QGW approach and derivations of the linear response formalism for mixtures.

## 2 Model and Theory

In Sec. 2.1, we analyze the two-component BH model using the $\mathbb{C}$-number Gutzwiller ansatz. Subsequently, in Sec. 2.2.1 (2.2.2), we discuss in depth the ground state and elementary excitations for repulsive (attractive) interactions between the two components of the system. We complete our theoretical background in Sec. 2.3, where we go beyond the mean-field Gutzwiller approximation by generalizing the QGW method introduced in Ref. [36] to Bose mixtures.

### 2.1 Lagrangian formulation within the Gutzwiller ansatz for mixtures

We start from the two-component Bose-Hubbard (BH) model [1]

$$\hat{H} = \sum_{i=1}^{2}\sum_{\mathbf{r}}\left[-J_i\sum_{j=1}^{d}\left(\hat{a}_{i,\mathbf{r}}^{\dagger}\hat{a}_{i,\mathbf{r}+\mathbf{e}_j}+\text{h.c.}\right)+\frac{U_i}{2}\,\hat{n}_{i,\mathbf{r}}\left(\hat{n}_{i,\mathbf{r}}-1\right)-\mu_i\,\hat{n}_{i,\mathbf{r}}\right]+U_{12}\sum_{\mathbf{r}}\hat{n}_{1,\mathbf{r}}\hat{n}_{2,\mathbf{r}}, \quad (2)$$

where $\mathbf{r}$ is the site index in $d$ dimensions with unit vector $\mathbf{e}_j$, while $J_i$, $\mu_i$, $U_i$ and $U_{12}$ are the species-dependent tunneling coefficient between neighboring sites, chemical potential and on-site intra/interspecies interaction strengths, respectively. In this work, we examine only the $\mathbb{Z}_2$-symmetric case where $J_1 = J_2 = J$, $\mu_i = \mu_2 = \mu$ and $U_1 = U_2 = U$, keeping always $|U_{12}/U| < 1$ to avoid phase separation, as well as to prevent the system from collapsing [45]. The bosonic operators $\hat{a}_{i,\mathbf{r}}^{\dagger}$ and $\hat{a}_{i,\mathbf{r}}$ create and destroy, respectively, a particle of species $i$ at the lattice site $\mathbf{r}$, and are related to the corresponding local density operator via $\hat{n}_{i,\mathbf{r}} = \hat{a}_{i,\mathbf{r}}^{\dagger}\hat{a}_{i,\mathbf{r}}$. We define also the total and spin densities as given by $\hat{n}_{d,\mathbf{r}} = \sum_i \hat{n}_{i,\mathbf{r}}$ and $\hat{n}_{s,\mathbf{r}} = \hat{n}_{1,\mathbf{r}} - \hat{n}_{2,\mathbf{r}}$, respectively. In the following, we consider a uniform square lattice of volume $V$ composed by $I$ sites with lattice spacing $a^1$, such that the "bare" effective mass is $m \equiv \hbar^2/(2Ja^2)$.

The two-component Gutzwiller mean-field ansatz for the many-body wave function introduced in Eq. (1) is a site-factorized product of local Fock states weighted by normalized

---

[1]In this work, the QGW results are obtained on a lattice of size $I = 128^2$ unless otherwise specified. This lattice is sufficiently large such that our calculations are free from finite-size effects.

complex amplitudes $c_{n_1,n_2}(\mathbf{r})$ such that $\sum_{n_1,n_2} \left| c_{n_1,n_2}(\mathbf{r}) \right|^2 = 1$. Within the Gutzwiller ansatz, the local density is given by

$$n_i(\mathbf{r}) = \langle \hat{n}_{i,\mathbf{r}} \rangle = \sum_{n_1,n_2}^{\infty} \left( n_1 \, \delta_{i,1} + n_2 \, \delta_{i,2} \right) \left| c_{n_1,n_2}(\mathbf{r}) \right|^2 . \tag{3}$$

In addition to the Mott insulator (MI) and superfluid (SF) phases in common with the one-component case (cf. Ref. [29]), the two-component Bose mixture exhibits also the possibility of counterflow superfluid (CFSF) and pair superfluid (PSF) phases, see Refs. [6,7,10,46]. To distinguish between these phases, besides the one-body order parameters

$$\psi_1(\mathbf{r}) = \langle \hat{a}_{1,\mathbf{r}} \rangle = \sum_{n_1,n_2}^{\infty} \sqrt{n_1} \, c_{n_1-1,n_2}^*(\mathbf{r}) \, c_{n_1,n_2}(\mathbf{r}), \tag{4a}$$

$$\psi_2(\mathbf{r}) = \langle \hat{a}_{2,\mathbf{r}} \rangle = \sum_{n_1,n_2}^{\infty} \sqrt{n_2} \, c_{n_1,n_2-1}^*(\mathbf{r}) \, c_{n_1,n_2}(\mathbf{r}), \tag{4b}$$

which are non-zero only in the SF phase, we introduce the pair and antipair order parameters

$$\psi_P(\mathbf{r}) = \langle \hat{a}_{1,\mathbf{r}} \hat{a}_{2,\mathbf{r}} \rangle - \langle \hat{a}_{1,\mathbf{r}} \rangle \langle \hat{a}_{2,\mathbf{r}} \rangle = \sum_{n_1,n_2} \sqrt{n_1 \, n_2} \, c_{n_1-1,n_2-1}^*(\mathbf{r}) \, c_{n_1,n_2}(\mathbf{r}) - \psi_1(\mathbf{r}) \, \psi_2(\mathbf{r}), \tag{5a}$$

$$\psi_C(\mathbf{r}) = \langle \hat{a}_{1,\mathbf{r}} \hat{a}_{2,\mathbf{r}}^\dagger \rangle - \langle \hat{a}_{1,\mathbf{r}} \rangle \langle \hat{a}_{2,\mathbf{r}}^\dagger \rangle = \sum_{n_1,n_2} \sqrt{n_1(n_2+1)} \, c_{n_1-1,n_2+1}^*(\mathbf{r}) \, c_{n_1,n_2}(\mathbf{r}) - \psi_1(\mathbf{r}) \, \psi_2^*(\mathbf{r}), \tag{5b}$$

which identify univocally the PSF and CFSF phases, respectively. Notice that, in constructing the pair/antipair order parameters, possible contributions of the one-body order parameters have been explicitly removed to ensure that $\psi_{P/C} \neq 0$ reflects *intrinsically* particle + hole (CFSF) or particle + particle (hole + hole) (PSF) correlations between the two species.

From the Hamiltonian expectation value $\langle \Psi_G | \hat{H} | \Psi_G \rangle$, we can readily build a Lagrangian for the Gutzwiller ansatz,

$$\mathcal{L}[c,c^*] = \sum_{\mathbf{r}} \sum_{n_1,n_2} \left\{ \frac{i\hbar}{2} \left[ c_{n_1,n_2}^*(\mathbf{r}) \, \dot{c}_{n_1,n_2}(\mathbf{r}) - \text{c.c.} \right] - H_{n_1,n_2} \left| c_{n_1,n_2}(\mathbf{r}) \right|^2 \right\}$$
$$+ J \sum_{i=1}^{2} \sum_{\mathbf{r}} \sum_{j=1}^{d} \left[ \psi_i^*(\mathbf{r}) \, \psi_i(\mathbf{r} + \mathbf{e}_j) + \text{c.c.} \right], \tag{6}$$

where the dots indicate temporal derivatives and

$$H_{n_1,n_2} = \sum_{i=1}^{2} \left[ \frac{U}{2} \, n_i(n_i - 1) - \mu \, n_i \right] + U_{12} \, n_1 \, n_2 . \tag{7}$$

The classical Euler-Lagrange equations of motion for the Gutzwiller amplitudes, with the complex conjugate parameters $c_{n_1,n_2}^*(\mathbf{r}) = \partial \mathcal{L}/\partial \dot{c}_{n_1,n_2}(\mathbf{r})$ playing the role of canonical momenta, are given by the two-component time-dependent Gutzwiller equations (2GE)

$$i\hbar \dot{c}_{n_1,n_2}(\mathbf{r}) = H_{n_1,n_2} c_{n_1,n_2} \tag{8}$$

$$-J \sum_{i=1}^{d} \left\{ \sqrt{n_1+1} \, c_{n_1+1,n_2}(\mathbf{r}) \left[ \psi_1^*(\mathbf{r}+\mathbf{e}_i) + \psi_1^*(\mathbf{r}-\mathbf{e}_i) \right] + \sqrt{n_1} \, c_{n_1-1,n_2}(\mathbf{r}) \left[ \psi_1(\mathbf{r}+\mathbf{e}_i) + \psi_1(\mathbf{r}-\mathbf{e}_i) \right] \right.$$
$$\left. + \sqrt{n_2+1} \, c_{n_1,n_2+1}(\mathbf{r}) \left[ \psi_2^*(\mathbf{r}+\mathbf{e}_i) + \psi_2^*(\mathbf{r}-\mathbf{e}_i) \right] + \sqrt{n_2} \, c_{n_1,n_2-1}(\mathbf{r}) \left[ \psi_2(\mathbf{r}+\mathbf{e}_i) + \psi_2(\mathbf{r}-\mathbf{e}_i) \right] \right\},$$

which were derived previously in Ref. [32]. The 2GE are straightforward generalizations of their one-component counterparts (cf. Ref. [29]) with the additional contribution of the diagonal interspecies coupling $U_{12}$. In order to explore solutions of the 2GE, we search first for the stationary solutions $c_{n_1,n_2}(\mathbf{r}) = c^0_{n_1,n_2} e^{-i\omega_0 t}$ independent of the site index $\mathbf{r}$. The ground state energy is then given by

$$\hbar\omega_0 = -4dJ\sum_{i=1}^{2}|\psi_{0,i}|^2 + \sum_{n_1,n_2}\left\{\sum_{i=1}^{2}\left[\frac{U}{2}n_i(n_i-1)-\mu n_i\right] + U_{12}n_1 n_2\right\}\left|c^0_{n_1,n_2}\right|^2, \quad (9)$$

where the "0" sub/superscript indicates quantities evaluated with respect to the ground state Gutzwiller amplitudes $c^0_{n_1,n_2}$ obtained by diagonalizing Eq. (8). For instance, the expression of the mean-field total density is simply given by

$$n_{0,d} = \sum_{n_1,n_2}^{\infty}(n_1 + n_2)\left|c^0_{n_1,n_2}\right|^2. \quad (10)$$

In the remainder, it is assumed always that the ground state parameters $c^0_{n_1,n_2}$ are real numbers.

In order to study the linear response dynamics of the system around the ground state, we consider small perturbations around the stationary solution of the form

$$c_{n_1,n_2}(\mathbf{r}) = \left[c^0_{n_1,n_2} + c^1_{n_1,n_2}(\mathbf{r},t)\right]e^{-i\omega_0 t}, \quad (11)$$

which can be expanded in terms of plane waves as

$$c^1_{n_1,n_2}(\mathbf{r},t) = \sum_{\mathbf{k}}\left[u_{\mathbf{k},n_1,n_2}e^{i(\mathbf{k}\cdot\mathbf{r}-\omega_\mathbf{k}t)} + v^*_{\mathbf{k},n_1,n_2}e^{-i(\mathbf{k}\cdot\mathbf{r}-\omega_\mathbf{k}t)}\right]. \quad (12)$$

Linearizing the 2GE with respect to the amplitudes $u_{\mathbf{k},n_1,n_2}$ and $v_{\mathbf{k},n_1,n_2}$, one obtains an eigenvalue problem

$$\hbar\omega_\mathbf{k}\begin{pmatrix}u_\mathbf{k}\\\underline{v}_\mathbf{k}\end{pmatrix} = \mathcal{L}_\mathbf{k}\begin{pmatrix}u_\mathbf{k}\\\underline{v}_\mathbf{k}\end{pmatrix} \quad , \quad \hat{\mathcal{L}}_\mathbf{k} = \begin{pmatrix}H_\mathbf{k} & K_\mathbf{k}\\-K_\mathbf{k} & -H_\mathbf{k}\end{pmatrix}, \quad (13)$$

where the vectors $\underline{u}_{\alpha,\mathbf{k}}\left(\underline{v}_{\alpha,\mathbf{k}}\right)$ contain the particle (hole) components $u_{\alpha,\mathbf{k},n_1,n_2}\left(v_{\alpha,\mathbf{k},n_1,n_2}\right)$, respectively. The positive eigenvalues $\omega_\mathbf{k}$ of the pseudo-Hermitian matrix $\hat{\mathcal{L}}_\mathbf{k}$ describe the multibranch excitation spectrum of the fluctuations and are identified with the collective modes of the system. The elements of the matrix blocks $H_\mathbf{k}$ and $K_\mathbf{k}$ have expressions

$$\begin{aligned}
H_\mathbf{k}^{n_1,n_2,n_1',n_2'} = &\left\{\sum_{i=1}^{2}\left[\frac{U}{2}n_i(n_i-1)-\mu n_i\right] + U_{12}n_1 n_2 - \hbar\omega_0\right\}\delta_{n_1,n_1'}\delta_{n_2,n_2'}\\
&-J(\mathbf{0})\psi_{0,1}\left(\sqrt{n_1'}\,\delta_{n_1',n_1+1} + \sqrt{n_1}\,\delta_{n_1,n_1'+1}\right)\delta_{n_2,n_2'}\\
&-J(\mathbf{0})\psi_{0,2}\left(\sqrt{n_2'}\,\delta_{n_2',n_2+1} + \sqrt{n_2}\,\delta_{n_2,n_2'+1}\right)\delta_{n_1,n_1'}\\
&-J(\mathbf{k})\left[\sqrt{n_1+1}\sqrt{n_1'+1}\,c^0_{n_1+1,n_2}c^0_{n_1'+1,n_2'} + \sqrt{n_1}\sqrt{n_1'}\,c^0_{n_1-1,n_2}c^0_{n_1'-1,n_2'}\right]\\
&-J(\mathbf{k})\left[\sqrt{n_2+1}\sqrt{n_2'+1}\,c^0_{n_1,n_2+1}c^0_{n_1',n_2'+1} + \sqrt{n_2}\sqrt{n_2'}\,c^0_{n_1,n_2-1}c^0_{n_1',n_2'-1}\right],
\end{aligned} \quad (14a)$$

$$\begin{aligned}
K_\mathbf{k}^{n_1,n_2,n_1',n_2'} = &-J(\mathbf{k})\left[\sqrt{n_1+1}\sqrt{n_1'}\,c^0_{n_1+1,n_2}c^0_{n_1'-1,n_2'} + \sqrt{n_1}\sqrt{n_1'+1}\,c^0_{n_1-1,n_2}c^0_{n_1'+1,n_2'}\right]\\
&-J(\mathbf{k})\left[\sqrt{n_2+1}\sqrt{n_2'}\,c^0_{n_1,n_2+1}c^0_{n_1',n_2'-1} + \sqrt{n_2}\sqrt{n_2'+1}\,c^0_{n_1,n_2-1}c^0_{n_1',n_2'+1}\right],
\end{aligned} \quad (14b)$$

with $J(\mathbf{k}) = 2 d J - \epsilon(\mathbf{k})$ and where

$$\epsilon(\mathbf{k}) = 4 J \sum_{j=1}^{d} \sin^2\left(\frac{k_j a}{2}\right), \tag{15}$$

is the free-particle dispersion law on a $d$-dimensional lattice. It follows that the dependence on $\mathbf{k}$ of the excitations is solely determined by the variable

$$x = \left[\frac{1}{d}\sum_{j=1}^{d}\sin^2\left(\frac{k_j a}{2}\right)\right]^{1/2}, \tag{16}$$

which varies from 0 to 1 and scales as $x \approx |\mathbf{k}| a / \left(2\sqrt{d}\right)$ at small momenta.

For later convenience, we conclude this subsection by introducing some useful definitions. Making use of Eq. (12), the linear response dynamics of the one-body order parameter $\psi_i(\mathbf{r}, t) = \psi_{0,i} + \psi_i^1(\mathbf{r}, t)$ can be expanded in plane waves as

$$\psi_i^1(\mathbf{r}, t) = \sum_{\mathbf{k}}\left[U_{i,\mathbf{k}}\, e^{i(\mathbf{k}\cdot\mathbf{r}-\omega_{\mathbf{k}} t)} + V_{i,\mathbf{k}}^*\, e^{-i(\mathbf{k}\cdot\mathbf{r}-\omega_{\mathbf{k}} t)}\right], \tag{17}$$

where we have introduced the particle-hole amplitudes

$$U_{1,\mathbf{k}} = \sum_{n_1,n_2}\sqrt{n_1}\left(c^0_{n_1-1,n_2}\, u_{\mathbf{k},n_1,n_2} + c^0_{n_1,n_2}\, v_{\mathbf{k},n_1-1,n_2}\right), \tag{18a}$$

$$V_{1,\mathbf{k}} = \sum_{n_1,n_2}\sqrt{n_1}\left(c^0_{n_1,n_2}\, u_{\mathbf{k},n_1-1,n_2} + c^0_{n_1-1,n_2}\, v_{\mathbf{k},n_1,n_2}\right), \tag{18b}$$

$$U_{2,\mathbf{k}} = \sum_{n_1,n_2}\sqrt{n_2}\left(c^0_{n_1,n_2-1}\, u_{\mathbf{k},n_1,n_2} + c^0_{n_1,n_2}\, v_{\mathbf{k},n_1,n_2-1}\right), \tag{18c}$$

$$V_{2,\mathbf{k}} = \sum_{n_1,n_2}\sqrt{n_2}\left(c^0_{n_1,n_2}\, u_{\mathbf{k},n_1,n_2-1} + c^0_{n_1,n_2-1}\, v_{\mathbf{k},n_1,n_2}\right). \tag{18d}$$

It is possible also to study the linear response dynamics of the pair and antipair order parameters by defining analogous pair/antipair amplitudes $U_{\mathrm{P/C},\mathbf{k}}$ and $V_{\mathrm{P/C},\mathbf{k}}$, which are discussed further in App. A.2. Finally, the linear response of the local density for each species is given by

$$n_i(\mathbf{r}, t) = n_{0,i} + \sum_{\mathbf{k}}\left[N_{i,\mathbf{k}}\, e^{i(\mathbf{k}\cdot\mathbf{r}-\omega_{\mathbf{k}} t)} + \text{c.c.}\right], \tag{19}$$

where the density fluctuation amplitude reads

$$N_{i,\mathbf{k}} = \sum_{n_1,n_2} n_i\, c^0_{n_1,n_2}\left(u_{\mathbf{k},n_1,n_2} + v_{\mathbf{k},n_1,n_2}\right). \tag{20}$$

In the following, we use the above formal results to analyze in the detail the specific properties of the ground state and collective excitations of the MI and superfluid (SF, CFSF and PSF) phases of the BH model for both a repulsive (Sec. 2.2.1) and an attractive (Sec. 2.2.2) coupling between the two Bose components.

## 2.2 Ground state and excitations

In this subsection, we explore the phase diagram of the system for repulsive ($U_{12} > 0$) and attractive ($U_{12} < 0$) interspecies interactions. Moreover, we perform a detailed characterization of the excitations of the system across the various quantum phase transitions of the BH model. Looking ahead, the analysis of this subsection will facilitate an in-depth understanding of the quantum correlations in terms of the spectral structure of the collective modes by means of the QGW approach, which is the subject of the remainder of this work.

We preface our discussion by noting that in contrast to the single-species BH model, in the present context one also finds CFSF and PSF phases intruding between the MI regions, signalled by non-zero values of the pairing order parameters $\psi_{0,C}$ and $\psi_{0,P}$, respectively. Additionally, these quantities may be non-zero in the vicinity of the various phase transitions alongside a finite one-body condensate $\psi_{0,i}$, which marks the entrance into the SF region. Ultimately, the pair/antipair order parameters vanish in the limit $2dJ/U \gg 1$ as expected.

### 2.2.1 Repulsive interaction ($U_{12} > 0$)

(*Mott Insulator*) – In Figs. 1(a)-(b), we show the ground state phase diagram for two values of $U_{12} > 0$. In general, for strong enough $U$ we find MI regions [light blue areas] at even total filling whose ground state is $|n_{0,d}/2, n_{0,d}/2\rangle$ with energy

$$\hbar\omega_0 = U\frac{n_{0,d}}{2}\left(\frac{n_{0,d}}{2} - 1\right) - \mu\, n_{0,d} + U_{12}\frac{n_{0,d}^2}{4}. \tag{21}$$

Within the MI lobes, the excitation spectrum can be calculated analytically from Eq. (13) as

$$\hbar\omega_{\pm,\mathbf{k}} = \frac{1}{2}\left\{\sqrt{J^2(\mathbf{k}) - 2J(\mathbf{k})U(n_{0,d}+1) + U^2} \pm \left[J(\mathbf{k}) - U(n_{0,d}-1) - 2U_{12} + 2\mu\right]\right\}, \tag{22}$$

which is a modification of the one-component result to include the mean-field interaction energy between different species. This spectrum describes four dispersive branches in total, a pair of degenerate particle ('+') branches and a pair of degenerate hole ('−') branches.[2]

The second-order phase transition boundary between the MI and SF phases is determined by the onset of a finite one-body order parameter $\psi_{0,i}$ and the disappearance of the gap in the excitation spectrum ($\omega_{\mathbf{k}=0} = 0$). From Eq. (22), we find that this occurs for

$$2d\left(\frac{J}{U}\right)_c = \frac{(n_{0,d}/2 - \mu/U + U_{12}/U)(\mu/U - U_{12}/U - n_{0,d}/2 + 1)}{1 + \mu/U - U_{12}/U}, \tag{24}$$

which can be linked to the MI boundary of the one-component case via the mapping $\mu \to \mu - U_{12}$. The maximal value of $2d(J/U)_c$ determines the locations of the O(2) tip transitions at

$$2d\left(\frac{J}{U}\right)_c^{\max} = \left(\sqrt{\frac{n_{0,d}}{2} + 1} - \sqrt{\frac{n_{0,d}}{2}}\right)^2. \tag{25}$$

---

[2]On top of the particle-hole excitations, Eq. (13) presents an infinity of non-zero, uncoupled diagonal elements which describe non-dispersive bands with energy

$$\hbar\omega_{n_1,n_2} = \sum_{i=1}^{2}\left[\frac{U}{2}n_i(n_i-1) - \mu\, n_i\right] + U_{12}n_1 n_2 - \omega_0, \tag{23}$$

where the occupation indices $n_1$ and $n_2$ must be chosen not to fall into the $4 \times 4$ block that yields the dispersive bands (22). Therefore, Eq. (23) applies to non-negative integers $n_1$ and $n_2$ such that *neither* $(n_1, n_2) = (j, j\pm 1)$ nor $(j\pm 1, j)$ are satisfied, with $j \in \mathbb{N}$. We note also that, as a consequence of the $\mathbb{Z}_2$ symmetry, these bands are doubly degenerate $\omega_{n_1,n_2} = \omega_{n_2,n_1}$ for $n_1 \neq n_2$ and singly degenerate for $n_1 = n_2$.

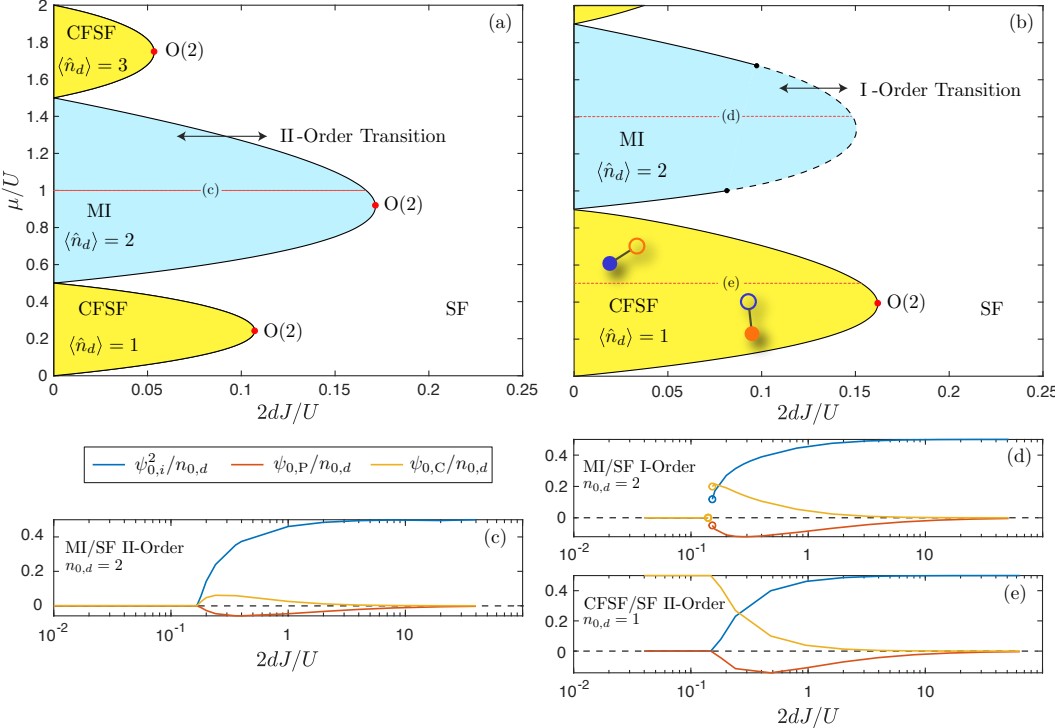

Figure 1: Ground state phase diagram for repulsive interspecies interactions (a) $U_{12}/U = 0.5$ and (b) $U_{12}/U = 0.9$. First and second-order transition lines are indicated by dashed and solid lines respectively, separating the MI (CFSF) lobes, identified by light blue (yellow) areas, from the SF region. The O(2) tip transitions are indicated by red dots, while the dot-shaped illustrations depict the particle-hole antipairs coupling the two components in the CFSF phase. (c) Behavior of the order parameters at the crossing of the second-order MI-to-SF transition for fixed $n_{0,d} = 2$ and $U_{12}/U = 0.5$. (d) Behavior of the order parameters at the crossing of the first-order MI-to-SF transition for fixed $n_{0,d} = 2$ and $U_{12}/U = 0.9$. (e) Behavior of the order parameters at the crossing of the second-order CFSF-to-SF transition for fixed $n_{0,d} = 1$ and $U_{12}/U = 0.9$. The solid red lines in (a) and (b) correspond to the $\mu(U)$ lines along which the data in (c), (d), and (e) are evaluated prior to the superfluid transition.

Again, we note that the chemical potential of the tip critical points is similarly shifted from the one-component value and is given by

$$\left(\frac{\mu}{U}\right)_c = \sqrt{\frac{n_{0,d}}{2}\left(\frac{n_{0,d}}{2}+1\right)} + \frac{U_{12}}{U} - 1. \tag{26}$$

In Fig. 2(a), we show how the band structure changes as the second-order MI-to-SF transition is traversed through the edge of the lobe (namely, away from the tip), specifically for $n_{0,d} = 2$ at $2\,d\,(J/U)_c \approx 0.167$ for $U_{12}/U = 0.5$ and $\mu/U = 1$. In general, in the MI phase, if the chemical potential is set above (below) its value at the tip, the first two bands are a pair of degenerate gapped particle (hole) bands of mixed spin or density character, whereas the next two bands correspond to degenerate gapped hole (particle) excitations. The doubly degenerate non-dispersive band $\omega_{0,2}$ is also visible as a dotted horizontal line. At the transition point, the gap of the lowest pair of degenerate particle (hole) bands vanish, while the degenerate hole (particle) bands remain gapped. Additionally, the gapless modes are purely

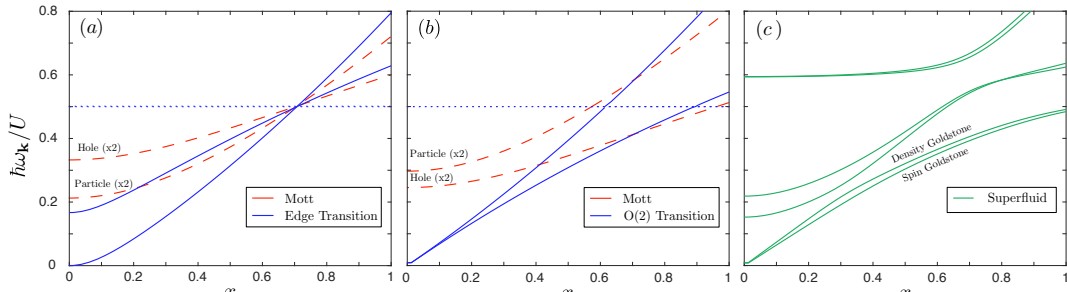

Figure 2: Excitation spectra across the MI-to-SF second-order (a) edge transition at $\mu/U = 1$ and (b) O(2) transition at $(\mu/U)_c \approx 0.914$, as well as (c) in the superfluid phase for $\mu/U = (\mu/U)_c$, choosing $U_{12}/U = 0.5$ at fixed $n_{0,d} = 2$ in $d = 2$. For the edge transition in panel (a), the chosen hopping energies are $2\,d\,J/U = 0.12$ (red lines) and $2\,d\,(J/U)_c \approx 0.167$ (blue lines); for the O(2) transition in panel (b), the hopping energies $2\,d\,J/U = 0.12$ (red lines) and $2\,d\,(J/U)_c^{\max} \approx 0.172$ (blue lines) are considered. The hopping energy in the SF phase in panel (c) is $2\,d\,J/U = 0.18$. The parentheses in panels (a)-(b) refer to the degeneracy of the hybridized particle and hole bands, while the spin and density Goldstone modes are indicated explicitly. The blue dotted horizontal lines correspond to the doubly degenerate non-dispersive band $\omega_{0,2}$.

quadratic at low momenta, signalling the characteristic vanishing of the speed of sound in common with the one-component case [29]. In Fig. 2(b), we perform a similar analysis of the spectral features of the second-order MI-to-SF transition when traversed through the tip critical point at $2\,d\,(J/U)_c^{\max} \approx 0.172$ for $U_{12}/U = 0.5$ and $(\mu/U)_c \approx 0.914$. Most importantly, we observe that at the tip transition point the gaps of the doubly degenerate particle and hole bands vanish simultaneously. Also, their dispersion becomes degenerate and perfectly linear at low momenta, which is characteristic of a finite sound speed at the O(2) transition [29].

Once the SF phase takes over, both at the edge and at the tip transition, the excitation bands which become gapless hybridize into spin and density modes (see Fig. 2(c)). These excitations correspond to the Goldstone modes that result from the breaking of the two $U(1)$ symmetries of the model, one for the density and the other for the spin channel. Their dispersion relation approaches the Bogoliubov bands [29]

$$\hbar\omega_{d/s,\mathbf{k}} = \sqrt{\epsilon(\mathbf{k})\left[\epsilon(\mathbf{k}) + n_{0,d}\,(U \pm U_{12})\right]}, \tag{27}$$

in the weakly-interacting limit $2\,d\,J/U \gg 1$. Close to the transition, the first two gapped branches, which also display individual density and spin character, are referred to as the Higgs modes of the system, in analogy to the single-component BH model [47–50]. The $\hbar\,\omega_{0,2}$ band becomes dispersive and hybridizes into spin and density excitations as well.

The transition between MI and SF phase can also be of first-order, as discussed in Refs. [6, 32, 46, 51]. In this case, the one-body order parameters $\psi_{0,i}$ exhibit a discontinuity across the critical boundary, as shown in Fig. 1(d). The behavior of the discontinuity at the first-order transition was studied in the mean-field Gutzwiller analysis of Ref. [32], where it was found that: (i) the jump of the order parameter increases with $U_{12}$ and then rapidly goes to zero as the phase separation boundary $U_{12}/U \sim 1$ is approached; (ii) the hopping window corresponding to the first-order transition widens with increasing $U_{12}$ starting from the tip and reaching $J = 0$ when approaching the phase separation condition. Across the first-order critical point, the structure of the excitation spectrum changes discontinuously between the different spectral features explored before, such that the modes in the MI and SF phases cannot



be smoothly connected.

(*Counterflow Superfluid*) – In Figs. 1(a)-(b), the phase diagram displays also CFSF phases [yellow areas] at odd total filling, characterized by a finite antipair order parameter $\psi_{0,C}$ and whose size increases with larger $U_{12}$. Within the CFSF lobes, one finds that the Fock states $|(n_{0,d}+1)/2,(n_{0,d}-1)/2\rangle$ and $|(n_{0,d}-1)/2,(n_{0,d}+1)/2\rangle$ are doubly degenerate with ground state energy

$$\hbar\,\omega_0 = U\,(n_{0,d}-1)^2 + U_{12}\,n_{0,d}\,(n_{0,d}-1) + \mu\,(1-2\,n_{0,d})\,. \tag{28}$$

Comparing this energy with Eq. (21), we find the boundary between neighboring MI and CFSF lobes at $J=0$, which in the case of the $n_{0,d}=2$ MI lobe and $n_{0,d}=1$ CFSF lobe occurs at $\mu=U_{12}$. In order to obtain the correct ground state, we symmetrize the antipair state as $[\,|(n_{0,d}+1)/2,(n_{0,d}-1)/2\rangle + |(n_{0,d}-1)/2,(n_{0,d}+1)/2\rangle\,]/\sqrt{2}$ as predicted in Refs. [6, 7, 9, 10, 47, 52].

In the CFSF phase, single-particle and hole excitations involve the subset of states $\{|(n_{0,d}+1)/2\pm1,(n_{0,d}-1)/2\rangle;|(n_{0,d}+1)/2,(n_{0,d}-1)/2\pm1\rangle\}$, which amount to three particle-like excitations and one (three) hole-excitations for the CFSF phase at $n_{0,d}=1$ ($n_{0,d}\geq 3$). From those excitations, one can construct three (four) modes belonging to the density channel, plus one (two) belonging to the spin channel. The general behaviour is such that a pair of particle and hole excitations in the density channel lower their energy while moving towards the boundary of the CFSF lobe, with the particle (hole) excitation closing the gap if the transition in crossed above (below) the tip chemical potential. Exactly at the tip, the lowest particle and hole excitations in the density channel close the gap simultaneously. Remarkably, the distinction between the edge and tip critical points in the density channel closely resembles the properties of the MI-to-SF transition.

For the CFSF phase at $n_{0,d}=1$, the excitation spectrum can be calculated analytically from Eq. (13). In that case, the spin channel hosts only one particle branch, decoupled from the rest and given by

$$\hbar\,\omega_{\mathbf{k}} = U - \mu - J(\mathbf{k})\,. \tag{29}$$

This corresponds to a free-particle dispersion, shifted by the mean-field interaction energy $U-\mu$. In the density sector, we extract two particle branches and one hole branch, whose excitation energies are the solutions of the equation

$$J(\mathbf{k})\big[U\,U_{12} - (\hbar\,\omega_{\mathbf{k}}\pm\mu)^2\big] \mp (\hbar\,\omega_{\mathbf{k}}\pm\mu)(U\mp\hbar\,\omega_{\mathbf{k}}-\mu)(U_{12}\mp\hbar\,\omega_{\mathbf{k}}-\mu) = 0\,. \tag{30}$$

Similarly to the spectrum of the MI phase, all the other excitations consist in an infinite sequence of non-dispersive bands, the first two of which have energies $\hbar\,\omega_{0,1} = \hbar\,\omega_{1,0} = -\mu - \hbar\,\omega_0 = 0$. In particular, these branches correspond to cost-free excitations and mirror the degeneracy of antipair states characterizing the mean-field ground state of the CFSF phase[3]. In the density sector, we extract two particle branches and one hole branch, whose

The second-order phase transition boundary between the CFSF and SF phases is determined from the closure of the smallest gap in the CFSF excitation spectrum. From Eq. (30), we find that for $n_{0,d}=1$ the gap closing occurs at

$$2\,d\left(\frac{J}{U}\right)_c = \frac{\mu}{U}\frac{(1-\mu/U)(\mu/U-U_{12}/U)}{(\mu/U)^2-U_{12}/U}\,, \tag{31}$$

---

[3]We note that the flat bands $\omega_{0,1}=\omega_{1,0}$ describe (trivial) excitations within the the antipair sector $\{u_{0,1},u_{1,0},v_{0,1},v_{1,0}\}$ and cannot describe the excitation and tunnelling of antipairs out of the ground state. Describing pairing collective modes within our model leads to violated completeness relations (see App. A.3). In Ref. [32], these bands were found to acquire a sound-like profile when higher-order hopping processes are included perturbatively in CFSF phase. These processes are however neglected in the 2GE (8), such that we find higher gapped excitation modes, absent in that work, whose low-energy behavior is strongly tied to the appearance of the SF phase and determines the one-body correlations in the CSFS phase (see Sec. 3).

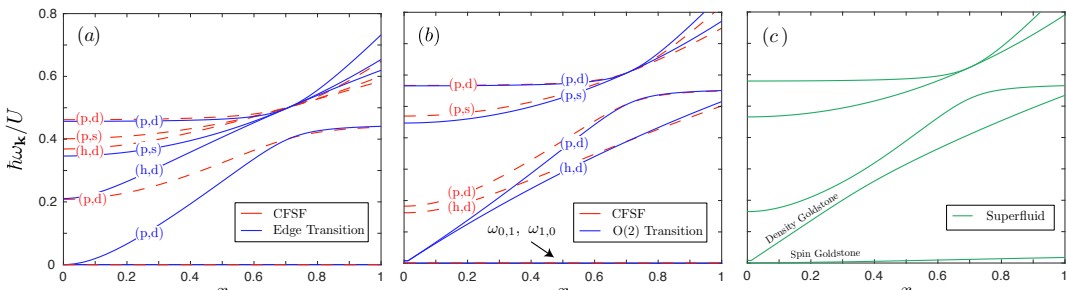

Figure 3: Excitation spectra across the CFSF-to-SF second-order for $U_{12}/U = 0.9$ at fixed $n_{0,d} = 1$ in $d = 2$ for: (a) the edge transition at $\mu/U = 0.5$; (b) the O(2) transition at $(\mu/U)_c \approx 0.391$; (c) the superfluid phase for $\mu/U = (\mu/U)_c$. For the edge transition in panel (a), the chosen hopping energies are $2\,d\,J/U = 0.1$ (red lines) and $2\,d\,(J/U)_c \approx 0.154$ (blue lines); for the O(2) transition in panel (b), the hopping energies are $2\,d\,J/U = 0.14$ (red lines) and $2\,d\,(J/U)_c^{\max} \approx 0.162$ (blue lines). The hopping energy in the SF phase in panel (c) is $2\,d\,J/U = 0.172$. The particle (hole) and spin (density) characters of gapped excitations, as well as the physical nature of the Goldstone modes, are indicated explicitly by the labels "p" ("h") and "s" ("d"), respectively.

whose maximal value gives the location of the tip of the CFSF lobe. As a reference, for $U_{12}/U = 0.9$, the location of the tip of the $n_{0,d} = 1$ CFSF lobe shown in Fig. 1(b) is located at $\{(\mu/U)_c \approx 0.391, 2\,d\,(J/U)_c^{\max} \approx 0.164\}$.

In Fig. 3(a), we show how the excitation spectrum appears as the second-order CFSF-to-SF transition for $n_{0,d} = 1$ and $U_{12}/U = 0.9$ is crossed through the edge point at $2\,d\,(J/U)_c \approx 0.154$ at $\mu/U = 0.5$. Above (below) the tip chemical potential, the gap is closed by the lowest of particle (hole) band in the density channel, while the other bands remain gapped. More specifically, the particle band in the spin channel corresponding to Eq. (29), which in this case at $x = 0$ is the third in ascending order, never participates in the gap closure. In Fig. 3(b), we consider the evolution of the band structure across the tip transition at $2\,d\,(J/U)_c^{\max} \approx 0.162$ for $U_{12}/U = 0.9$ and $(\mu/U)_c \approx 0.391$, while holding $n_{0,d} = 1$ fixed. Indeed, we observe that at the transition point the gaps of the two lowest-energy bands in the density channel vanish, while the remaining modes retain a finite gap. In the SF region, illustrated in Fig. 3(c), the two lowest bands in the density channel participate in the creation of the density Goldstone and the density Higgs excitation, respectively. The spin Goldstone mode emerges from the non-dispersive antipair band as described before.

### 2.2.2 Attractive interaction ($U_{12} < 0$)

(*Mott Insulator*) – In Figs. 4(a)-(b), we show the ground state phase diagram for two different values of $U_{12} < 0$. For attractive interactions, the MI lobes present the same ground state and spectral properties as their repulsive counterparts, with the spin/density character of the excitation modes being reversed. Furthermore, the MI-to-SF criticality is found again to be of either first or second order. However, at odds with the repulsive case, the first-order boundaries appear initially at small $J/U$ rather than close to the tip, as shown in Fig. 4(b) for $U_{12}/U = -0.4$, eventually spanning the entire lobe boundary, as shown in Fig. 4(b) for $U_{12}/U = -0.7$. In addition, the boundary between the vacuum lobe [green area] and the superfluid phase becomes also a first-order transition line. Along this boundary, the density Goldstone mode acquires a purely quadratic dispersion $\omega_{d,\mathbf{k}} \propto \mathbf{k}^2$ at small momenta, as shown in Fig. 4(d), which indicates the vanishing of the sound velocity of density excitations ($c_d \to 0$), while the spin sound

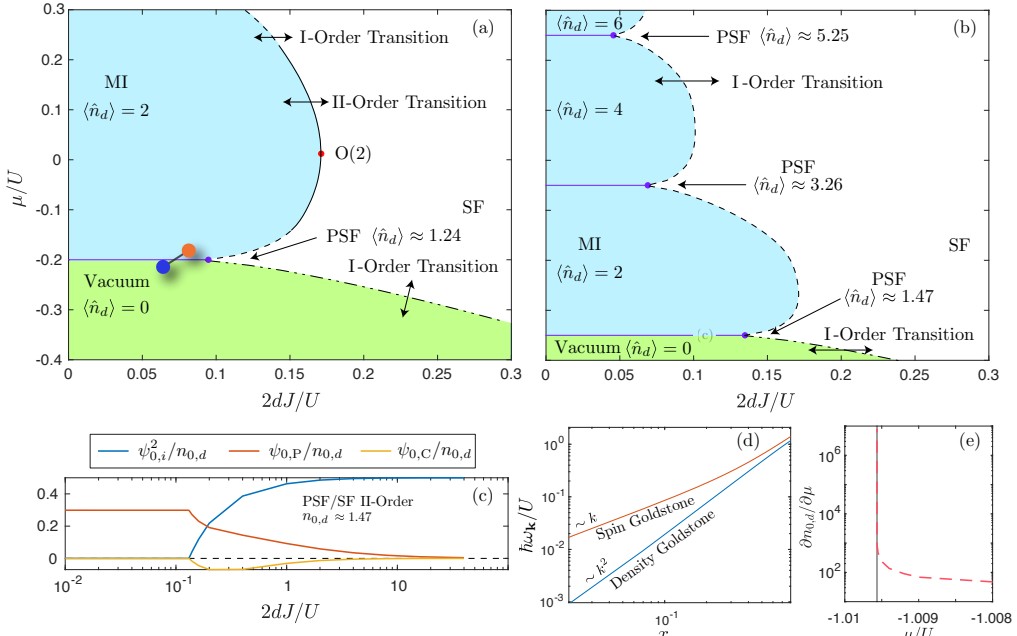

Figure 4: Ground state phase diagram for attractive interspecies interactions (a) $U_{12}/U = -0.4$ and (b) $U_{12}/U = -0.7$, with the order of the transition lines and O(2) critical points within the MI lobes indicated as in Fig. 1. The PSF phase is identified by the purple horizontal lines, with total filling $n_{0,d}$ evaluated in the $\eta \to 0^+$ limit. The dot-shaped illustrations depict the particle-particle pairs that develop between the two components of the gas in the PSF phase. (c) Behavior of the order parameters along the PSF-to-SF transition line at the border between the vacuum (green-shaded area) and the $n_{0,d} = 2$ MI lobes for $U_{12}/U = -0.7$ and fixed $n_{0,d} \approx 1.47$. (d)-(e) Excitation spectrum and compressibility near the first-order vacuum to superfluid transition for $\mu/U \approx -1.00956$, $2\,d\,J/U = 1$, and $U_{12}/U = -0.6$ with critical filling $n_{0,d} \approx 0.239$. The log-log scale of the vertical axis of panel (d) reveals the quadratic power law of the density Goldstone mode at all momenta.

velocity remains finite ($c_s > 0$). Accordingly, the (mean-field) compressibility $\partial n_{0,d}/\partial \mu$ diverges due to the discontinuity in the filling, as shown in Fig. 4(e). These behaviors indicate that the system *collapses* along the first-order vacuum-to-SF transition boundary. Along this line, the critical filling decreases for increasing $2\,d\,J/U$, vanishing eventually in the deep SF regime (not shown) where the transition becomes again of second order. As $U_{12}/U$ becomes more attractive, the collapse transition line extends towards larger values of $2\,d\,J/U$.

(*Pair Superfluid*) – The MI lobes shown in Figs. 4(a)-(b) are separated by sharp transition lines [purple horizontal lines], extending towards increasingly large values of $2\,d\,J/U$ as $U_{12}$ becomes more and more attractive. On these lines at the boundary of the $n_{0,d} = 2j$ and $n_{0,d} = 2(j+1)$ MI lobes, one finds that the states with the lowest eigenenergies $|j+1, j+1\rangle$ and $|j, j\rangle$ are degenerate, thus corresponding to a PSF phase with a finite pair coherence $\psi_{0,P} \neq 0$. When imposed in the 2GE, such degeneracy condition pinpoints the location of the PSF lines on a discrete set of critical chemical potentials $(\mu/U)_c$ through the equation

$$-U_{12} - 2j(U + U_{12}) + 2\mu_c = 0, \qquad (32)$$

valid for any non-negative integer $j$. Thus, in general the ground state is given by the superposition $\alpha |j+1, j+1\rangle + \beta |j, j\rangle$, where the coefficients $\alpha$ and $\beta$ are restricted to lie on the unit

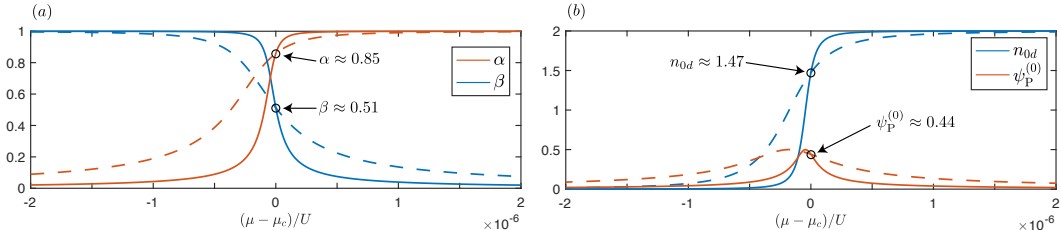

Figure 5: Variation of (a) $n_{0,d}$ and $\psi_{0,\mathrm{P}}$ and (b) $\alpha$ and $\beta$ as functions of $\mu/U$ in the vicinity of the PSF line at $(\mu/U)_c = -0.35$ for fixed $2dJ/U = 0.12$ and $U_{12}/U = -0.7$. Solid (dashed) lines correspond to the symmetry-breaking offset $\eta = 10^{-3}(2 \times 10^{-3})$.

circle $|\alpha|^2 + |\beta|^2 = 1$. However, these coefficients are *undetermined* within the usual Gutzwiller approximation, as the individual states are already $\mathbb{Z}_2$-symmetric.

In the QMC results of Ref. [25], $\alpha$ and $\beta$ were found to vary with $\mu$ at fixed $U_{12}$ and $J$. The determination of $\alpha$ and $\beta$ in the mean-field Gutzwiller theory requires the introduction of an *ad hoc* perturbative order parameter $\psi_{0,i} = \eta > 0$ along the PSF line mimicking the tunneling of residual background fluctuations, such that ground states with broken U(1) symmetry can be accessed. As we show in Fig. 5(a), this turns out to uniquely fix $\alpha$ [orange lines] and $\beta$ [light blue lines] along the PSF lines described by Eq. (32) *regardless* of the value of $\eta$, provided that $\eta$ remains sufficiently small. In particular, for the PSF line separating the vacuum region from the $n_{0,d} = 2$ MI lobe, we obtain $\alpha \approx 0.858$ and $\beta \approx 0.513$, from which we derive $n_{0,d} \approx 1.47$. Similarly, for the PSF line separating the $n_{0,d} = 2$ and $n_{0,d} = 4$ MI lobes, we find $\alpha \approx 0.794$ and $\beta \approx 0.608$, which gives $n_{0,d} \approx 3.260$ in the PSF phase. In this way, we reliably obtain a PSF ground state with a well-defined value of the pair order parameter $\psi_{0,\mathrm{P}} \neq 0$.

The symmetry-breaking offset $\eta \neq 0$ makes the PSF phase line to develop a finite width, as shown in Fig. 5(b), where the variation of $n_{0,d}$ and $\psi_{0,\mathrm{P}}$ is studied for fixed $\mu/U$. We find that $n_{0,d}$ [light blue lines] changes continuously between the filling of the two neighboring MI lobes, which is qualitatively consistent with the QMC results of Ref. [25], with the major difference being the size of the transition width in the chemical potential $\mu/U$. Secondly, we also see that $\psi_{0,\mathrm{P}}$ [orange lines] behaves smoothly, vanishing identically as one enters the MI lobes and reaching a maximum in between. This reveals that the PSF-to-MI transitions are of second-order within the present approach. In general, the maximum of $\psi_{0,\mathrm{P}}$ is found to occur for $\mu/U$ below the exact PSF line, becoming increasingly shifted for larger $\eta$. Also, the maximum is located where $\alpha = \beta = 1/\sqrt{2}$ [black solid line], which can be seen easily by matching the results of the two panels of Fig. 5. However, we observe that not only $\alpha$ and $\beta$, but also $n_{0,d}$ and $\psi_{0,\mathrm{P}}$ are invariant with respect to the choice of the value of $\eta$ if taken precisely along the PSF line ($\mu - \mu_c = 0$). Moreover, as a function of $2dJ/U$, the width of the PSF region shrinks with decreasing $2dJ/U$, collapsing onto the PSF line in the strongly-interacting limit. Therefore, an infinitesimal symmetry-breaking perturbation $\eta \to 0^+$ can be safely applied for practical purposes (cf. Ref. [53]). In the remainder of this work, this limit is *assumed* whenever the PSF phase is discussed, in virtue of its essential insensitivity to the choice of $\eta$.

Along the PSF line, Eq. (13) can be treated analytically to obtain the excitation spectrum. For the $n_{0,d} \approx 1.47$ PSF phase separating the vacuum from the $n_{0,d} = 2$ MI lobe, one obtains

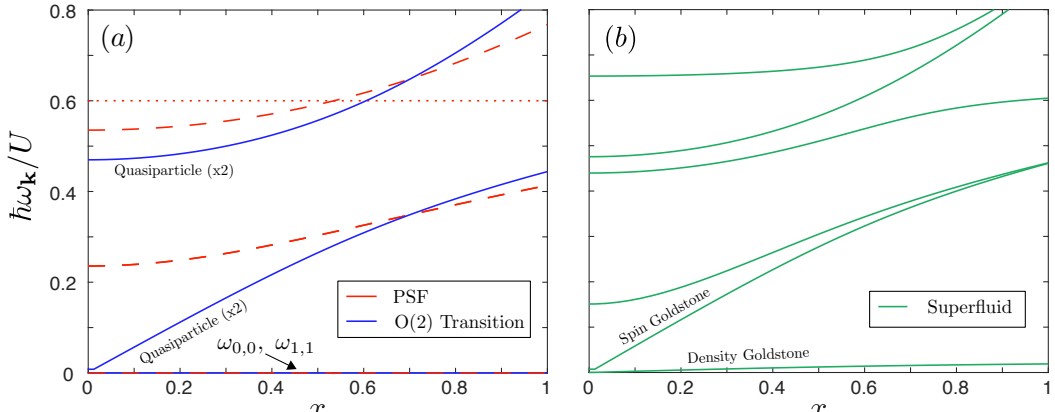

Figure 6: Excitation spectra (a) along the PSF-to-SF second-order transition with $n_{0,d} \approx 1.47$ and $(\mu/U)_c = -0.35$ and (b) in the SF phase for $\mu/U \approx -0.35$ and $U_{12}/U = -0.7$ in $d = 2$. In panel (a), the hopping energies are $2dJ/U = 0.08$ (red lines) and $2d(J/U)_c \approx 0.131$ (blue lines). In the superfluid phase, we consider $2dJ/U = 0.14$. The parentheses refer to the degeneracy of the hybridized quasiparticle bands.

the implicit equation

$$
\begin{aligned}
&\Big\{ J^2(\mathbf{k}) \big[ \big( 8\beta^4 - 8\beta^2 + 1 \big) \mu^2 + \big( 1 - 2\beta^2 \big)^2 U^2 + 2 \big( 4\beta^4 - 2\beta^2 - 1 \big) \mu U - \hbar^2 \omega_{\mathbf{k}}^2 \big] \\
&+ 2J(\mathbf{k}) \big[ \mu U^2 + 2\beta^2 \mu^2 U - 2 \big( \beta^2 - 1 \big) U \omega_{\mathbf{k}}^2 + \big( 2\beta^2 - 1 \big) \mu \big( \mu^2 - \hbar^2 \omega_{\mathbf{k}}^2 \big) \big] \\
&+ \big( \mu^2 - \hbar^2 \omega_{\mathbf{k}}^2 \big) \big[ (\mu + U)^2 - \hbar^2 \omega_{\mathbf{k}}^2 \big] \Big\}^2 = 0,
\end{aligned}
\tag{33}
$$

which describes two distinct gapped bands, both doubly degenerate and with a mixed quasiparticle characteristic. In analogy with the CFSF phase, all the remaining excitations have a flat band dispersion; in particular, the zero-energy modes $\hbar\omega_{0,0} = \hbar\omega_{1,1} = 0$ describe cost-free density fluctuations due to the formation of local particle-hole pairs in the PSF ground state predicted by Gutzwiller mean-field theory.

Once again, the second-order phase transition from the PSF to the SF phases is identified by the closure of the gap in the excitation spectrum. From Eq. (33), we find that along the $n_{0,d} \approx 1.47$ PSF line, the critical hopping strength is given by

$$
2d \left( \frac{J}{U} \right)_c = \frac{\mu}{U} \frac{1 + \mu/U}{\mu/U (2 - 2\beta^2) - (1 + \mu/U)(2\alpha\beta + 1)}.
\tag{34}
$$

In Fig. 6(a), we show how the excitation spectrum of the PSF phase of density $n_{0,d} \approx 1.47$ evolves as the second-order transition point $2d(J/U)_c \approx 0.131$ is reached for $U_{12}/U = -0.7$ along the critical line at $(\mu/U)_c = -0.35$. At the PSF-to-SF transition point, only the energy gap of the lowest quasiparticle branches vanishes, such that the band structure appears reminiscent of the MI-to-SF and CFSF-to-SF second-order tip transitions in Fig. 2(b) and Fig. 3(b) respectively but with the two gapless modes being degenerate. More distinctly, the O(2) nature of the PSF-to-SF transition is suggested in Fig. 6(b), where we observe that the gapless quasiparticle band splits into the Goldstone and Higgs modes active in the spin channel, while the density Goldstone mode emerges from the non-dispersive band $\omega_{0,0} = \omega_{1,1}$.

It is important to note that the flatness of the lowest spin bands $\omega_{0,1}$ and $\omega_{1,0}$ for CFSF and density bands $\omega_{0,0}$ and $\omega_{1,1}$ for PSF is an artifact resulting from the mean-field approximation. A more careful treatment of non-local fluctuations would alter these modes to produce linear Goldstone dispersions as a result of the broken $U(1)$ spin or density symmetries, respectively.

### 2.3 Quantum Gutzwiller theory for mixtures

In this subsection, we briefly review the quantum Gutzwiller (QGW) theory developed in Ref. [36] for the one-component BH model and extend it to the two-component case. The quantization of the Gutzwiller theory provides us with a simple tool for going beyond the mean-field approximation, which proves to be of key importance in determining the structure of non-trivial correlation functions and the superfluid components of the system.

Following the derivation outlined in Ref. [36], the essential idea behind the QGW method is to include quantum fluctuations on top of the Gutzwiller state via a canonical quantization [54, 55] of the coordinates of the Lagrangian (6). Specifically, we promote the Gutzwiller variables and their conjugate momenta to operators as $c_{n_1,n_2}(\mathbf{r}) \to \hat{c}_{n_1,n_2}(\mathbf{r})$ and impose equal-time commutation relations between them, namely

$$\left[\hat{c}_{n_1,n_2}(\mathbf{r}_1), \hat{c}^\dagger_{m_1,m_2}(\mathbf{r}_2)\right] = \delta_{\mathbf{r}_1,\mathbf{r}_2}\,\delta_{n_1,n_2}\,\delta_{m_1,m_2}. \tag{35}$$

We stress that, in the same way as the Gutzwiller ansatz (1) assigns a weight to each local configuration, the corresponding Gutzwiller operators cannot be decomposed into single-species operators without overlooking a relevant fraction of the interspecies correlations: this sharply contrasts with Bogoliubov's theory [34, 56], where the quantum fields of different species are always treated separately. It follows that the two-component QGW approach can take into accurate account *local* pair correlations, while quantization allows for an approximate view on the *non-local* quantum correlations missed by mean-field theory.

In analogy with the number-conserving approaches in dilute ultracold atomic gases [57, 58], the next step consists in expanding the operators $\hat{c}_{n_1,n_2}$ around their ground state $\mathbb{C}$-values $c^0_{n_1,n_2}$ as

$$\hat{c}_{n_1,n_2}(\mathbf{r}) = \hat{A}(\mathbf{r})\,c^0_{n_1,n_2} + \delta\hat{c}_{n_1,n_2}(\mathbf{r}). \tag{36}$$

The normalization operator $\hat{A}(\mathbf{r})$ is a functional of the fluctuation fields $\delta\hat{c}_{n_1,n_2}(\mathbf{r})$ and is defined in order to ensure that the physical constraint $\sum_{n_1,n_2} \hat{c}^\dagger_{n_1,n_2}(\mathbf{r})\,\hat{c}_{n_1,n_2}(\mathbf{r}) = \hat{\mathbb{1}}$ is fulfilled exactly. By restricting ourselves to local fluctuations orthogonal to the ground state $\sum_{n_1,n_2} \delta\hat{c}^\dagger_{n_1,n_2}(\mathbf{r})\,c^0_{n_1,n_2} = 0$ as usual, one has

$$\hat{A}(\mathbf{r}) = \left[\hat{\mathbb{1}} - \sum_{n_1,n_2} \delta\hat{c}^\dagger_{n_1,n_2}(\mathbf{r})\,\delta\hat{c}_{n_1,n_2}(\mathbf{r})\right]^{1/2}. \tag{37}$$

Physically, $\hat{A}(\mathbf{r})$ in Eqs. (36)-(37) serves to deplete the weight of the ground state parameters $c^0_n$ in response to the onset of quantum fluctuations. This role is especially important in determining quantum correlations in the strongly-interacting regime.

The Fourier transform of the Gutzwiller operators is given by

$$\delta\hat{c}_{n_1,n_2}(\mathbf{r}) \equiv \frac{1}{\sqrt{I}} \sum_{\mathbf{k}} e^{i\mathbf{k}\cdot\mathbf{r}}\,\delta\hat{C}_{n_1,n_2}(\mathbf{k}). \tag{38}$$

Inserting Eq. (38) in the quantized Gutzwiller Hamiltonian

$$\hat{H}_{QGW} = \sum_{\mathbf{r}} \left\{ -J \sum_{i=1}^{2} \sum_{j=1}^{d} \left[\hat{\psi}^\dagger_i(\mathbf{r})\,\hat{\psi}_i(\mathbf{r}+\mathbf{e_j}) + \text{h.c.}\right] + \sum_{n_1,n_2} H_{n_1,n_2}\,\hat{c}^\dagger_{n_1,n_2}(\mathbf{r})\,\hat{c}_{n_1,n_2}(\mathbf{r}) \right\}, \tag{39}$$

and keeping only terms up to the quadratic order in the fluctuations, we obtain

$$\hat{H}^{(2)}_{QGW} = E_0 + \frac{1}{2} \sum_{\mathbf{k}} [\delta\underline{\hat{C}}^\dagger(\mathbf{k}), -\delta\underline{\hat{C}}(-\mathbf{k})]\,\hat{\mathcal{L}}_{\mathbf{k}} \begin{bmatrix} \delta\underline{\hat{C}}(\mathbf{k}) \\ \delta\underline{\hat{C}}^\dagger(-\mathbf{k}) \end{bmatrix}, \tag{40}$$

where $E_0 = \hbar\omega_0 + zJ \sum_{i=1}^{2} |\psi_{0,i}|^2$ is the mean-field ground state energy, the vector $\delta\underline{\hat{C}}(\mathbf{k})$ collects the Fock components $\delta\hat{C}_{n_1,n_2}(\mathbf{k})$ and $\hat{\mathcal{L}}_{\mathbf{k}}$ coincides with the pseudo-Hermitian matrix introduced in Eq. (13). The quadratic form (40) can be easily diagonalized by a suitable Bogoliubov rotation of the Gutzwiller operators in terms of the many-body excitation modes of the system,

$$\delta\hat{C}_{n_1,n_2}(\mathbf{k}) = \sum_{\alpha} u_{\alpha,\mathbf{k},n_1,n_2} \, \hat{b}_{\alpha,\mathbf{k}} + \sum_{\alpha} v^*_{\alpha,-\mathbf{k},n_1,n_2} \, \hat{b}^\dagger_{\alpha,-\mathbf{k}} \,. \tag{41}$$

This allows a recasting of the Hamiltonian as

$$\hat{H}^{(2)}_{QGW} = \sum_{\alpha} \sum_{\mathbf{k}} \hbar\omega_{\alpha,\mathbf{k}} \, \hat{b}^\dagger_{\alpha,\mathbf{k}} \hat{b}_{\alpha,\mathbf{k}} \,, \tag{42}$$

where each mode $\hat{b}_{\alpha,\mathbf{k}}$ corresponds to a different collective excitation with frequency $\omega_{\alpha,\mathbf{k}}$, labeled by its branch index $\alpha$ and momentum $\mathbf{k}$. Bosonic commutation relations between the annihilation and creation operators $\hat{b}_{\alpha,\mathbf{k}}$ and $\hat{b}^\dagger_{\alpha,\mathbf{k}}$,

$$\left[ \hat{b}_{\alpha,\mathbf{k}}, \hat{b}^\dagger_{\beta,\mathbf{p}} \right] = \delta_{\mathbf{k},\mathbf{p}} \, \delta_{\alpha,\beta} \,, \tag{43}$$

are enforced by choosing the usual Bogoliubov normalization condition

$$\underline{u}^*_{\alpha,\mathbf{k}} \cdot \underline{u}_{\beta,\mathbf{k}} - \underline{v}^*_{\alpha,-\mathbf{k}} \cdot \underline{v}_{\beta,-\mathbf{k}} = \delta_{\alpha,\beta} \,. \tag{44}$$

The effective, quadratic description of the two-component BH model in terms of its quasi-particle excitations provides a simple and versatile tool for the computation of any expectation value. Drawing on the quantization procedure described above and the Bogoliubov rotation (41), the evaluation of a generic observable $\left\langle \hat{O}\!\left[ \hat{a}_{i,\mathbf{r}}, \hat{a}^\dagger_{i,\mathbf{r}} \right] \right\rangle$ amounts to the application of a straightforward calculation protocol first outlined in Ref. [36] and included for completeness in App. A.1. Briefly, in a similar fashion as the QGW quantization of the BH Hamiltonian (39), each observable is projected along its Gutzwiller representation $\hat{\mathcal{O}}\!\left[ \hat{c}, \hat{c}^\dagger \right]$ and expanded in terms of the quantum fluctuations. Therefore, expectation values are calculated via the simple application of Wick's theorem to the quantized excitations $\hat{b}_{\alpha,\mathbf{k}}$ with respect to the ground state of the system.

In the following section, the QGW quantization protocol is applied to calculate a range of experimentally relevant one-body and two-body observables in the two-component BH model. These observables are calculated always assuming a zero-temperature vacuum of the quasi-particle modes of the theory $|\Omega\rangle$, formally defined by $\hat{b}_{\alpha,\mathbf{k}}|\Omega\rangle = 0$ for each $(\alpha, \mathbf{k})$. We clarify that the quantization procedure does not lead to a substantial change in the ground state and spectral properties discussed in Secs. 2.2.1-2.2.2, but rather yields a transparent formalism for accessing quantum fluctuations in a systematic manner. In order to make our discussion of the QGW result consistent, from now on we calculate the lattice filling $n_i$ by always including the second-order quantum corrections as detailed in App. A. We refer the interested reader to this appendix for further details on the application of the quantization protocol to a specific observable.

## 3 Correlation Functions

In this section, we apply the QGW approach to calculate several relevant correlation functions in two dimensions: the density-density and spin-spin structure factors (Sec. 3.1), the current-current correlation functions and associated superfluid densities (Sec. 3.2), the coherence function (Sec. 3.3), and the local density-density and spin-spin fluctuations (Sec. 3.4).

With the QGW we develop a formalism that can be straightforwardly applied to study these quantities across the entire phase diagram of the two-component BH model. In the following discussion, we provide directly the semi-analytical expressions as predicted by the QGW evaluation protocol, whose derivation is explicitly carried out in Apps. A and B.

## 3.1 Density and spin response

As a first application of the QGW approach, we investigate the role of quantum fluctuations in the linear response of the two-component BH system to density/spin probes. The density and spin response of an ultracold system reflects the underlying correlations and collective modes of the system, and can be probed experimentally using a variety of methods, e.g. Bragg scattering (cf. Ref. [34]). In the present case, the spin and density susceptibilities represent important tools in differentiating the CFSF and PSF transitions, where density and spin degrees of freedom are expected to separate. In general terms, we consider the effect of an external field applied at a fixed frequency $\omega$. The local operator associated with the perturbation is denoted by $\hat{G}_{\mathbf{r}}$, while we denote by $\hat{F}_{\mathbf{r}}$ the local operator whose linear response dynamics is under study.

**Dynamic structure factors**    In a two-component system, the most interesting response functions are the density and the spin or magnetisation response functions, corresponding to $\hat{G}_{\mathbf{r}} = \hat{F}_{\mathbf{r}} = \hat{n}_{1,\mathbf{r}} + \hat{n}_{2,\mathbf{r}}$ and $\hat{G}_{\mathbf{r}} = \hat{F}_{\mathbf{r}} = \hat{n}_{1,\mathbf{r}} - \hat{n}_{2,\mathbf{r}}$, respectively. Such response functions are related to the density and spin structure factors of the system. Let us start from the species-resolved density response function, i.e. $\hat{G}_{\mathbf{r}} = \hat{F}_{\mathbf{r}} = \hat{n}_{i,\mathbf{r}}$, which, up to second-order in the quantum fluctuations, reads (see App. B.2 for the derivation details)

$$\chi_{\hat{n}_i}(\mathbf{q}, \omega) = \frac{2}{\hbar} \sum_\alpha \frac{N_{i,\alpha,\mathbf{q}}^2 \, \omega_{\alpha,\mathbf{q}}}{(\omega + i\,0^+)^2 - \omega_{\alpha,\mathbf{q}}^2}, \tag{45}$$

at zero temperature. The response functions for the total density and spin channels are obtained by simple extensions of Eq. (45), namely

$$\chi_{\hat{n}_d}(\mathbf{q}, \omega) = \frac{2}{\hbar} \sum_\alpha \frac{\left(N_{1,\alpha,\mathbf{q}} + N_{2,\alpha,\mathbf{q}}\right)^2 \omega_{\alpha,\mathbf{q}}}{(\omega + i\,0^+)^2 - \omega_{\alpha,\mathbf{q}}^2}, \tag{46}$$

and

$$\chi_{\hat{n}_s}(\mathbf{q}, \omega) = \frac{2}{\hbar} \sum_\alpha \frac{\left(N_{1,\alpha,\mathbf{q}} - N_{2,\alpha,\mathbf{q}}\right)^2 \omega_{\alpha,\mathbf{q}}}{(\omega + i\,0^+)^2 - \omega_{\alpha,\mathbf{q}}^2}, \tag{47}$$

respectively. At zero temperature, the imaginary part of the response functions is proportional to the corresponding dynamic structure factors via the relation $S_{\hat{F}}(\mathbf{q}, \omega) = -\Im[\chi_{\hat{F}}(\mathbf{q}, \omega)/\pi]$. Useful information on and from the dynamical structure factors is provided by their energy momenta, a.k.a. sum rules

$$m_{\hat{F}}^p(\mathbf{q}) = \int_0^{+\infty} d\omega\, \omega^p\, S_{\hat{F}}(\mathbf{q}, \omega). \tag{48}$$

In particular, $m_{\hat{F}}^0(\mathbf{q}) = S_{\hat{F}}(\mathbf{q})$ is the so-called static structure factor. Within our approximation in Eqs. (45)-(47), the dynamic structure factors are simply given by a sum of Dirac delta functions at the values of the quasiparticle energies $\omega_{\alpha,\mathbf{q}}$,

$$S_{\hat{n}_d}(\mathbf{q}, \omega) = \frac{1}{\hbar} \sum_\alpha \left(N_{1,\alpha,\mathbf{q}} + N_{2,\alpha,\mathbf{q}}\right)^2 \left[\delta\left(\omega - \omega_{\alpha,\mathbf{q}}\right) - \delta\left(\omega + \omega_{\alpha,\mathbf{q}}\right)\right], \tag{49}$$

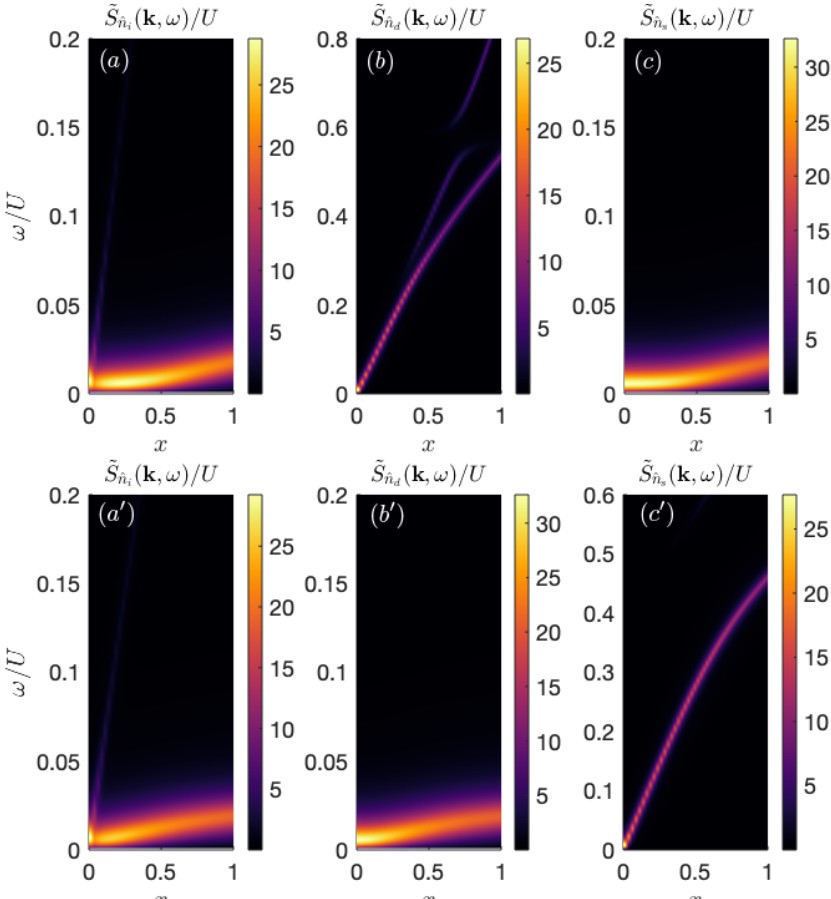

Figure 7: Normalized dynamical structure factors $\tilde{S}_{\hat{F}}(\mathbf{k}, \omega) = S_{\hat{F}}(\mathbf{k}, \omega)/S_{\hat{F}}(\mathbf{k})$ for (a)-(a′) the one-component density channel $\hat{F}_{\mathbf{r}} = \hat{n}_{1,\mathbf{r}}$, (b)-(b′) the total density channel $\hat{F}_{\mathbf{r}} = \hat{n}_{1,\mathbf{r}} + \hat{n}_{2,\mathbf{r}}$ and (c)-(c′) the spin channel $\hat{F}_{\mathbf{r}} = \hat{n}_{1,\mathbf{r}} - \hat{n}_{2,\mathbf{r}}$ in the SF phase in the vicinity of CFSF (a)-(c) and PSF (a′)-(c′) transitions for $d = 2$. For panels (a)-(c), the parameters are $U_{12}/U = 0.9$, $\mu/U = 0.391$, $2dJ/U = 0.172$ and $n_d = 1$, see Fig. 3(c). For (a′)-(c′) panels, the parameters are $U_{12}/U = -0.7$, $\mu/U \approx -0.35$, $2dJ/U = 0.14$ and $n_d \approx 1.47$, see Fig. 6(c).

$$S_{\hat{n}_s}(\mathbf{q}, \omega) = \frac{1}{\hbar} \sum_{\alpha} \left(N_{1,\alpha,\mathbf{q}} - N_{2,\alpha,\mathbf{q}}\right)^2 \left[\delta(\omega - \omega_{\alpha,\mathbf{q}}) - \delta(\omega + \omega_{\alpha,\mathbf{q}})\right], \tag{50}$$

and therefore the sum rules are easily determined.

In Ref. [32], the normalized dynamical structure factor $\tilde{S}_{\hat{F}}(\mathbf{q}, \omega) = S_{\hat{F}}(\mathbf{q}, \omega)/S_{\hat{F}}(\mathbf{q})$ (cf. Ref. [34]) has been analyzed in the SF regime for both repulsive and attractive interspecies interactions, for which it was found that: (i) the low-momentum part of the gapped modes does not respond significantly to any of the density-type probes, in agreement with the single-component case [29, 47]; (ii) the density and spin Goldstone modes are strongly excited by the total density and spin probes, respectively; (iii) the single-species density fluctuations are strongest for the lowest gapless mode. In [32] the density response in the PSF and CFSF regimes were also considered, although for an effective Hamiltonian including *ad hoc* perturbative hopping processes in the strongly-interacting limit of the model.

In Fig. 7, we show our results for the normalized dynamical structure factors in the immediate vicinity of the CFSF (a)-(c) and PSF (a′)-(c′) transitions, confirming the qualitative picture of Ref. [32] and displaying detailed signatures of the pairing phase transitions. These calculations have been performed for a total filling $n_d = 1$ which includes the second-order

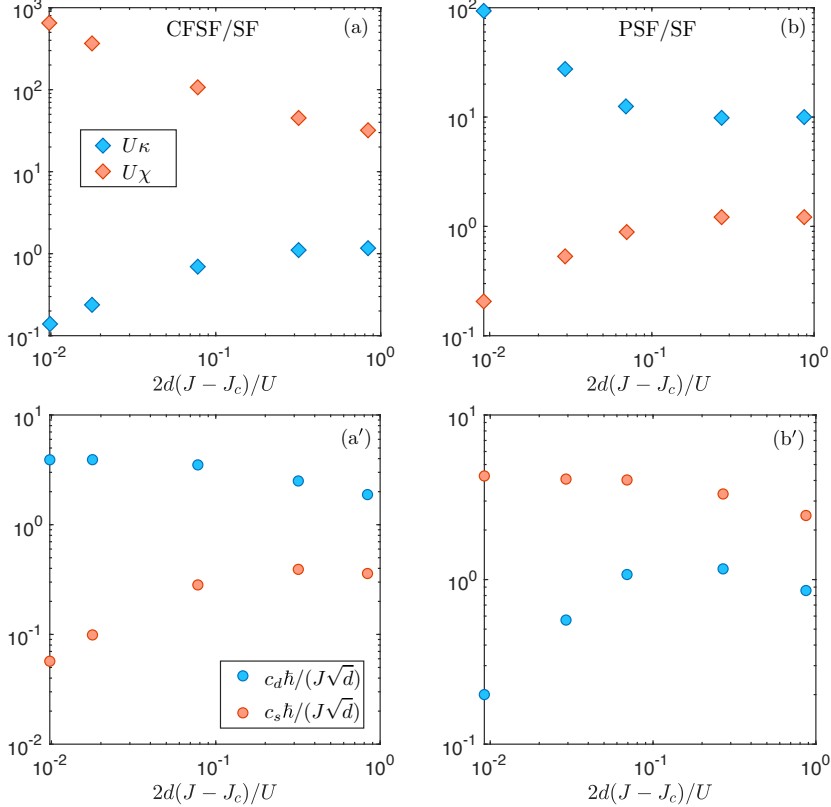

Figure 8: Upper panels: dependence of the compressibility (blue diamonds) and susceptibility (red diamonds) in the vicinity of the O(2) (a) CFSF and (b) PSF transitions. Lower panels: dependence of the density (blue circles) and spin (red circles) sound velocities close to the same transitions. The data in (a) and (a′) correspond to fixed $n_d = 1$ and $U_{12}/U = 0.9$, while the data in (b) and (b′) are derived for fixed $n_d \approx 1.47$ and $U_{12}/U = -0.7$. The calculations are always performed for $d = 2$.

quantum corrections to the mean-field filling $n_{0,d}$ as detailed in App. A.3. We carry this notation throughout the remainder of the work. Starting from panels (a)-(a′), we illustrate that single-species perturbations are sufficient for testing the proximity of the antipaired and paired phases, as $\tilde{S}_{\hat{F}}(\mathbf{q}, \omega)$ receives a dominant contribution from the lowest-lying Goldstone mode, while both the density and spin Goldstone excitations are found to have approximately the same weight in the deep SF region. More precisely, close to the CFSF (PSF) transition, the spin (density) mode (which softens at the transition) dominates the system response and enhances the amplitude of the structure factor at low energies for all momenta. Moving to the total density channel, in panel (b) we see that, despite the CFSF phase being dominated by spin fluctuations, the structure factor receives an increasing contribution by the density Goldstone mode at low momenta as the critical point is approached; as expected, the same mode controls entirely the total density response of the system close to the PSF transition, depicted in panel (b′). The situation is reversed in the case of the spin channel, considered in panels (c)-(c′): here, the projection of the structure factor over the spin Goldstone mode acts as a marker of both the CFSF and the PSF transitions, albeit on different momenta and energy ranges.

**Static response and sum rules** The energy moments of the dynamic structure factors allow to obtain a number of important quantities and identities that characterize the system. First of

all, the uniform limit of the inverse-energy-weighted sum rule $m_{\hat{F}}^{-1}(\mathbf{q})$ gives the static response of the system to the selected perturbation. Using the QGW expressions of Eq. (46), we obtain the relations

$$m_d^{-1}(\mathbf{q}) = \frac{1}{\hbar}\sum_\alpha \frac{\left(N_{1,\alpha,\mathbf{q}} + N_{2,\alpha,\mathbf{q}}\right)^2}{\omega_{\alpha,\mathbf{q}}} \underset{\mathbf{q}\to 0}{=} \frac{\kappa}{2}, \tag{51}$$

$$m_s^{-1}(\mathbf{q}) = \frac{1}{\hbar}\sum_\alpha \frac{\left(N_{1,\alpha,\mathbf{q}} - N_{2,\alpha,\mathbf{q}}\right)^2}{\omega_{\alpha,\mathbf{q}}} \underset{\mathbf{q}\to 0}{=} \frac{\chi}{2}, \tag{52}$$

for the compressibility $\kappa$ and the susceptibility $\chi$ of the two-component BH system, where the subscripts $d$ and $s$ are just shorthand notations for $\hat{F} = \hat{G} = \hat{n}_{d/s}$. In particular, we note that the former compressibility relation generalizes the result of Ref. [29] to mixtures.

Our results for the compressibility and the spin susceptibility are shown in Fig. 8 in the vicinity of the CFSF transition [panel (a)] and the PSF transition [panel (b)]. Notably, we find that the spin susceptibility (compressibility) diverges near the CFSF (PSF) transition. Close to the CFSF regime, this finding parallels the decreasing energy cost to produce spin excitations (see Fig. 3), which corresponds to an increase in the response of the system towards magnetic perturbations. On the other hand, in the PSF regime, the divergence of the compressibility corresponds to the decreasing energy of density excitations (see Fig. 6), hence the increasing sensitivity of the system to density fluctuations. Additionally, we find that the compressibility (susceptibility) tends to vanish near the CFSF (PSF) transition due to the opening of a spectral gap in the density (spin) channel.

The divergence of the static response functions – which would suggest an instability of the system towards phase separation or collapse – is due the lack of a proper inclusion of pairing quantum correlations when describing the CFSF and the PSF phases within the Gutzwiller approximation: specifically, it is simply related to the presence of a zero-energy flat dispersion relation for the density and the spin modes, respectively. As shown in Fig. 8(a')-(b'), the appearance of such modes reflects into the vanishing behaviour of the spin (density) sound velocities at the CSF (PSF) critical points.

An explicit relationship between sound velocities and static response functions can be directly uncovered by making use again of sum rules. Indeed, upon approaching the CFSF (PSF) phase, the low-momentum response function is exhausted by the spin (density) Goldstone mode, as shown in Fig. 7. Thus, the sum rules satisfy the general relation

$$m_{d(s)}^p(\mathbf{q}\to 0) \simeq \omega_{d(s),\mathbf{q}}^{p-k} m_{d(s)}^k \simeq \left(c_{d(s)}|\mathbf{q}|\right)^{p-k} m_{d(s)}^k(\mathbf{q}), \tag{53}$$

from which a number of identities can be derived. For instance, by considering $p = 0$ and $k = -1$, one can write the identities

$$m_d^0(\mathbf{q}, \omega) = \left(N_{1,d,\mathbf{q}} + N_{2,d,\mathbf{q}}\right)^2 \underset{\mathbf{q}\to 0}{=} \frac{\kappa}{2} c_d |\mathbf{q}|, \tag{54}$$

and

$$m_s^0(\mathbf{q}, \omega) = \left(N_{1,s,\mathbf{q}} - N_{2,s,\mathbf{q}}\right)^2 \underset{\mathbf{q}\to 0}{=} \frac{\chi}{2} c_s |\mathbf{q}|, \tag{55}$$

which we verified numerically and generalize the single-component identity given in Ref. [29] relating the Goldstone mode structure factor to the compressibility and the sound speed. The latter equations are moreover not independent. In fact, within the present Bogoliubov-like approach, the $p = 1$ sum rule (a.k.a. $f$-sum rule) is given its exact value [34]: $m_{d(s)}^1(\mathbf{q}) = \langle[\delta\hat{\rho}_{d(s),\mathbf{q}}, [\hat{H}, \delta\hat{\rho}_{d(s),\mathbf{q}}]]\rangle \propto \mathbf{q}^2$ for $|\mathbf{q}| \to 0$, where $\delta\hat{\rho}_{d(s),\mathbf{q}} = \hat{\rho}_{d(s),\mathbf{q}} - \langle\hat{\rho}_{d(s),\mathbf{q}}\rangle_{\mathrm{eq}}$ and $\hat{n}_{d(s),\mathbf{r}} = \sum_\mathbf{q} \hat{\rho}_{d(s),\mathbf{q}} e^{i\mathbf{q}\cdot\mathbf{r}}/\sqrt{I}$. From this, using the single-mode sum rule relation $m_{d(s)}^1(\mathbf{q}) = \omega_{d(s),\mathbf{q}}^2 m_{d(s)}^{-1}(\mathbf{q})$, we immediately infer that $c_{d(s)}^2 \propto 1/\kappa(1/\chi)$. Therefore, if the sound dispersion becomes flat, the corresponding static response must diverge.

## 3.2 Current response and superfluid components

We now investigate the role of quantum fluctuations in the linear response of the two-component BH system to current probes. The response of an ultracold system to current probes reflects its superfluid properties, which are expected to behave remarkably different in the CFSF and PSF phases due to a large collisionless superfluid drag between the two components [25].

In this subsection, we consider the transverse (i) *intraspecies* current response with $\hat{G}_{\mathbf{r}} = \hat{F}_{\mathbf{r}} = \hat{j}_i$ and (ii) *interspecies* current response with $\hat{F}_{\mathbf{r}} = \hat{j}_1$ and $\hat{G}_{\mathbf{r}} = \hat{j}_2$, evaluated at $\omega = 0$ along $x$-directed links of the square lattice. Here, $\hat{j}_i$ is the current operator referred to $i^{\text{th}}$ species taken in the uniform limit $\mathbf{q} \to \mathbf{0}$, that is

$$\hat{j}_{i,\mathbf{q}\to\mathbf{0}}|_x = \frac{\hbar}{m a} \sum_{\mathbf{k}} \sin(k_x a) \, \hat{a}^{\dagger}_{i,\mathbf{k}} \, \hat{a}_{i,\mathbf{k}} \,. \tag{56}$$

In App. B.3, the intra/interspecies current response functions are derived within the QGW formalism, giving the results

$$\chi^T_{\hat{j}_i,\hat{j}_i}(\mathbf{q}\to\mathbf{0}, \omega=0) = -\frac{\hbar}{(ma)^2} \sum_{\alpha,\beta} \sum_{\mathbf{k}} \frac{\left(U_{i,\alpha,\mathbf{k}} V_{i,\beta,\mathbf{k}} - U_{i,\beta,\mathbf{k}} V_{i,\alpha,\mathbf{k}}\right)^2}{\omega_{\alpha,\mathbf{k}} + \omega_{\beta,\mathbf{k}}} \sin^2(k_x a), \tag{57}$$

$$\chi^T_{\hat{j}_1,\hat{j}_2}(\mathbf{q}\to\mathbf{0}, \omega=0) = -\frac{\hbar}{(ma)^2} \sum_{\alpha,\beta} \sum_{\mathbf{k}} \frac{\prod_{i=1}^{2}\left(U_{i,\alpha,\mathbf{k}} V_{i,\beta,\mathbf{k}} - U_{i,\beta,\mathbf{k}} V_{i,\alpha,\mathbf{k}}\right)}{\omega_{\alpha,\mathbf{k}} + \omega_{\beta,\mathbf{k}}} \sin^2(k_x a), \tag{58}$$

respectively. The above equations naturally generalize the findings of Ref. [27] provided by the Bogoliubov approximation. In that case, one has only spin and density Goldstone modes given by Eq. (27) and consequently the current response functions satisfy the relation $\chi^T_{\hat{j}_1,\hat{j}_1}(\mathbf{q}\to\mathbf{0}, \omega=0) = -\chi^T_{\hat{j}_1,\hat{j}_2}(\mathbf{q}\to\mathbf{0}, \omega=0)$ exactly. Due to the presence of additional excitation bands with a substantial spectral weight for strong enough interactions, the very same equality is approximately fulfilled within the QGW approach only in the deep SF phase. Physically, the violation of the aforementioned identity reflects the breaking of Galilean invariance (c.f. Ref. [27]), which allows for a non-zero normal component of the gas contributing to superfluidity even at zero temperature, as we discuss in the following.

We proceed to introduce the relevant superfluid quantities in the two-species BH model. For a homogeneous two-component superfluid system, the relations between the mass current densities $m_i \mathbf{j}_i$ of each species and the velocities of the gas components are governed by the two-fluid model originally introduced in Ref. [11] and read

$$m_1 \mathbf{j}_1 = \rho_{n,1} \mathbf{v}_n + \rho_{s,1} \mathbf{v}_1 + \rho_{12} \mathbf{v}_2 \,, \tag{59a}$$

$$m_2 \mathbf{j}_2 = \rho_{n,2} \mathbf{v}_n + \rho_{s,2} \mathbf{v}_2 + \rho_{12} \mathbf{v}_1 \,, \tag{59b}$$

where $m_i$ is the bare mass of the $i^{\text{th}}$ species of the system, $\rho_{s,i}$ and $\mathbf{v}_i$ are respectively the superfluid (mass) densities and velocities for each component, $\rho_{12}$ is the so-called drag interaction and $\rho_{n,i}$ are the normal (mass) densities of the system, which are assumed to flow both at the same velocity $\mathbf{v}_n$. The physical role of the superfluid drag, also known as the Andreev-Bashkin effect, has a self-evident explanation: a superflow in one component can be induced by the collisionless drag from the superflow in the other component of the system and vice versa. For a continuous system, by Galilean invariance Eqs. (59) are supplemented by a close relationship $m_i N_i / V = \rho_{n,i} + \rho_{s,i} + \rho_{12}$ between the normal part of the system and the superfluid densities, such that at zero temperature, where $\rho_{n,i} = 0$, the whole volume density of the system $(N_1 + N_2)/V$ participates in the superfluid flow [59]. On a lattice, the breaking

of Galilean invariance requires the mass current densities to satisfy a different transformation rule in presence of a vector potential acting as a probe, such that the density $N_i/V$ is replaced by $-K_i/\left(2J\,a^d\right)$, where $K_i = \left\langle \hat{K}_i \right\rangle$ is the local kinetic energy referring to the $i^{\text{th}}$ species along the direction of the phase twist induced by the vector potential [27]. In particular, the local kinetic operator acting along $x$-directed links of a square lattice is given by

$$\hat{K}_i(\mathbf{r})|_x = -J\left(\hat{a}^\dagger_{i,\mathbf{r}+\mathbf{e}_x}\,\hat{a}_{i,\mathbf{r}} + \text{h.c.}\right). \tag{60}$$

As noted earlier, the normal component $n_{n,i}$ may *not* vanish at zero temperature on a lattice. Therefore, recovering a result derived in Ref. [21], for the two-component BH model we obtain

$$-\frac{K_i|_x}{2J} = n_{n,i} + n_{s,i} + n_{12}\,, \tag{61}$$

where we have rescaled the superfluid (normal) number densities $n_{s(n),i} \equiv a^d \rho_{s(n),i}/m$ and the drag $n_{12} \equiv a^d \rho_{12}/m$ to facilitate a direct comparison to the filling factors on a lattice. The average kinetic energy $K_i|_x$ within the QGW theory is calculated explicitly in App. B.4 and reads

$$K_i|_x \approx -2J\left|\psi_{0,i}\right|^2 - \frac{2J}{I}\sum_\alpha\sum_\mathbf{k}\left|V_{i,\alpha,\mathbf{k}}\right|^2\cos(k_x\,a). \tag{62}$$

To quantify how the superfluid drag impacts on the effective mass of the gas components, we define the dimensionless parameter $\xi_i^*$ through the equations

$$n_{n,i} + n_{s,i} = -\xi_i^*\frac{K_i|_x}{2J}\,, \tag{63}$$

$$n_{12} = -\frac{K_i|_x}{2J}\left(1-\xi_i^*\right)\,, \tag{64}$$

where $\xi^* = m/m^*$ reflects the extent to which the bare effective mass $m = \hbar^2/\left(2J\,a^2\right)$ on a lattice is renormalized by the interaction between the superfluid flows to produce the renormalized effective mass $m^*$. Explicitly, when $\xi_i^* > 1$ ($\xi_i^* < 1$) the renormalized effective mass is smaller (larger) than the bare effective mass. This extends the concept of the renormalized effective mass discussed in Refs. [11, 22] to the lattice, as Eq. (64) is exactly analogous to Eq. (4) in the second work. We note that in those works, translational invariance ensures that the renormalized effective mass remains always larger than the bare effective mass due to the guaranteed positivity of $n_{12}$. On a lattice, the inclusion of $n_{n,i}$ in Eq. (63), which is trivially absent for a continuous superfluid at zero temperature, adds additional complexity while ensuring that the variation of $\xi_i^*$ is due solely to the collisionless drag.

Having introduced the relevant superfluid components, we now resort to the formal results of Ref. [27] to relate them to the current-current response functions of the two-component BH model. Explicitly, one has

$$n_{s,i} = \frac{m}{I}\chi^T_{\hat{j}_i,\hat{j}_i}(\mathbf{q}\to\mathbf{0},\omega=0) - \frac{K_i|_x}{2J}\,, \tag{65}$$

$$n_{12} = \frac{m}{I}\chi^T_{\hat{j}_1,\hat{j}_2}(\mathbf{q}\to\mathbf{0},\omega=0)\,. \tag{66}$$

A third relation provides a sum rule for the total normal fraction of the system $n_n = n_{n,1} + n_{n,2}$ as the total transverse current response function, reading

$$n_n = -\frac{m}{I}\left[\chi^T_{\hat{j}_1,\hat{j}_1}(\mathbf{q}\to\mathbf{0},\omega=0) + \chi^T_{\hat{j}_2,\hat{j}_2}(\mathbf{q}\to\mathbf{0},\omega=0) + 2\,\chi^T_{\hat{j}_1,\hat{j}_2}(\mathbf{q}\to\mathbf{0},\omega=0)\right]. \tag{67}$$

From Eq. (67), we notice how, upon exchanging $\hat{j}_1$ with $\hat{j}_2$, opposite values of the current response functions $\chi^T_{\hat{j}_1,\hat{j}_1}(\mathbf{q}\to\mathbf{0},\omega=0) = -\chi^T_{\hat{j}_1,\hat{j}_2}(\mathbf{q}\to\mathbf{0},\omega=0)$ results in $n_n = 0$, as would

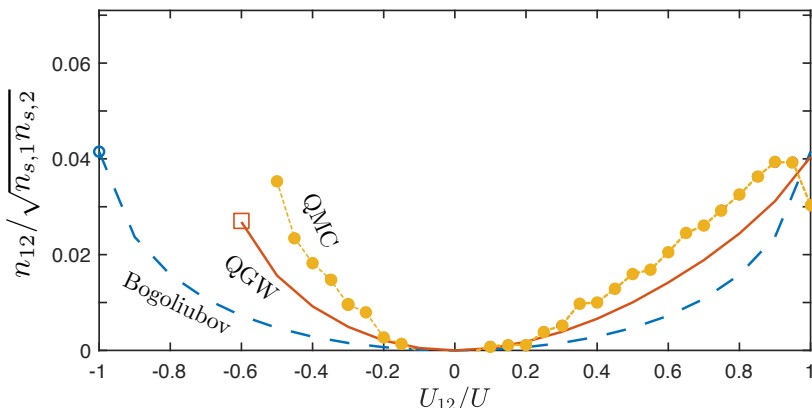

Figure 9: Collisionless drag versus the interspecies coupling strength $U_{12}/U$ for fixed $n_d \approx 0.5$ and $2\,d\,J/U = 0.4$ in $d = 2$. Here, we compare the QGW prediction (orange solid line) with the Bogoliubov result from Refs. [21, 27] (light-blue dashed line) and the QMC data (yellow dotted line) from Ref. [25] evaluated on a lattice of size $I = 10^2$. The hollow square and circle for the QGW and Bogoliubov results, respectively, indicate the point of the collapse.

be predicted by the Bogoliubov approximation, in clear contrast with the correct physics of a strongly-interacting lattice system at zero temperature.

It is worth mentioning that measuring the superfluid drag is a challenging task experimentally, however recently it has been proposed to use fast response to directly access the entrainment in continuous systems [27, 60]. We note that the same proposals can be applied in the presence of a lattice.

### 3.2.1 Superfluid regime

In the deep SF regime, the current response functions and the average kinetic energy match their expressions given by the mean-field Gutzwiller approximation, namely $\chi^T_{\hat{j}_i,\hat{j}_j}(\mathbf{q} \to \mathbf{0}, \omega = 0) = 0$ and $K_{0,i}|_x = -2J\left|\psi_{0,i}\right|^2$, respectively. This leads to equal superfluid and condensate densities, namely $n_{s,i} = \left|\psi_{0,i}\right|^2$, as well as a vanishing drag $n_{12} = 0$ and a trivial renormalization of the effective mass $\xi^* \approx 1$.

At intermediate $2\,d\,J/U$, the superfluid drag was calculated using quantum Monte Carlo (QMC) simulations in Ref. [25]. The comparison of the QGW results with the Bogoliubov predictions (see Refs. [21,27]) and the QMC data is shown in Fig. 9, for a hopping energy equal to $2\,d\,J/U = 0.4$ and a fixed total filling $n_d \approx 0.5$. We remark here that the accuracy of our drag estimation hinges on the correct calculation of the total filling, which must include second-order quantum corrections accounted for by the QGW theory (see App. A.3). For repulsive interactions, these corrections are always less than $\sim 10\%$ and therefore can be essentially neglected. By contrast, for attractive interactions quantum corrections increase for larger $|U_{12}|$ and can be as large as $\sim 25\%$ due to the diverging compressibility near the collapse transition (see Sec. 2.2.2). We find that including the quantum corrections has the effect of shifting the collapse transition towards more attractive $U_{12}$ than one would obtain holding instead the mean-field filling constant. This fact restricts the calculation of the drag to well before the Bogoliubov prediction for the collapse point ($U_{12}/U \leq -1$) [26,27].

The QGW results of Fig. 9 [orange solid line] are obtained on the same $I = 10^2$ lattice considered in the QMC calculations of Ref. [25] [yellow dotted line]. We find that in general

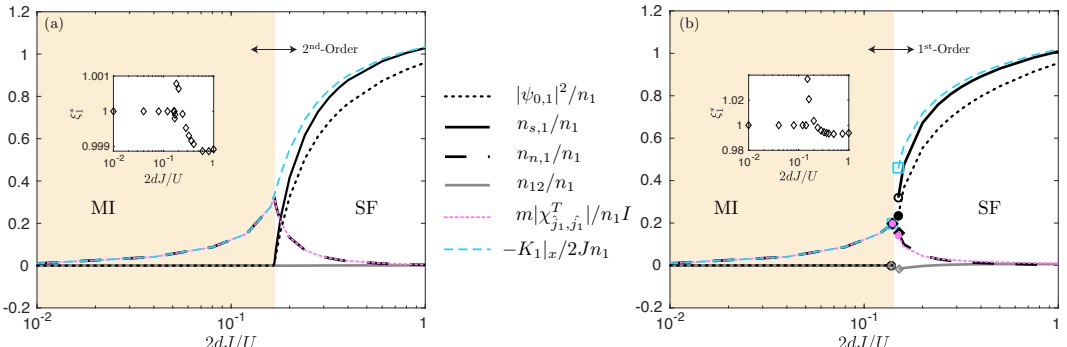

Figure 10: Transport quantities in two dimensions across the MI-to-SF second (left panel) and first (right panel) order transitions at fixed $\mu/U = 1.4$ and $\mu/U = 1$ for $U_{12}/U = 0.5$ and $U_{12}/U = 0.9$, corresponding to the $n_d = 2$ MI lobes in Fig. 1(a) and Fig. 1(b), respectively. The one-component densities $n_i$ include the corrections of the second-order quantum fluctuations. The tan-shaded area indicates the MI region. Thick black and grey lines indicate the QGW predictions for the superfluid density $n_{s,1}$ and the (vanishingly small) superfluid drag $n_{12}$ respectively, while the black dashed and dotted lines refer to the normal component $n_{n,1}$ and condensate fraction $|\psi_{0,1}|^2$. Pink dotted and blue dashed lines are the contributions to $n_{s,1}$ from the average kinetic energy $K_i|_x$ and the intraspecies current response function $\chi^T_{\hat{j}_i,\hat{j}_i}$, respectively. (Insets) Renormalization of the effective mass across the MI-to-SF transitions due to the collisionless drag.

the drag increases for smaller lattice sizes, in contrast to the results of that work. Furthermore, we also observe that the collapse point is shifted towards less attractive $U_{12}$ for smaller lattices. Additionally, we note that, regardless of the sign of $U_{12}$, the collisionless drag remains always positive in the SF regime, and consequently the renormalized effective mass is larger than the bare one. Physically, this indicates that the dressing effect of individual particle motions of one species due to the superfluid flow of the other component has a particle (rather than hole) character.

On the whole, the QGW predictions underestimate the QMC results, but qualitatively reproduce the asymmetry of the superfluid drag with respect to $U_{12}$. On the other hand, the Bogoliubov results [light-blue dashed line] are symmetric with respect to $U_{12}$, do not capture the first-order transition and lie well below of the QMC points. In this regard, the QGW approach can be viewed as a major improvement over the standard Bogoliubov treatment of quantum fluctuations in presence of strong correlations [21, 27, 35]. Indeed, it is well known that Bogoliubov's theory underestimates the current response functions, as it takes into account only the excitation vertex of the Goldstone modes in Eqs. (57)-(58) and neglects the contribution of all the other excitation modes, which acquire a sizeable spectral weight away from the deep SF limit.

### 3.2.2 Phase transitions for $U_{12} > 0$

In this subsection, we study the superfluid components across the various phase transitions characterized in Sec. 2.2.1 for repulsive interspecies interactions. To that end, we begin by discussing the first and second-order MI-to-SF transitions, passing to the second-order CFSF-to-SF transition in the second place.

(*Mott Insulator to Superfluid*) – In Fig. 10(a) and Fig. 10(b), we show the QGW results for

the relevant transport quantities, which include the superfluid components, the intraspecies current response, and the average kinetic energy across the second and first-order MI-to-SF transitions of the $n_{0,d} = 2$ lobes for $U_{12}/U = 0.5$ and $U_{12}/U = 0.9$, respectively. Qualitatively, the behavior of the superfluid fraction exhibits a number of features in common with the QGW results of Ref. [36] for the single-component BH model. Specifically, in the SF regime, the superfluid density $n_{s,i}$ [black solid line] remains always larger than the condensate fraction [black dotted line] and approaches the total density of the corresponding species in the deep SF limit. Furthermore, $n_{s,i}$ vanishes discontinuously (continuously) at the first-order (second-order) critical point and is exactly zero in the MI phase as in the single component case. As found in the single-component BH model [36], this latter feature is ensured by the exact cancellation between the current response function $\chi^T_{\hat{j}_i, \hat{j}_i}$ [pink dashed line], given by Eq. (57), and the contribution of zero-point fluctuations to the average kinetic energy $K_i|_x$ [cyan dashed line] provided by Eq. (62). As expected, both the quantities tend towards zero in the strongly-interacting limit $2\,d\,J/U \to 0$. The collisionless drag [gray solid line] remains always on the order of a few percent of $n_{s,i}$, reaching the maximal value close to the MI-to-SF critical point and vanishing entirely within the MI phase. Because the superfluid drag is small, the renormalization of the effective mass is also negligible and the normal components $n_{n,i}$ [black dashed line] are dominated by the contribution of the intraspecies current response function $\chi^T_{\hat{j}_i, \hat{j}_i}$, as one can see by comparing Eqs. (66)-(67).

It is now worth commenting on the relative weight of the various excitation vertices that contribute the most to the results displayed in Fig. 10. In the SF regime, the spin-density Goldstone vertex makes the largest contribution to the expression of $\chi^T_{\hat{j}_i, \hat{j}_i}$ in Eq. (57), followed by smaller terms coming from the vertex between the Goldstone mode and the Higgs mode with density character. This is in strong contrast to the single component case treated in Ref. [36], which showed that $\chi^T_{\hat{j}_i, \hat{j}_i}$ was nearly saturated by the Goldstone-Higgs vertex, since in this case only one Goldstone excitation is present. Similarly, the interspecies current response $\chi^T_{\hat{j}_1, \hat{j}_2}$, and therefore the superfluid drag, is nearly saturated by the vertex involving spin and density Goldstone modes. On the other hand, the zero-point fluctuations in the average kinetic energy $K_i|_x$ [cyan dashed line] are dominated by the contribution of the spin Goldstone mode, in addition to the mean-field effect of the order parameter $\psi_{0,i}$.

Inside the MI phase, the spectral summation of the intraspecies response $\chi^T_{\hat{j}_i, \hat{j}_i}$ is saturated by the vertices of particle-hole excitations, while the average kinetic energy $K_i|_x$ is dominated by the contribution of the lowest particle or hole bands, depending on the chemical potential. As the MI-to-SF transition is crossed, these excitations turn into the Goldstone modes, such that the spectral content of $K_i|_x$ changes smoothly across the criticality. An analogous reasoning applies to $\chi^T_{\hat{j}_i, \hat{j}_i}$. On the other hand, we note that the interspecies response $\chi^T_{\hat{j}_1, \hat{j}_2}$ is not fully saturated by the vertices involving only the first few particle-hole bands, requiring also the contribution of higher-energy excitations in order to give a perfect cancellation of the superfluid drag in the MI lobe.

(*Counterflow Superfluid to Superfluid*) – The QGW results for the superfluid components, the intraspecies current response, and the average kinetic energy across the second-order CFSF-to-SF transition of the $n_{0,d} = 1$ lobe for $U_{12}/U = 0.9$ are shown in Fig. 11.

In the SF region close to the CFSF phase, the results are equivalent to the ones found in the SF phase close to the MI lobe previously discussed. In the CFSF regime, we find that the superfluid drag reaches the negative saturation threshold $n_{12}/\sqrt{n_{s,1}\,n_{s,2}} = -100\%$, a result that strongly recalls the QMC calculations in Ref. [25], performed in $d = 2$ as well. Notably, we also observe that in the CFSF phase, even though the drag is saturated, the net superfluidity $n_{s,1} + n_{12} = 0$ vanishes. This mirrors the fact that counterflow superfluidity occurs when particles and holes of different species flow along counter-directed paths, such that equal and

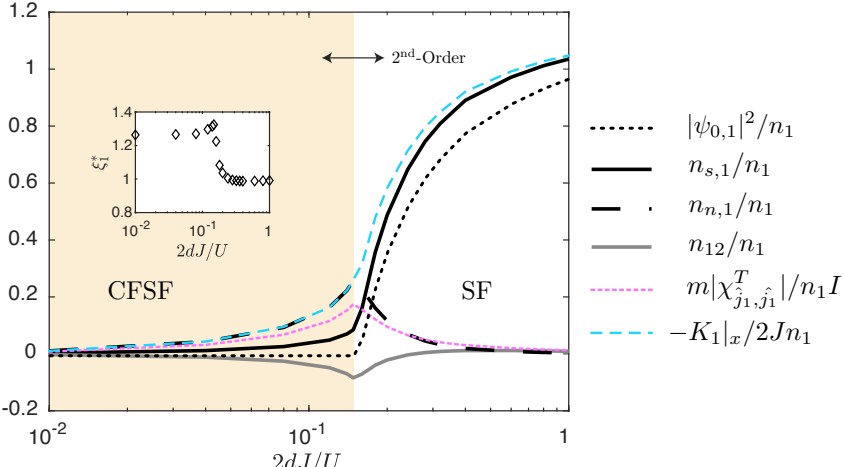

Figure 11: Transport quantities in two dimensions across the CFSF-to-SF second-order edge transition, traversed at fixed chemical potential $\mu/U = 0.5$ for $U_{12}/U = 0.9$, corresponding to the $n_d = 1$ CFSF lobe in Fig. 1(b). The one-component filling $n_i$ includes the corrections of the second-order quantum fluctuations. The tan-shaded area indicates the MI region. Thick black and grey lines indicate the QGW predictions for the superfluid density $n_{s,1}$ and the superfluid drag $n_{12}$ respectively, while the black dashed and dotted lines refer to the normal component $n_{n,1}$ and condensate fraction $|\psi_{0,1}|^2$. Pink dotted and blue dashed lines are the contributions to $n_{s,1}$ from the average kinetic energy $K_i|_x$ and the intraspecies current response function $\chi^T_{\hat{j}_i, \hat{j}_i}$, respectively. (Inset) Renormalization of the effective mass across the CFSF transition due to the collisionless drag.

opposite current densities for each component are established [6, 25]. Importantly, within the QGW formalism this perfect balance is due *solely* to quantum fluctuations, as the mean-field Gutzwiller theory trivially predicts $n_{s,1} = n_{12} = 0$. Qualitatively speaking, such finding agrees with the statement in Ref. [32] that superfluidity in the CFSF phase arises through second-order hopping processes not captured by mean-field theory; however, the *ad hoc* introduction of such processes performed in that work is not explicitly comprised in the O$(J^2)$ contribution of quantum fluctuations to Eqs. (57) and (58), which is thus a genuine result of the QGW quantum theory. The collisionless drag remains large and negative across the CFSF-to-SF transition, so that the renormalized effective mass becomes significantly less than its bare value, particularly at the critical point. We remark here that the onset of a negative superfluid drag indicates that a traveling particle transports, in addition to it's own bare mass, holes of the other species resulting in a reduced effective mass. As the deep SF regime is approached, the drag $n_{12}$ changes sign leading to an increased effective mass as discussed in Sec. 3.2.1.

As before, we comment on the relative weight of the various excitations vertices that contribute to the quantities shown in Fig. 11. In the CFSF regime, the average kinetic energy $K_i|_x$ is almost saturated by the gapped hole mode with the lowest energy. Correspondingly, both the intraspecies $\chi^T_{\hat{j}_i, \hat{j}_i}$ and interspecies $\chi^T_{\hat{j}_1, \hat{j}_2}$ current response functions are dominated by the all-to-all vertices of the lowest four particle-hole bands. It is curious to observe that the spectral weight of these particle-hole excitations is sufficient to obtain the $-100\%$ saturation of the superfluid drag, even when the antipair excitations are described by flat bands $\omega_{0,1}$, which do not contribute to the transport physics in our theory. As the transition point is crossed into the SF regime, $K_i|_x$ is saturated by the density Goldstone mode, while the response functions $\chi^T_{\hat{j}_i, \hat{j}_i}$

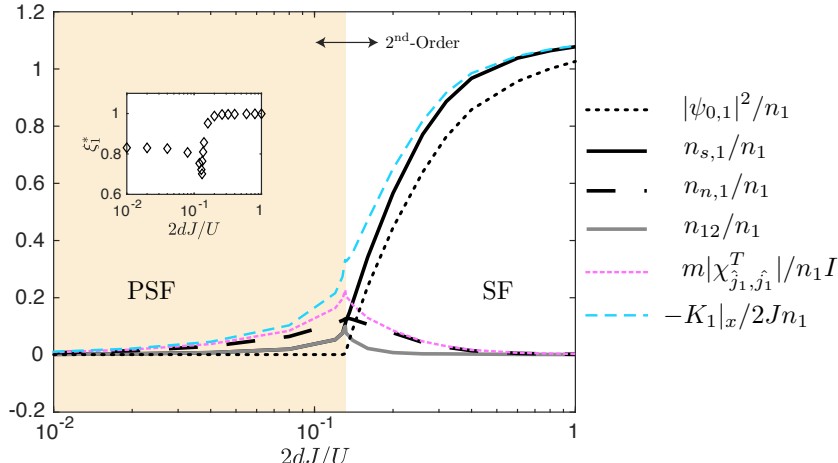

Figure 12: Transport quantities in two dimensions across the PSF-to-SF phase transition along the $(\mu/U)_c = -0.35$ critical line, which separates the vacuum region from the $n_d = 2$ MI lobe for $U_{12}/U = -0.7$, see Fig. 4(b). The one-component filling $n_i$ include the corrections of the second-order quantum fluctuations. The tan-shaded area indicates the MI region. Thick black and grey lines indicate the QGW predictions for the superfluid density $n_{s,1}$ and the superfluid drag $n_{12}$ respectively, while the black dashed and dotted lines refer to the normal component $n_{n,1}$ and condensate fraction $|\psi_{0,1}|^2$. Pink dotted and blue dashed lines are the contributions to $n_{s,1}$ from the average kinetic energy $K_i|_x$ and the intraspecies current response function $\chi^T_{\hat{j}_i, \hat{j}_i}$, respectively. (Inset) Renormalization of the effective mass across the PSF transition due to the collisionless drag.

and $\chi^T_{\hat{j}_1, \hat{j}_2}$ get a large contribution by the vertices between the density Goldstone mode and the first gapped modes of the SF phase.

### 3.2.3 Phase transitions for $U_{12} < 0$

In this last subsection, we study the superfluid components across the the second-order PSF-to-SF transition described in Sec. 2.2.2. The analysis of the MI-to-SF critical behavior given in the previous subsection is found to remain valid also in the case $U_{12} < 0$, with the only difference that the physical roles of the spin and density Goldstone modes are swapped. Therefore, in the following we proceed to investigate in detail the behavior of the superfluid components across the PSF-to-SF transition.

(*Pair Superfluid to Superfluid*) – The QGW predictions for the transport quantities across the second-order PSF-to-SF for $U_{12}/U = -0.7$ with $(\mu/U)_c = -0.35$ and $n_{0,d} \approx 1.47$ are presented in Fig. 12.

In the PSF regime, we find that the superfluid drag fulfills the positive saturation condition $n_{12}/\sqrt{n_{s,1} n_{s,2}} = +100\%$, which is again compatible with the QMC findings in Ref. [25]. Unlike the CFSF phase, however, here the saturation of the drag results in net non-zero superfluidity $n_{s,1} + n_{12} > 0$. Physically speaking, this can be interpreted from the point of view of pairs of particles of different species following co-directed paths and resulting in pair superfluidity [25]. Similarly to the QGW representation of the CFSF phase, this result can be directly ascribed to the quantum fluctuations captured by our theory. Indeed, the collisionless drag remains large and positive along the whole PSF line and through the transition into the SF phase. Consequently, the effective mass is strongly renormalized, so as to be significantly

larger than the bare mass, in particular at the transition point. We note that this increase of the effective mass is much larger than in the SF and MI regimes due to the tendency of a traveling particle to transport, in addition to its own mass, particles of the other species that consequently follow co-directed paths and increasing the dressing effect of the medium. We note that as the system approaches the deep SF regime, the drag remains positive but becomes increasingly small such that the effective mass always remains slightly larger than the bare mass as expected from Sec. 3.2.1.

We conclude by inspecting the spectral composition of the various quantities shown in Fig. 12. On the verge of the PSF side of the transition, the average kinetic energy $\langle \hat{K}_i|_x \rangle$ is strongly dominated by the lowest quasiparticle band (with hybrid spin and density character). The same observation partially applies to both the current response functions $\chi^T_{\hat{j}_i,\hat{j}_i}$ and $\chi^T_{\hat{j}_1,\hat{j}_2}$, in which the coupling between different quasiparticle bands plays a major role. As for the CFSF phase in Fig. 11, it is surprising to verify that these gapped quasiparticle modes are sufficient to reproduce the superfluid drag saturation of the PSF phase, such that they can be regarded as the only excitations responsible for the transport of particle pairs. Once the critical point is reached and the SF phase develops, $K_i|_x$ gets a major contribution from the spin Goldstone mode, while the current response functions are saturated by the coupling of the same mode with the Higgs-like branches that populate the high-energy part of the spectrum, in strict analogy with the case of repulsive interactions.

## 3.3 Coherence function

To obtain a better understanding of the role of quantum fluctuations across the phase diagram, we now turn our attention to the study of the equal-time correlation functions directly, which can also be probed experimentally using e.g. quantum gas microscopy [61, 62]. In particular, we anticipate that the separation between density and spin degrees of freedom in the CFSF and PSF phases is shown to be connected to Mott or superfluid-like behaviors of the correlation functions in the two excitation channels. In analyzing the single-particle coherence functions, we restrict ourselves to the investigation of intraspecies correlations, as the approximate description of the CFSF and PSF phases within the Gutzwiller framework [32] is expected to miss non-local coherence effects involving strongly-correlated pairs, in contrast with the results for the current response functions studied in Sec. 3.2.

The normalized single-particle coherence function for the $i^{\text{th}}$ species is defined as

$$g_i^{(1)}(\mathbf{r}) \equiv \frac{\langle \hat{a}_{i,\mathbf{r}}^\dagger \hat{a}_{i,\mathbf{0}} \rangle}{\langle \hat{a}_{i,\mathbf{0}}^\dagger \hat{a}_{i,\mathbf{0}} \rangle} \,. \tag{68}$$

The QGW quantization scheme maps the *microscopic* operator $\hat{a}_{i,\mathbf{r}}$ into the effective Bose field $\hat{\psi}_i(\mathbf{r})$, which carries both a *macroscopic* contribution due to condensation and the effect of short-range quantum correlations on the one-body coherence. In strict analogy with Bogoliubov's theory, the first-order expansion of $\hat{\psi}_i(\mathbf{r})$ in terms of quantum fluctuations reads

$$\hat{\psi}_i(\mathbf{r}) \approx \psi_{0,i} + \frac{1}{\sqrt{I}} \sum_\alpha \sum_{\mathbf{k}} \left[ U_{i,\alpha,\mathbf{k}} e^{i\,\mathbf{k}\cdot\mathbf{r}} \hat{b}_{\alpha,\mathbf{k}} + V_{i,\alpha,\mathbf{k}}^* e^{-i\,\mathbf{k}\cdot\mathbf{r}} \hat{b}_{\alpha,\mathbf{k}}^\dagger \right], \tag{69}$$

where the quasiparticle (quasihole) amplitudes $U_{i,\alpha,\mathbf{k}} \left( V_{i,\alpha,\mathbf{k}} \right)$ have been introduced in Eqs. (18) and saturate the Bogoliubov normalization condition

$$\sum_\alpha \left( \left| U_{i,\alpha,\mathbf{k}} \right|^2 - \left| V_{i,\alpha,\mathbf{k}} \right|^2 \right) = 1 \,, \tag{70}$$

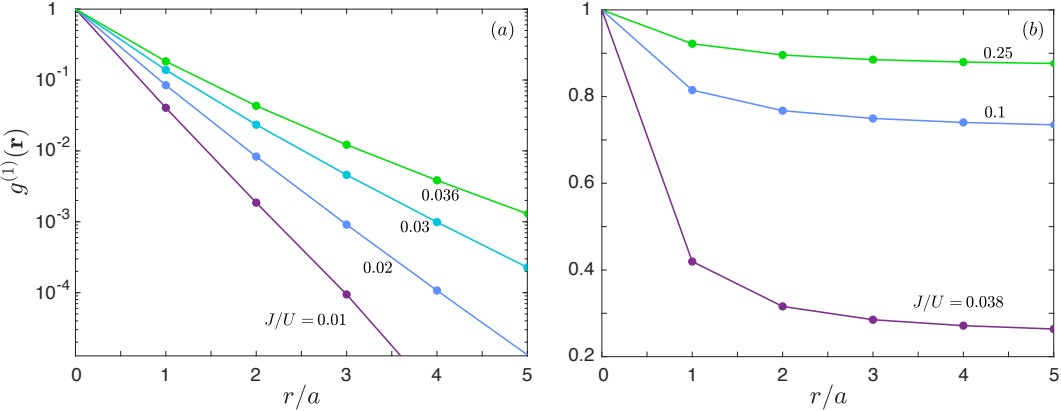

Figure 13: One-body coherence function $g_i^{(1)}(\mathbf{r})$ for $d = 2$ and $U_{12}/U = 0.9$ evaluated at fixed $\mu/U = 1.4$ across the (a) MI to (b) SF first-order transition.

in the whole phase diagram. Most importantly, the completeness relation (70) implies that the QGW Bose field (69) satisfies the usual Bose commutation relation $\left[\hat{\psi}_i(\mathbf{r}_1), \hat{\psi}_i^\dagger(\mathbf{r}_2)\right] = \delta_{\mathbf{r}_1,\mathbf{r}_2}$ identically, thus justifying the physical interpretation of $\hat{\psi}_i(\mathbf{r})$ (see App. A.2 for a detailed derivation). Applying our evaluation protocol to Eq. (68) and inserting the operator expansion (69), the single-particle coherence function can be recast into the form

$$g_i^{(1)}(\mathbf{r}) = \frac{\langle \hat{\psi}_i^\dagger(\mathbf{r}) \hat{\psi}_i(\mathbf{0}) \rangle}{\langle \hat{\psi}_i^\dagger(\mathbf{0}) \hat{\psi}_i(\mathbf{0}) \rangle} \approx \frac{\left|\psi_{0,i}\right|^2 + I^{-1} \sum_\alpha \sum_\mathbf{k} \left|V_{i,\alpha,\mathbf{k}}\right|^2 \cos(\mathbf{k} \cdot \mathbf{r})}{\left|\psi_{0,i}\right|^2 + I^{-1} \sum_\alpha \sum_\mathbf{k} \left|V_{i,\alpha,\mathbf{k}}\right|^2}, \tag{71}$$

which is a straightforward generalization of the one-component result in Ref. [36]. In the numerator of the right-hand side of Eq. (71), the first term reflects the long-range order of the one-body density matrix in the SF phase, while the second term reproduces the destructive interference of quantum fluctuations at finite distances, such that only the condensate fraction $\left|\psi_{0,i}\right|^2$ survives in the $\mathbf{r} \to \infty$ limit (see the relevant discussion in Sec. 2.2 of Ref. [53]).

We do not show here explicit results for the second-order MI-to-SF transition, because it presents the same features as in the one-component case discussed in [36]. Specifically, the QGW approach drastically improves mean-field theory by predicting the onset of off-site coherence in the strongly-interacting regime: in the MI phase, the one-body correlations are generally suppressed exponentially as $g^{(1)}(\mathbf{r}) \sim \exp(-r/\xi)$ with a finite coherence length $\xi$, whereas in the SF phase $g^{(1)}(\mathbf{r})$ always decays as a power law. More generally, in the deep SF limit $(2dJ/U \gg 1)$ the spectral sum in Eq. (71) is almost saturated by the density and spin Goldstone modes only and the behavior of a weakly-interacting gas is recovered [35, 36]. As a strongly-correlated SF develops, the contribution of other excitation modes to the quantum depletion becomes relevant and the Bogoliubov predictions are naturally amended by the QGW scenario.

In the two-component system, we find that the very same behavior carries over also to the first-order MI-to-SF transition, shown in Fig. 13, and to the second-order CFSF-to-SF transition and the PSF-to-SF critical point, illustrated in Fig. 14. In particular, in analogy with the physics of the MI-to-SF transition in the one-component system [36], the QGW theory is able to capture the different critical behaviors of the CFSF-to-SF transition depending on whether this is approached at integer or non-integer filling, namely across either the tip or the edge of the CFSF lobes. Upon reaching the SF phase through the edge of the CFSF lobe [Fig. 14(a), from purple to green lines], the correlation length $\xi$ grows monotonically but remains bounded. As soon as one enters the SF phase, the long-range behavior of $g_i^{(1)}(\mathbf{r})$ changes abruptly into a

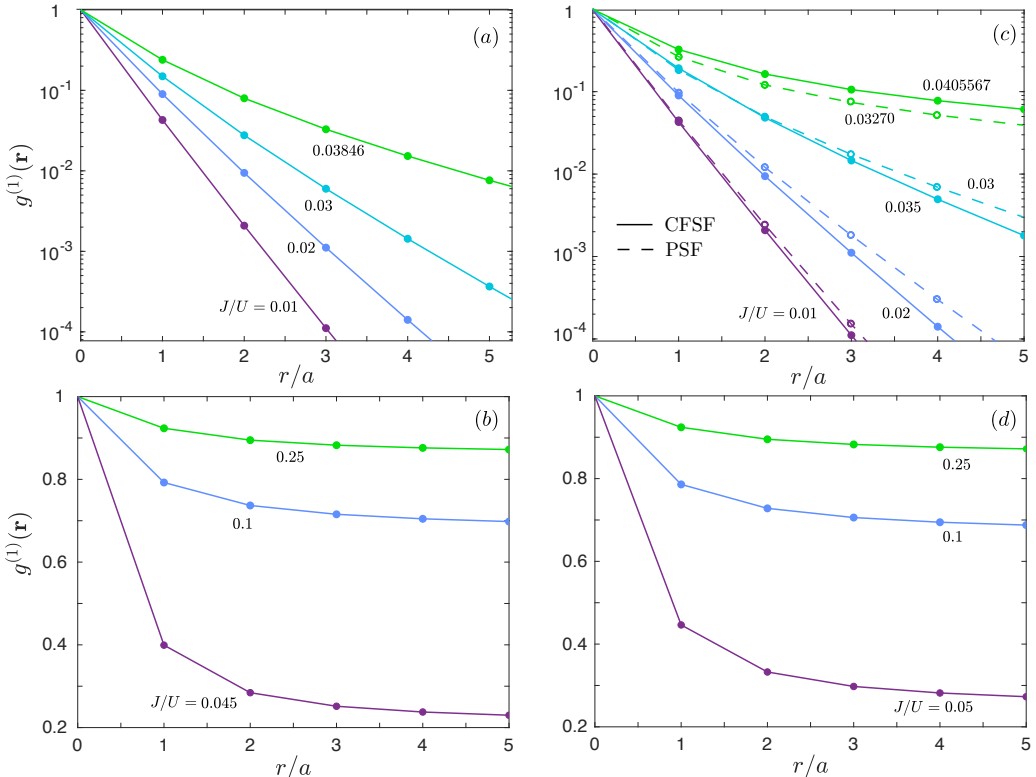

Figure 14: One-body coherence function $g^{(1)}(\mathbf{r})$ for $d = 2$ and $U_{12}/U = 0.9$ evaluated at fixed $\mu/U = 0.5$ (a-b) and $\mu/U \approx 0.39$ (c-d) across the edge and O(2) CFSF-to-SF transitions, respectively. In panel (c), the approaching of the PSF-to-SF transition from the PSF side is also shown for $(\mu/U)_c = -0.35$ and $U_{12}/U = -0.7$.

power-law scaling [Fig. 14(b)]. On the contrary, when approaching the SF phase at the tip of the CFSF region, the correlation length $\xi$ diverges [Fig. 14(c)] and a power-law dependence for $g_i^{(1)}(\mathbf{r})$ gradually sets in [Fig. 14(d)]. In panel (c), the evolution of $g_i^{(1)}(\mathbf{r})$ along the PSF critical line is also shown, displaying an analogous behavior.

As in the single particle case, this difference in the behavior of $g_i^{(1)}(\mathbf{r})$ is due to the different spectral properties of the collective excitations close to the edge/tip critical points of the CFSF lobe. At the edge transition, either a particle or a hole excitation out of the four dispersive branches becomes gapless. Since our description of the short-distance coherence in the CFSF phase relies on virtual particle-hole excitations via Eq. (71), the exponential decay of $g_i^{(1)}(\mathbf{r})$ is dominated by the gap of the particle (hole) excitation that remains finite at the transition. On the contrary, at the tip critical point, both the lowest-energy particle and hole modes become gapless (before turning into the density and spin Goldstone modes on the SF side of the transition), which explains the divergent coherence length [63].

## 3.4 Density and spin fluctuations

In this subsection, we address the local structure of equal-time two-body correlations for both the density and spin channels, focusing on their behavior across the quantum phase transitions of the system as determined by second-order quantum fluctuations. Crucially, the QGW method treats local and non-local observables separately. Whereas non-local two-body correlations $G_{d/s}^{(2)}(\mathbf{r} \neq \mathbf{0}) = \langle \hat{n}_{d/s}(\mathbf{r} \neq \mathbf{0}) \hat{n}_{d/s}(\mathbf{0}) \rangle$ are directly related to the Fourier transform of the

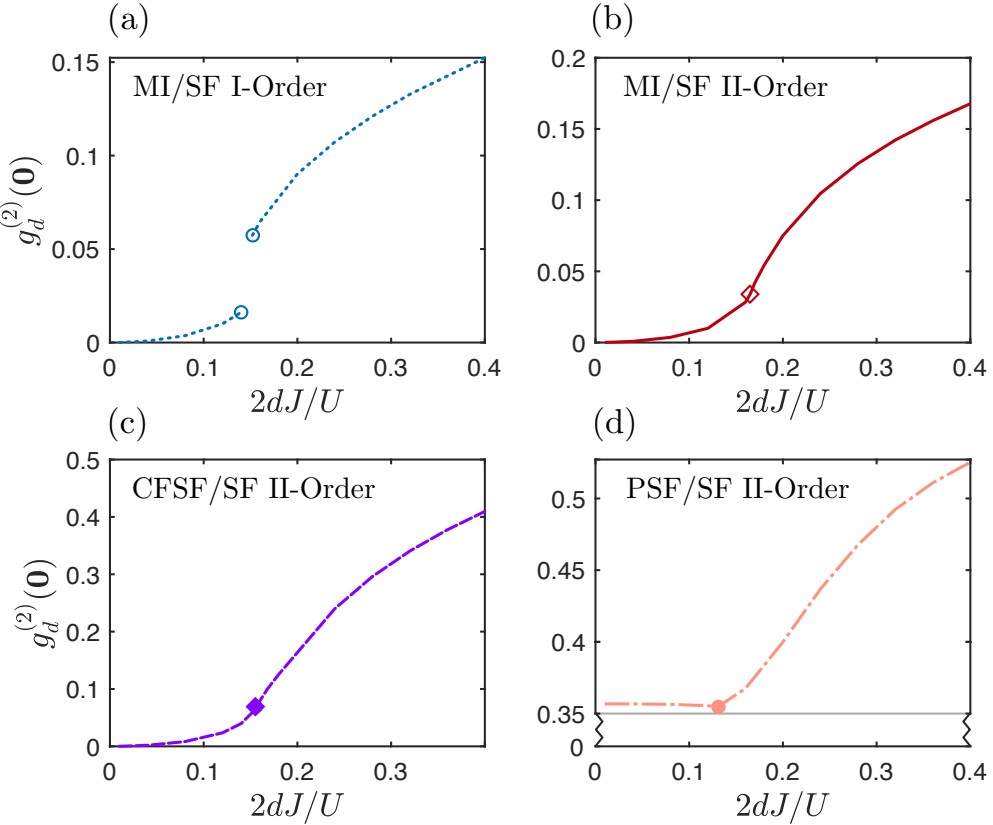

Figure 15: Local density-density correlation $g_d^{(2)}(\mathbf{r}=\mathbf{0})$ evaluated for $d=2$ as a function of $2dJ/U$ across (a) the first-order MI-to-SF edge transition (blue dotted lines) for $U_{12}/U=0.9$ and $\mu/U=1.4$ at fixed $n_d=2$, (b) the second-order MI-to-SF edge transition (red solid line) for $U_{12}/U=0.5$ and $\mu/U=1$ at fixed $n_d=2$, (c) the second-order CFSF-to-SF edge transition (purple dashed line) for $U_{12}/U=0.9$ and $\mu/U=0.5$ at fixed $n_d=1$, and (d) the second-order PSF-to-SF transition (pink dashed-dotted line) for $U_{12}/U=-0.7$ and $(\mu/U)_c=-0.35$ with $n_d=1.47$. The point located on each curve indicates the location of the respective phase transition.

static structure factors $S_{d/s}(\mathbf{q})$ (and share then the pathologies discussed in Sec. 3.1), on-site fluctuations can be always computed as expectation values of individual local operators, as shown explicitly in the following. Ultimately, the Gutzwiller ansatz (1) is a local approximation of the ground state, and therefore we expect predictions presented in this subsection for local two-body correlations to remain accurate even in the vicinity of the pair and anti-paired phases.

With these caveat in mind, we first consider the two-body correlation function for an individual species

$$G_i^{(2)}(\mathbf{r}=\mathbf{0}) = \langle \hat{n}_i^2(\mathbf{0}) \rangle \longrightarrow \langle \hat{\mathcal{D}}_i(\mathbf{0}) \rangle, \qquad (72)$$

where we have introduced the QGW square density operator

$$\hat{\mathcal{D}}_i(\mathbf{r}) = \sum_{n_1,n_2} \left( n_1^2 \delta_{i,1} + n_2^2 \delta_{i,2} \right) \hat{c}_{n_1,n_2}^\dagger(\mathbf{r}) \hat{c}_{n_1,n_2}(\mathbf{r}), \qquad (73)$$

which, importantly, is distinct from the square of the density operator $\hat{\mathcal{N}}_i(\mathbf{r})$ (see App. A.2). Expanding $\hat{\mathcal{D}}_i(\mathbf{0})$ up to second-order in the fluctuations and calculating its quantum average,

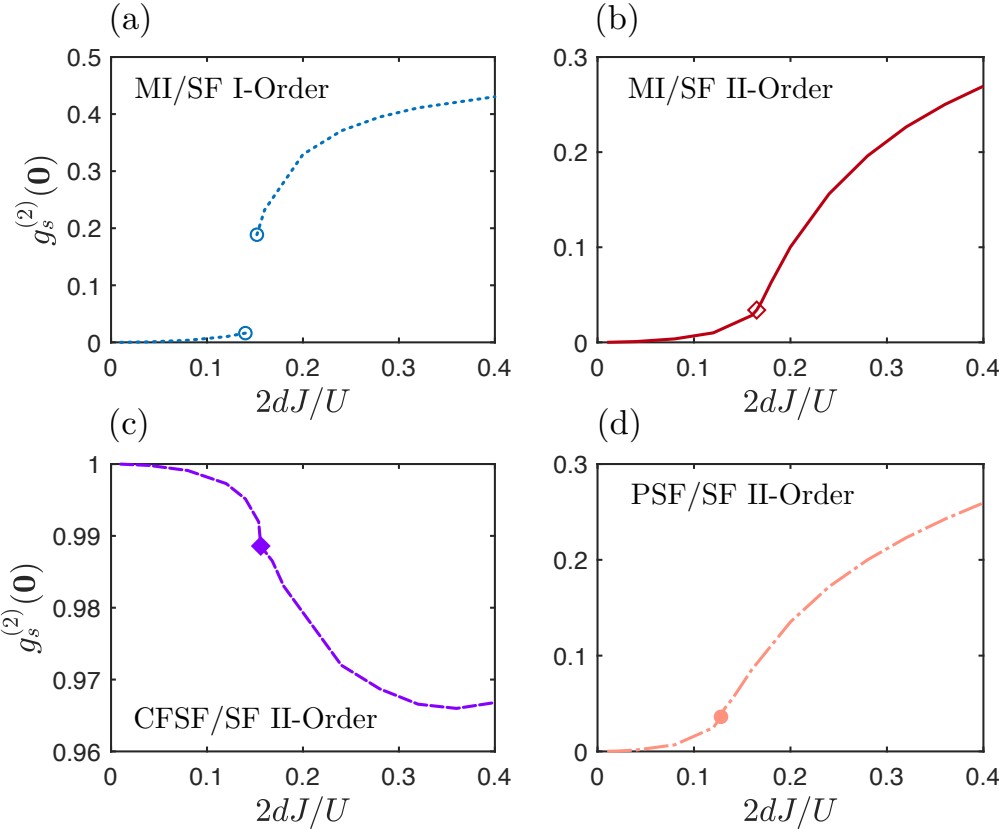

Figure 16: Local spin-spin correlation $g_s^{(2)}(\mathbf{r} = \mathbf{0})$ evaluated for $d = 2$ as a function of $2dJ/U$ across the same phase transitions as in Fig. 15.

we obtain

$$G_i^{(2)}(\mathbf{0}) = (1-F)\, d_{0,i} + \frac{1}{I} \sum_{\alpha} \sum_{\mathbf{k}} \sum_{n_1,n_2} \left( n_1^2\, \delta_{i,1} + n_2^2\, \delta_{i,2} \right) \left| v_{\alpha,\mathbf{k},n_1,n_2} \right|^2 , \qquad (74)$$

at zero temperature, which is a generalization of the result found in Ref. [36] for the one-component BH model. Here, $d_{0,i} = \sum_{n_1,n_2} \left( n_1^2\, \delta_{i,1} + n_2^2\, \delta_{i,2} \right) \left| c_{n_1,n_2}^0 \right|^2$ is the mean-field value of the square density, while

$$F = \langle \hat{A}^2(\mathbf{0}) \rangle = \frac{1}{I} \sum_{\alpha} \sum_{\mathbf{k}} \sum_{n_1,n_2} \left| v_{\alpha,\mathbf{k},n_1,n_2} \right|^2 , \qquad (75)$$

is the control parameter of the theory, see App. (A.3) for a detailed derivation. The quantization protocol corrects the mean-field value of local observables in a two-fold way. On the one hand, the second term of Eq. (74) makes a positive contribution due to quantum fluctuations; on the other hand, the quantity $F$, which measures the magnitude of quantum fluctuations, renormalizes the mean-field value $d_{0,i}$.

Next, we calculate the two-body correlation function between different species $G_{12}^{(2)}(\mathbf{r} = \mathbf{0}) = \langle \hat{n}_1(\mathbf{r} = \mathbf{0})\, \hat{n}_2(\mathbf{0}) \rangle$, which is given by

$$G_{12}^{(2)}(\mathbf{0}) = \langle \hat{\mathcal{D}}_{12}(\mathbf{0}) \rangle , \qquad (76)$$

where we have defined the composite density operator

$$\hat{\mathcal{D}}_{12}(\mathbf{r}) = \sum_{n_1,n_2} n_1\, n_2\, \hat{c}_{n_1,n_2}^\dagger(\mathbf{r})\, \hat{c}_{n_1,n_2}(\mathbf{r}) . \qquad (77)$$

We find the result

$$G_{12}^{(2)}(\mathbf{0}) = (1-F)\, d_{0,12} + \frac{1}{I} \sum_{\alpha} \sum_{\mathbf{k}} \sum_{n_1,n_2} n_1\, n_2 \left| v_{\alpha,\mathbf{k},n_1,n_2} \right|^2 , \tag{78}$$

where $d_{0,12} = \sum_{n_1,n_2} n_1\, n_2 \left| c_{n_1,n_2}^0 \right|^2$ is the mean-field prediction.

Having outlined the form of single-species and pair correlations, the on-site two-body correlation functions for the total density and spin channels can be obtained directly from

$$G_{d/s}^{(2)}(\mathbf{0}) = \langle [\hat{n}_1(\mathbf{0}) \pm \hat{n}_2(\mathbf{0})]^2 \rangle = G_1^{(2)}(\mathbf{0}) + G_2^{(2)}(\mathbf{0}) \pm 2\, G_{12}^{(2)}(\mathbf{0}) . \tag{79}$$

For convenience, we analyze the normalized density and spin variances $g_{d/s}^{(2)}(\mathbf{0})$. Estimating the variances amounts to shifting $G_{d/s}^{(2)}(\mathbf{0})$ by $n_{d/s}^2$; additionally, because $n_s = 0$ in our model, we normalize the correlation functions by the squared mean density $n_d^2$ to produce

$$g_{d/s}^{(2)}(\mathbf{0}) = \frac{G_{d/s}^{(2)}(\mathbf{0}) - n_{d/s}^2}{n_d^2} . \tag{80}$$

Our calculations for the density correlation $g_d^{(2)}(\mathbf{0})$ across the various phase transitions of the model are shown in Fig. 15. In the SF region, we observe the typical antibunching $g_d^{(2)}(\mathbf{0}) < 1$ of local fluctuations due to the onsite interaction $U$. Moving towards the MI phase [panels (a)-(b)], the qualitative features of density correlations are strongly reminiscent of the behavior of a single-component BH model [36], except for the the first-order MI-to-SF transition, at which the antibunching factor shows a discontinuity. In particular, in the MI region, where mean-field theory would predict $g_d^{(2)}(\mathbf{0}) = 0$, the QGW approach is able to account for the virtual excitation of doublon-hole pairs, which leads to the scaling $g_i^{(2)}(\mathbf{0}) \propto (J/U)^2$ at low $J$, in excellent agreement with perturbative calculations in the strongly-interacting limit [31,32]. Remarkably, we observe that the CFSF phase [panel (c)] shares the same properties of the MI state in the density channel. This can be understood as a consequence of the similarity between the spectral structure of the two phases. In the CFSF phase, density fluctuations build on the lowest-lying, gapped particle-hole excitations, which have a strong density character exactly as their counterparts in the MI phase. By contrast, $g_d^{(2)}(\mathbf{0})$ exhibits a quite different behavior across the PSF-to-SF transition. Instead of being suppressed, sizeable density fluctuations survive in the whole PSF region and saturate to a finite value at low $J/U$. This result clearly agrees with the physical picture of the PSF phase [panel (d)], where the formation of local pairs is explicitly favoured, differently from the CFSF state, in which particles belonging to different species repel each other. However, we believe that the independence of $g_d^{(2)}(\mathbf{0})$ on $J/U$ is a by-product of the inability of our theory to capture density excitations in the PSF phase.

A complementary view on two-body correlations is provided by the spin fluctuations $g_s^{(2)}(\mathbf{0})$, which are reported in Fig. 16 along the same phase transition paths of Fig. 15. Across the MI-to-SF transition [panels (a)-(b)], spin correlations mimic the behavior of density fluctuations, meaning that the Mott interactions freeze equally both density and spin degrees of freedom. This is not the case for the paired phases, where the density and spin channels decouple. On the one hand, despite its Mott-like character in the density channel, the CFSF phase [panel (c)] is characterized by a large $g_s^{(2)}(\mathbf{0})$, indicating that counterflow superfluidity is linked to the creation of local magnetic moments with large spin fluctuations, such as in a spinful Mott state [64,65]. Notice that, however, $g_s^{(2)}(\mathbf{0}) < 1$ for finite values of $J/U$, signalling that the local moments possess a finite stiffness due to the particle-hole excitations

of the CFSF phase. On the other hand, we find that $g_s^{(2)}(\mathbf{0})$ is strongly suppressed in the PSF phase [panel (d)], where the spin degrees of freedom interact repulsively, in analogy again with the physics of the MI-to-SF transition [5].

# 4 Conclusions and Outlook

In this work, we have studied the properties of quantum fluctuations in the two-component BH model at zero temperature, for both repulsive and attractive interspecies interactions. Expanding on the mean-field ground-state phase diagram first reported in Ref. [32], we have analyzed the band structure of the model in the whole phase diagram with particular attention paid to its quantum phase transitions and the limitations inherent in the local nature of the Gutzwiller ansatz for the many-body wave function. Through a canonical quantization of the variational parameters of the ansatz, we have generalized to bosonic mixtures the single-component QGW theory of Ref. [36], which permits a study of the quantum corrections to obsevables in a systematic, order-by-order fashion in the spirit of the Bogoliubov theory of dilute weakly-interacting gases. Importantly, we have shown how the QGW method provides a comprehensive description of both local and non-local correlations across the entire phase diagram of the model and, in particular, across its quantum critical regimes.

We have first illustrated that the formation of pair correlations can be directly connected to the analytical behavior of the compressibility and spin susceptibility, which reflect the strength of critical fluctuations upon reaching the PSF and CFSF transitions, respectively. Notably, by the application of spectral sum rules [34] together with the fluctuation-dissipation theorem, we are able to relate the physical picture of the response functions to the sound velocities of the Goldstone modes and the static structure factors of the SF state. These results indicate that experimental probes of strongly-correlated paired phases can yield direct information on the spectral properties of the system in quantum critical regimes.

Within the QGW approach, we have also studied the properties of superfluid transport, finding an interspecies collisionless drag whose origin is due solely to quantum fluctuations. In this respect, we have compared quantitatively to QMC predictions [25] within the SF phase over a range of interspecies couplings, and study the matrix of superfluid components across the various phase transitions. In particular, we have found that the drag is saturated in the vicinity of the PSF and CFSF phases, where strong pair correlations prevail over single-particle coherence. Moreover, we have offered a clear interpretation of the superfluid components in terms of multi-mode scattering processes involving the collective excitations of the system, including not only the density and spin Goldstone modes, but also the Higgs modes and gapped quasiparticle excitations appearing at strong interactions.

Finally, we have shown that the QGW theory gives an accurate account of the role of quantum fluctuations in equal-time few-body correlations in the whole phase diagram of the system. In particular, we have demonstrated how the critical behaviors of the one-body coherence function are analogous to those found for MI-SF transition in the single-component BH model in Ref. [36]. Remarkably, at the two-body level, we have also found that the CFSF/PSF phase transitions strongly mirror the MI transition physics in the density/spin channels, respectively. Throughout our analysis, we have highlighted how quantum correlations closely link with the character of collective modes in distinct interaction regimes.

The QGW approach developed in this work proves to be a flexible, semi-analytical method to study the rich phase diagram of bosonic mixtures in an optical lattice, including the effects of quantum fluctuations while remaining computationally inexpensive. Although only the zero-temperature case is considered in this work, we also note that finite temperature effects can be incorporated in a straightforward manner. In particular, this method is expected to

be capable of describing also non-linear interactions between quasiparticles, for instance in Landau/Beliaev-type decay processes [66,67].

The generalization of the present theory to multi-component Bose-Fermi and Bose-Bose mixtures and to different trapping geometries, where novel types of superfluid drag are predicted [28,68], poses intriguing problems for future research. Additionally, an improvement in the description of non-local fluctuations, crucial for instance to introduce hopping-induced correlations into the paired phases, appears possible within a cluster extension of the Gutzwiller theory (c.f. Refs. [69–77]). Such an extension is required, for example, to describe long-range interactions or magnetic ordering in supersolid phases where translational symmetry is spontaneously broken (c.f. Refs. [5,78,79]). Relaxing the $\mathbb{Z}_2$-symmetry constraint also opens exciting questions on the imbalanced two-component BH model, whose fermionic counterpart is currently the subject of intense interest in the ultracold community (c.f. Refs. [80,81]).

From a more formal perspective, the systematic way in which quantum fluctuations can be incorporated in the QGW theory raises interesting curiosities about its diagrammatic representation in quantum field theory. Conversely, such a connection might enable the translation of the comparatively sizeable literature of diagrammatic techniques into the language of QGW theory to diagnose possible issues in the method or introduce concepts already well-known in that context (c.f. [82,83]).

# Acknowledgements

The authors thank K. Sellin and E. Babaev for providing their quantum Monte Carlo data from Ref. [25] used in Fig. 9. We acknowledge also D. Contessi for fruitful discussions.

**Funding information** This project has received financial support from Provincia Autonoma di Trento and the Italian MIUR through the PRIN2017 project CEnTraL (Protocol Number 20172H2SC4).

# A    Quantum corrections within the QGW approach

In this appendix, we provide additional technical details on the structure of quantum fluctuations within the QGW theory and outline the explicit derivation of the quantum correlations incorporated by the quantization procedure.

## A.1    Evaluation protocol for the observables

Here, we provide a brief recap of the QGW quantization protocol of Ref. [36] for evaluating an expectation value of a given observable $\hat{O}\left[\hat{a}_{i,\mathbf{r}}, \hat{a}_{i,\mathbf{r}}^{\dagger}\right]$:

1. Determine the average value of the observable $\mathcal{O}[c,c^*] = \left\langle \Psi_G \middle| \hat{O} \middle| \Psi_G \right\rangle$ in terms of the $\mathbb{C}$-valued Gutzwiller variables.

2. Define the operator $\hat{\mathcal{O}}\left[\hat{c}, \hat{c}^{\dagger}\right]$ by promoting the Gutzwiller parameters in $\mathcal{O}[c,c^*]$ to the corresponding operators without modifying their ordering.

3. Expand the operator $\hat{\mathcal{O}}$ order by order in the fluctuations $\delta\hat{c}_{n_1,n_2}$ and $\delta\hat{c}_{n_1,n_2}^{\dagger}$, including the contribution due to the normalization operator $\hat{A}$ properly. Using Eq. (41), this step provides a convenient decoding of the original microscopic operator $\hat{O}$ in terms of the quantized modes of the theory.

4. Taking advantage of the quadratic character of the QGW Hamiltonian (42), invoke Wick's theorem to compute the expectation value of products of the quasiparticle operators $\hat{b}_{\alpha,\mathbf{k}}$ on Gaussian states – such as ground or thermal states obtained from $H_{QGW}^{(2)}$.

In the following, we apply the above quantization procedure to the single and two-particle observables addressed in this work and clarify the order-by-order derivation of the quantum corrections accessed by our quantum theory.

## A.2 First-order fluctuations

The spectral features of the first-order fluctuations can be estimated either by using the linearized 2GE (8) as done in Ref. [32] or alternatively employing the QGW protocol, which has the formal advantage of associating the creation/annihilation of a well-defined collective mode with a given spectral amplitude, as outlined in this subsection.

*(Density)* – Within the QGW formalism, the species-resolved density $\hat{n}_i(\mathbf{r})$ is mapped into the operator

$$\hat{\mathcal{N}}_i(\mathbf{r}) = \sum_{n_1,n_2} \left( n_1\,\delta_{i,1} + n_2\,\delta_{i,2} \right) \hat{c}_{n_1,n_2}^\dagger(\mathbf{r})\,\hat{c}_{n_1,n_2}(\mathbf{r}). \tag{A.1}$$

Expanding the $\hat{c}$'s to lowest order in the fluctuations, one finds

$$\hat{\mathcal{N}}_i(\mathbf{r}) \approx n_{0,i} + \delta_1 \hat{n}_i(\mathbf{r}) = n_{0,i} + \sum_{n_1,n_2} \left( n_1\,\delta_{i,1} + n_2\,\delta_{i,2} \right) c_{n_1,n_2}^0 \left[ \delta \hat{c}_{n_1,n_2}(\mathbf{r}) + \delta \hat{c}_{n_1,n_2}^\dagger(\mathbf{r}) \right], \tag{A.2}$$

where $n_{0,i} = \sum_{n_1,n_2} \left( n_1\,\delta_{i,1} + n_2\,\delta_{i,2} \right) \left| c_{n_1,n_2}^0 \right|^2$ is the mean-field density of the $i^{\text{th}}$ species. Using Eq. (41), the first-order operator $\delta_1 \hat{n}_i(\mathbf{r})$ can be expressed straightforwardly in terms of the quasiparticle operators as

$$\delta_1 \hat{n}_i(\mathbf{r}) = \frac{1}{\sqrt{I}} \sum_\alpha \sum_\mathbf{k} N_{i,\alpha,\mathbf{k}} \left( e^{i\,\mathbf{k}\cdot\mathbf{r}}\,\hat{b}_{\alpha,\mathbf{k}} + e^{-i\,\mathbf{k}\cdot\mathbf{r}}\,\hat{b}_{\alpha,\mathbf{k}}^\dagger \right), \tag{A.3}$$

where the spectral weight of single-mode density excitations $N_{i,\alpha,\mathbf{k}}$ is given by Eq. (20).

*(Square density)* – According to the QGW mapping, the upgrade of the square density operator reads $\hat{d}_i(\mathbf{r}) = \hat{n}_i^2(\mathbf{r}) \to \hat{\mathcal{D}}_i(\mathbf{r})$ as given by Eq. (73). Specifically, up to the lowest order in the fluctuations, one obtains

$$\hat{\mathcal{D}}_i(\mathbf{r}) = \sum_{n_1,n_2} \left( n_1^2\,\delta_{i,1} + n_2^2\,\delta_{i,2} \right) \hat{c}_{n_1,n_2}^\dagger(\mathbf{r})\,\hat{c}_{n_1,n_2}(\mathbf{r}) \approx d_{0,i} + \delta_1 \hat{D}_i(\mathbf{r}), \tag{A.4}$$

where $d_{0,i} = \sum_{n_1,n_2} \left( n_1^2\,\delta_{i,1} + n_2^2\,\delta_{i,2} \right) \left| c_{n_1,n_2}^0 \right|^2$ is the mean-field square density of the $i^{\text{th}}$ species. The expansion of the first-order operator $\delta_1 \hat{D}_i(\mathbf{r})$ in terms of the quasiparticle excitations is

$$\begin{aligned}
\delta_1 \hat{D}_i(\mathbf{r}) &= \sum_{n_1,n_2} \left( n_1^2\,\delta_{i,1} + n_2^2\,\delta_{i,2} \right) c_{n_1,n_2}^0 \left[ \delta \hat{c}_{n_1,n_2}(\mathbf{r}) + \delta \hat{c}_{n_1,n_2}^\dagger(\mathbf{r}) \right], \\
&= \frac{1}{\sqrt{I}} \sum_\alpha \sum_\mathbf{k} D_{i,\alpha,\mathbf{k}} \left( e^{i\,\mathbf{k}\cdot\mathbf{r}}\,\hat{b}_{\alpha,\mathbf{k}} + e^{-i\,\mathbf{k}\cdot\mathbf{r}}\,\hat{b}_{\alpha,\mathbf{k}}^\dagger \right),
\end{aligned} \tag{A.5}$$

where we have defined the amplitude

$$D_{i,\alpha,\mathbf{k}} = \sum_{n_1,n_2} n_i^2\, c_{n_1,n_2}^0 \left( u_{\alpha,\mathbf{k},n_1,n_2} + v_{\alpha,\mathbf{k},n_1,n_2} \right). \tag{A.6}$$

*(One-body boson field)* – The quantization protocol of the QGW method upgrades the order parameter of the one-body condensate to an effective boson field as $\psi_i(\mathbf{r}) \rightarrow \hat{\psi}_i(\mathbf{r})$. It is important to observe that $\hat{\psi}_i(\mathbf{r})$ is a macroscopic object (related to the coherence properties of the system) and is *distinct* from the microscopic field operators $\hat{a}_{i,\mathbf{r}}$ defined within second quantization at the beginning of Sec. 2.1. To lowest order in the fluctuations, we find

$$\hat{\psi}_1(\mathbf{r}) = \sum_{n_1,n_2} \sqrt{n_1}\, \hat{c}^\dagger_{n_1-1,n_2}(\mathbf{r})\, \hat{c}_{n_1,n_2}(\mathbf{r}) \approx \psi_{0,1} + \delta_1\hat{\psi}_1(\mathbf{r}), \qquad (A.7)$$

$$\hat{\psi}_2(\mathbf{r}) = \sum_{n_1,n_2} \sqrt{n_2}\, \hat{c}^\dagger_{n_1,n_2-1}(\mathbf{r})\, \hat{c}_{n_1,n_2}(\mathbf{r}) \approx \psi_{0,2} + \delta_1\hat{\psi}_2(\mathbf{r}), \qquad (A.8)$$

where $\psi_{0,i}$ is the one-body order parameter of the $i^{\text{th}}$ species at the mean-field level – see Eq. (4) – and

$$\delta_1\hat{\psi}_i(\mathbf{r}) = \frac{1}{\sqrt{I}} \sum_\alpha \sum_{\mathbf{k}} \left[ U_{i,\alpha,\mathbf{k}}\, e^{i\,\mathbf{k}\cdot\mathbf{r}}\, \hat{b}_{\alpha,\mathbf{k}} + V^*_{i,\alpha,\mathbf{k}}\, e^{-i\,\mathbf{k}\cdot\mathbf{r}}\, \hat{b}^\dagger_{\alpha,\mathbf{k}} \right]. \qquad (A.9)$$

The previous result clearly shows that the spectral amplitudes $U_{i,\alpha,\mathbf{k}}\ \left(V_{i,\alpha,\mathbf{k}}\right)$, introduced in Eqs. (18), quantify the particle (hole) character of the excitation $(\alpha,\mathbf{k})$ for the $i^{\text{th}}$ component of the system. When dealing with the calculation of expectation values in momentum space, it is often convenient to replace the microscopic bosonic fields by the corresponding expression in terms of quasiparticle operators as

$$\hat{a}_{i,\mathbf{k}} \longrightarrow \sqrt{I}\, \psi_{0,i}\, \delta_{\mathbf{k},\mathbf{0}} + \sum_\alpha \left( U_{i,\alpha,\mathbf{k}}\, \hat{b}_{\alpha,\mathbf{k}} + V^*_{i,\alpha,\mathbf{k}}\, \hat{b}^\dagger_{\alpha,-\mathbf{k}} \right). \qquad (A.10)$$

The structure of the particle (hole) amplitudes $U_{i,\alpha,\mathbf{k}}\ \left(V_{i,\alpha,\mathbf{k}}\right)$ is strictly related to the bosonic statistics of the collective excitations, since

$$\left[ \delta_1\hat{\psi}_i(\mathbf{r}), \delta_1\hat{\psi}^\dagger_i(\mathbf{s}) \right] = \frac{1}{I} \sum_{\mathbf{k}} e^{i\,\mathbf{k}\cdot(\mathbf{r}-\mathbf{s})} \sum_\alpha \left( \left|U_{i,\alpha,\mathbf{k}}\right|^2 - \left|V_{i,\alpha,\mathbf{k}}\right|^2 \right) = \delta_{\mathbf{r},\mathbf{s}}, \qquad (A.11)$$

where last equality is verified provided that $\sum_\alpha \left( \left|U_{i,\alpha,\mathbf{k}}\right|^2 - \left|V_{i,\alpha,\mathbf{k}}\right|^2 \right) = 1$. In practice, we find that the previous identity is numerically satisfied if a sufficiently large number of excitation branches are included into the $\alpha$-summation.

*(Pair/antipair field)* – The quantized pair and antipair fields read

$$\hat{\psi}_{\text{P}}(\mathbf{r}) = \sum_{n_1,n_2} \sqrt{n_1\,n_2}\, \hat{c}^\dagger_{n_1-1,n_2-1}(\mathbf{r})\, \hat{c}_{n_1,n_2}(\mathbf{r}) - \hat{\psi}_1(\mathbf{r})\, \hat{\psi}_2(\mathbf{r}), \qquad (A.12)$$

$$\hat{\psi}_{\text{C}}(\mathbf{r}) = \sum_{n_1,n_2} \sqrt{n_1(n_2+1)}\, \hat{c}^\dagger_{n_1-1,n_2+1}(\mathbf{r})\, \hat{c}_{n_1,n_2}(\mathbf{r}) - \hat{\psi}_1(\mathbf{r})\, \hat{\psi}^\dagger_2(\mathbf{r}). \qquad (A.13)$$

Including first-order fluctuations only, the pairing operators can be expanded as

$$\hat{\psi}_{\text{P}}(\mathbf{r}) \approx \psi_{0,\text{P}} + \delta_1\hat{\psi}_{\text{P}}(\mathbf{r}), \qquad (A.14)$$

$$\hat{\psi}_{\text{C}}(\mathbf{r}) \approx \psi_{0,\text{C}} + \delta_1\hat{\psi}_{\text{C}}(\mathbf{r}), \qquad (A.15)$$

where the mean-field quantities $\psi_{0,\text{P}}$ and $\psi_{0,\text{C}}$ are respectively the pair and antipair order parameters given by Eqs. (5) and

$$\begin{aligned} \delta_1\hat{\psi}_{\text{P}}(\mathbf{r}) = \sum_{n_1,n_2} \sqrt{n_1\,n_2} \Big[ c^0_{n_1,n_2}\, \delta\hat{c}^\dagger_{n_1-1,n_2-1}(\mathbf{r}) + c^0_{n_1-1,n_2-1}\, \delta\hat{c}_{n_1,n_2}(\mathbf{r}) \Big] \\ - \psi_{0,2}\, \delta_1\hat{\psi}_1(\mathbf{r}) - \psi_{0,1}\, \delta_1\hat{\psi}_2(\mathbf{r}), \end{aligned} \qquad (A.16)$$

$$\delta_1\hat{\psi}_C(\mathbf{r}) = \sum_{n_1,n_2} \sqrt{n_1(n_2+1)}\Big[ c^0_{n_1,n_2}\, \delta\hat{c}^\dagger_{n_1-1,n_2+1}(\mathbf{r}) + c^0_{n_1-1,n_2+1}\, \delta\hat{c}_{n_1,n_2}(\mathbf{r}) \Big]$$
$$- \psi_{0,2}\, \delta_1\hat{\psi}_1(\mathbf{r}) - \psi_{0,1}\, \delta_1\hat{\psi}_2^\dagger(\mathbf{r})\,. \tag{A.17}$$

Recasting the $\delta\hat{c}$'s in the quasiparticle basis, we can rewrite the first-order expansions (A.16)-(A.17) in a compact, suggestive form

$$\delta_1\hat{\psi}_{P/C}(\mathbf{r}) = \frac{1}{\sqrt{I}} \sum_\alpha \sum_{\mathbf{k}} \Big[ U_{P/C,\alpha,\mathbf{k}}\, e^{i\mathbf{k}\cdot\mathbf{r}}\, \hat{b}_{\alpha,\mathbf{k}} + V^*_{P/C,\alpha,\mathbf{k}}\, e^{-i\mathbf{k}\cdot\mathbf{r}}\, \hat{b}^\dagger_{\alpha,\mathbf{k}} \Big]$$
$$- \psi_{0,2}\, \delta_1\hat{\psi}_1(\mathbf{r}) - \psi_{0,1}\, \delta_1\hat{\psi}_2^\dagger(\mathbf{r})\,, \tag{A.18}$$

where the particle-hole amplitudes $U_{P/C,\alpha,\mathbf{k}}$ and $V_{P/C,\alpha,\mathbf{k}}$ are explicitly given by

$$U_{P,\alpha,\mathbf{k}} = \sum_{n_1,n_2} \sqrt{n_1 n_2}\Big( c^0_{n_1-1,n_2-1}\, u_{\alpha,\mathbf{k},n_1,n_2} + c^0_{n_1,n_2}\, v_{\alpha,\mathbf{k},n_1-1,n_2-1} \Big)\,, \tag{A.19}$$

$$U_{C,\alpha,\mathbf{k}} = \sum_{n_1,n_2} \sqrt{n_1(n_2+1)}\Big( c^0_{n_1-1,n_2+1}u_{\alpha,\mathbf{k},n_1,n_2} + c^0_{n_1,n_2}\, v_{\alpha,\mathbf{k},n_1-1,n_2+1} \Big)\,, \tag{A.20}$$

$$V_{P,\alpha,\mathbf{k}} = \sum_{n_1,n_2} \sqrt{n_1 n_2}\Big( c^0_{n_1,n_2}\, u_{\alpha,\mathbf{k},n_1-1,n_2-1} + c^0_{n_1-1,n_2-2}\, v_{\alpha,\mathbf{k},n_1,n_2} \Big)\,, \tag{A.21}$$

$$V_{C,\alpha,\mathbf{k}} = \sum_{n_1,n_2} \sqrt{n_1(n_2+1)}\Big( c^0_{n_1,n_2}\, u_{\alpha,\mathbf{k},n_1-1,n_2+1} + c^0_{n_1-1,n_2+2}\, v_{\alpha,\mathbf{k},n_1,n_2} \Big)\,, \tag{A.22}$$

in complete analogy with Eqs. (18).

Recalling the discussion on the completeness relation concerning the one-body Bose fields (A.11), a natural question that arises from the decompositions in Eqs. (A.16)-(A.17) is whether they reproduce the canonical commutators

$$\Big[\hat{a}_{1,\mathbf{r}}\hat{a}_{2,\mathbf{r}}, \hat{a}^\dagger_{1,\mathbf{s}}\hat{a}^\dagger_{2,\mathbf{s}}\Big] = \delta_{\mathbf{r},\mathbf{s}}[1+\hat{n}_d]\,, \tag{A.23}$$

$$\Big[\hat{a}_{1,\mathbf{r}}\hat{a}^\dagger_{2,\mathbf{r}}, \hat{a}^\dagger_{1,\mathbf{s}}\hat{a}_{2,\mathbf{s}}\Big] = -\delta_{\mathbf{r},\mathbf{s}}\,\hat{n}_s\,, \tag{A.24}$$

where $\hat{n}_d$ and $\hat{n}_s$ were defined in Sec. 2.1. Inserting the QGW expression of the pairing fields (A.18), the first-order approximation of the commutation rules (A.23)-(A.24) reads

$$\Big[\delta_1\hat{\psi}_{P/C}(\mathbf{r}), \delta_1\hat{\psi}^\dagger_{P/C}(\mathbf{s})\Big] = \frac{1}{I}\sum_{\mathbf{k}} e^{i\mathbf{k}\cdot(\mathbf{r}-\mathbf{s})}\sum_\alpha \Big( \big|U_{\alpha,P/C,\mathbf{k}}\big|^2 - \big|V_{\alpha,P/C,\mathbf{k}}\big|^2 \Big)\,. \tag{A.25}$$

Notably, we find numerically that, due to the only approximate description of the excitation spectrum of the CFSF and PSF phases, in particular the flat bands describing pairing and antipairing excitations in Figs. 3 and 6 respectively, the above canonical relations are not satisfied in general. Likewise, we expect a similar result when the higher-order contributions are included. For these reasons, only intraspecies coherence functions have been considered in Sec. 3.3.

## A.3 Second-order fluctuations

Having illustrated how first-order quantized excitations build on top of the mean-field observables, we now estimate the corrections to observables due to second-order quantum fluctuations.

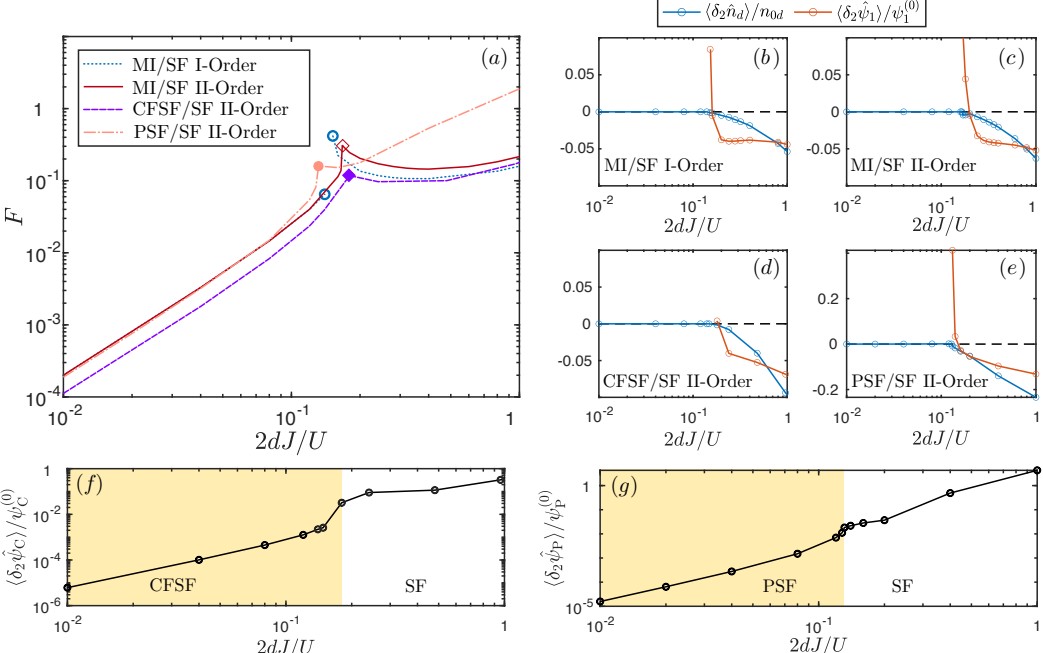

Figure 17: (a) Control parameter $F$ as defined in Eq. (A.26) plotted in two dimensions as a function of $2dJ/U$ for fixed $n_{0,d}$ corresponding to the transitions shown in the right panel. The point located on each curve indicates the location of the respective phase transition. (b)-(d) Quantum corrections to the order parameters and total density for the transitions considered in the (a). (b) First-order MI-to-SF transition for $U_{12}/U = 0.9$ and $n_d = 2$. (c) Second-order MI-to-SF transition for $U_{12}/U = 0.5$ and $n_d = 2$. (d) CFSF-to-SF transition for $U_{12}/U = 0.9$ and $n_d = 1$. (e) PSF-to-SF transition for $U_{12}/U = -0.7$ and $n_d \approx 1.47$. (f) Quantum corrections to the CFSF order parameter for fixed total density $n_d = 1$ at $U_{12}/U = 0.9$ and along the $\mu/U = 0.5$ line within the CFSF phase. (g) Quantum corrections to the PSF order parameter for fixed total density $n_d \approx 1.47$ at $U_{12}/U = -0.7$ and along the $(\mu/U)_c = -0.35$ critical line within the PSF phase.

*(Control parameter of the theory)* – Following Ref. [36], we start our analysis by inspecting the magnitude of quantum fluctuations around the Gutzwiller mean-field state, identified by the control parameter

$$F = 1 - \langle \hat{A}^2(\mathbf{r}) \rangle = \sum_{n_1,n_2} \langle \delta \hat{c}^{\dagger}_{n_1,n_2}(\mathbf{r}) \, \delta \hat{c}_{n_1,n_2}(\mathbf{r}) \rangle = \frac{1}{I} \sum_{\alpha} \sum_{\mathbf{k}} \sum_{n_1,n_2} \left| v_{\alpha,\mathbf{k},n_1,n_2} \right|^2 . \tag{A.26}$$

Within the QGW theory of the one-component Bose Hubbard model, it was found in Ref. [36] that, for $d = 3$ and fixed $\langle \hat{n}_d \rangle$, $F$ is peaked at the MI-to-SF transition and approaches zero in the limiting regimes away from the transition. For the present two-component case, we show results for $d = 2$ in Fig. 17(a) for fixed $n_d$ across the first and second-order MI-to-SF transitions [blue and orange lines, respectively], the PSF-to-SF transition [purple line], and the CFSF-to-SF transition [yellow line].

In fixing the total filling and approaching the deep SF regime, one travels along the line $\mu \sim -2dJ$ for increasing $J$, such that the limit $n_d U \to 0$ is reached. In three dimensions, this limit coincides with the dilute, weakly-interacting regime, as the two-body scattering amplitude is density-independent. On the contrary, from renormalization group arguments it is known that in two dimensions the dilute limit is more subtle, since the two-body scattering amplitude in the weakly-interacting regime depends logarithmically on the diluteness parameter

$n_d\, a_{2D}^2$, where $a_{2D}$ is the two-dimensional scattering length [34, 84–86]. The bare interaction strength $U$ used throughout our analysis cannot capture this behavior. Consequently, we do not extend our study of quantum fluctuations for $d = 2$ beyond $2\,d\,J/U \sim 1$.

The pathologies that arise when the dilute limit is approached in our $d = 2$ QGW theory can be seen in Fig. 17(a). Here, it is shown how the control parameter increases monotonically for large $2\,d\,J/U$, which strongly contrasts with the monotonic decay seen in the $d = 3$ QGW study of the one-component BH model [36]. Although the mean-field description provides the usual one-body coherent state in this limit [29], the growth of $F$ indicates that this is not the exact ground state of the dilute gas. Nonetheless, for smaller values of $2\,d\,J/U$, including the critical regions and the deep MI, CFSF and PSF regimes, we observe in Fig. 17(b)-(e) that $F$ remains small, which supports the overall reliability of our approach.

*(Quantum corrections)* – Now that we identified the range of validity of the QGW method, we turn to study the quantum corrections to the local order parameters and densities. Expanding the corresponding Gutzwiller operators to second order in the fluctuations, these quantities are modified as

$$\langle \hat{\mathcal{N}}_i(\mathbf{r}) \rangle = n_{0,i} + \langle \delta_2 \hat{n}_i(\mathbf{r}) \rangle, \tag{A.27}$$

$$\langle \hat{\mathcal{D}}_i(\mathbf{r}) \rangle = d_{0,i} + \langle \delta_2 \hat{D}_i(\mathbf{r}) \rangle, \tag{A.28}$$

$$\langle \hat{\psi}_i(\mathbf{r}) \rangle = \psi_{0,i} + \langle \delta_2 \hat{\psi}_i(\mathbf{r}) \rangle, \tag{A.29}$$

$$\langle \hat{\psi}_P(\mathbf{r}) \rangle = \psi_{0,P} + \langle \delta_2 \hat{\psi}_P(\mathbf{r}) \rangle, \tag{A.30}$$

$$\langle \hat{\psi}_C(\mathbf{r}) \rangle = \psi_{0,C} + \langle \delta_2 \hat{\psi}_C(\mathbf{r}) \rangle, \tag{A.31}$$

where we note that the first-order fluctuations analysed in App. A.2 make a vanishing contribution and hence do not correct the mean-field predictions. The second-order terms appearing on the right-hand side of Eqs. (A.27)-(A.31) are explicitly given by

$$\langle \delta_2 \hat{n}_i(\mathbf{r}) \rangle = -F\, n_{0,i} + \sum_{n_1,n_2} \left( n_1\, \delta_{i,1} + n_2\, \delta_{i,2} \right) \langle \delta \hat{c}^{\dagger}_{n_1,n_2}(\mathbf{r})\, \delta \hat{c}_{n_1,n_2}(\mathbf{r}) \rangle, \tag{A.32}$$

$$\langle \delta_2 \hat{D}_i(\mathbf{r}) \rangle = -F\, d_{0,i} + \sum_{n_1,n_2} \left( n_1^2\, \delta_{i,1} + n_2^2\, \delta_{i,2} \right) \langle \delta \hat{c}^{\dagger}_{n_1,n_2}(\mathbf{r})\, \delta \hat{c}_{n_1,n_2}(\mathbf{r}) \rangle, \tag{A.33}$$

$$\langle \delta_2 \hat{\psi}_1(\mathbf{r}) \rangle = -F\, \psi_{0,1} + \sum_{n_1,n_2} \sqrt{n_1}\, \langle \delta \hat{c}^{\dagger}_{n_1-1,n_2}(\mathbf{r})\, \delta \hat{c}_{n_1,n_2}(\mathbf{r}) \rangle, \tag{A.34}$$

$$\langle \delta_2 \hat{\psi}_2(\mathbf{r}) \rangle = -F\, \psi_{0,2} + \sum_{n_1,n_2} \sqrt{n_2}\, \langle \delta \hat{c}^{\dagger}_{n_1,n_2-1}(\mathbf{r})\, \delta \hat{c}_{n_1,n_2}(\mathbf{r}) \rangle, \tag{A.35}$$

$$\begin{aligned}\langle \delta_2 \hat{\psi}_P(\mathbf{r}) \rangle_c = &-F\, \psi_{0,P} + \sum_{n_1,n_2} \sqrt{n_1\, n_2}\, \langle \delta \hat{c}^{\dagger}_{n_1-1,n_2-1}(\mathbf{r})\, \delta \hat{c}_{n_1,n_2}(\mathbf{r}) \rangle \\ &- \left[ \psi_{0,2}\, \langle \delta_2 \hat{\psi}_1(\mathbf{r}) \rangle + \psi_{0,1}\, \langle \delta_2 \hat{\psi}_2(\mathbf{r}) \rangle \right],\end{aligned} \tag{A.36}$$

$$\begin{aligned}\langle \delta_2 \hat{\psi}_C(\mathbf{r}) \rangle_c = &-F\, \psi_{0,C} + \sum_{n_1,n_2} \sqrt{n_1(n_2+1)}\, \langle \delta \hat{c}^{\dagger}_{n_1-1,n_2+1}(\mathbf{r})\, \delta \hat{c}_{n_1,n_2}(\mathbf{r}) \rangle \\ &- \left[ \psi_{0,2}^*\, \langle \delta_2 \hat{\psi}_1(\mathbf{r}) \rangle + \psi_{0,1}\, \langle \delta_2 \hat{\psi}_2^{\dagger}(\mathbf{r}) \rangle \right],\end{aligned} \tag{A.37}$$

where the symbol $\langle \cdot \rangle_c$ on the left hand-side of Eqs. (A.36)-(A.37) emphasizes that one-body correlations are subtracted on the right hand side so that only genuine pairing/antipairing quantum fluctuations are retained, in accordance with Eqs. (5). At zero temperature, the second-order expectation values appearing in Eqs. (A.32)-(A.37) can be evaluated straightforwardly by a generalization of the following two examples,

$$\langle \delta_2 \hat{n}_i(\mathbf{r}) \rangle = -F\, n_{0,i} + \frac{1}{I} \sum_{\alpha} \sum_{\mathbf{k}} \sum_{n_1,n_2} \left( n_1\, \delta_{i,1} + n_2\, \delta_{i,2} \right) \left| v_{\alpha,\mathbf{k},n_1,n_2} \right|^2, \tag{A.38}$$

$$\langle \delta_2 \hat{\psi}_i(\mathbf{r}) \rangle = -F\,\psi_{0,i} + \frac{1}{I} \sum_\alpha \sum_\mathbf{k} \sum_{n_1,n_2} \left( \delta_{i,1}\,\sqrt{n_1}\,v_{\alpha,\mathbf{k},n_1-1,n_2} + \delta_{i,2}\,\sqrt{n_2}\,v_{\alpha,\mathbf{k},n_1,n_2-1} \right) v_{\alpha,\mathbf{k},n_1,n_2}. \quad \text{(A.39)}$$

from which we obtain self-contained expressions for the one-species filling and the one-body order parameter corrected by quantum fluctuations,

$$\langle \hat{\mathcal{N}}_i(\mathbf{r}) \rangle = (1-F)\,n_{0,i} + \frac{1}{I} \sum_\alpha \sum_\mathbf{k} \sum_{n_1,n_2} \left( n_1\,\delta_{i,1} + n_2\,\delta_{i,2} \right) \left| v_{\alpha,\mathbf{k},n_1,n_2} \right|^2, \quad \text{(A.40)}$$

$$\langle \hat{\psi}_1(\mathbf{r}) \rangle = (1-F)\,\psi_{0,1} + \frac{1}{I} \sum_\alpha \sum_\mathbf{k} \sum_{n_1,n_2} \sqrt{n_1}\,v_{\alpha,\mathbf{k},n_1-1,n_2}\,v_{\alpha,\mathbf{k},n_1,n_2}. \quad \text{(A.41)}$$

Accordingly, the expectation value of the total density is approximated by $n_d \approx n_{0,d} + \sum_i \langle \delta_2 \hat{n}_i(\mathbf{r}) \rangle$, as taken into account in Sec. 3 when discussing the results of the QGW approach.

It is interesting to observe that, within the QGW formalism, quantum-corrected local observables are always given by the sum of two distinct terms, one given by quantum fluctuations only and the other, proportional to the mean-field average, deriving exclusively from the normalization operator via the expectation value $\langle \hat{A}^2(\mathbf{r}) \rangle$ defining the control parameter $F$. This result makes more explicit the physical role of $\hat{A}(\mathbf{r})$, which accounts for the feedback of quantum fluctuations onto the Gutzwiller ground state. Along these lines, we also remark that this contribution is of key importance in giving accurate predictions for the local density and spin correlations presented in Sec. 3.4.

The relative quantum corrections to the total density and the one-body order parameters are shown in Fig. 17(b)-(e) for fixed $n_d$. In the deep CFSF, PSF and MI phases, quantum fluctuations are always small. This remains true also near the transition points, whereas the corrections grow as either the collapse point or the limit $n_d U \to 0$ is approached. These increasingly large corrections mirror the pathologies of this limit in two dimensions. It is worth noticing that, in Fig. 17(b)-(e), the corrections to the one-body order parameters appear to diverge on the brink of the phase transitions. We expect that the self-consistent inclusion of these effects into the phase diagram description would shift the CFSF, PSF and MI boundaries towards larger hopping energies in agreement with the results of quantum Monte Carlo simulations [36,52].

The quantum corrections to the CFSF and PSF order parameters are shown in Fig. 17(f)-(g) for fixed $n_d$ and across the respective phase transitions to the SF regime. In the deep CFSF and PSF phases, these corrections are small, which indicates that the effect of fluctuations in these phases within the QGW picture is minimal. The corrections remain of the order of $10^{-1}$ across the transitions and then monotonically increase as the pathological limit $n_d U \to 0$ is reached. However, we note that the amplitude of relative quantum corrections is amplified by the fact that both $\psi_{0,C}$ and $\psi_{0,P}$ decrease rapidly for weak interactions – see panels (c) of Fig. 1 and Fig. 4.

# B  Linear response formalism within the QGW approach

This appendix is dedicated to a detailed review of the linear response formalism (c.f. Ref. [55]) applied to the QGW quantum theory. For this purpose, we derive the relevant expressions for the density, spin and current response functions; in particular, the latter objects are the basic quantities from which the superfluid matrix is calculated in Sec. 3.2.

## B.1  Linear response formalism

Suppose that a Hermitian, time-dependent perturbation is applied to the Hamiltonian of the system at time $t = 0$,

$$\hat{H}_{\mathrm{p}}(t) = \hat{H} + \theta(t)\,\hat{G}(t)\,. \tag{B.42}$$

Let us define $|\psi_n^N\rangle$ as the $n^{\mathrm{th}}$ eigenstate of $\hat{H}$ with a total number of particles equal to $N$. For $t < 0$, the system is in the true many-body ground state $|\psi_0^N\rangle$ and the time evolution of the system is given by $|\psi(t)\rangle = \hat{U}(t)|\psi_0^N\rangle$ where $\hat{U}(t)$ is the evolution operator with boundary condition $\hat{U}(0) = 1$. Assuming that the perturbation $\hat{G}(t)$ is sufficiently small with respect to all the energy scales of $\hat{H}$, $\hat{U}(t)$ can be approximated by the Born approximation of the solution to the equation $i\hbar\,\partial_t\hat{U}(t) = \hat{H}_{\mathrm{p}}(t)\hat{U}(t)$, namely

$$\hat{U}(t) \approx e^{-i\hat{H}t/\hbar} - \frac{i}{\hbar}\int_0^t d\tau\, e^{-i\hat{H}(t-\tau)/\hbar}\,\hat{G}(\tau)\,e^{-i\hat{H}\tau/\hbar}\,. \tag{B.43}$$

Now, we wish to calculate the time evolution of some observable $\langle\hat{F}(t)\rangle$, which we determine in the Heisenberg picture via the relation $\hat{F}(t) = e^{i\hat{H}t}\,\hat{F}\,e^{-i\hat{H}t}$. Evaluating $\langle\hat{F}(t)\rangle$ using Eq. (B.43), we find

$$\langle\hat{F}(t)\rangle = \langle\psi_0^N\,|\hat{U}^\dagger(t)\,\hat{F}\,\hat{U}(t)|\,\psi_0^N\rangle \approx \langle\hat{F}\rangle_{\mathrm{eq}} - \frac{i}{\hbar}\int_0^t d\tau\,\langle\psi_0^N\,|[\hat{G}(\tau),\hat{F}(t-\tau)]|\,\psi_0^N\rangle\,, \tag{B.44}$$

where the "eq" subscript indicates the equilibrium expectation value. The kernel of the above integral is known as the time-domain response function

$$\chi_{\hat{F},\hat{G}}(t) = -\frac{i}{\hbar}\theta(t)\langle\psi_0^N\,|[\hat{F}(t),\hat{G}(0)]|\,\psi_0^N\rangle\,. \tag{B.45}$$

In frequency space, we have

$$\chi_{\hat{F},\hat{G}}(\omega) = \lim_{\varepsilon\to 0^+}\int_{-\infty}^{\infty} dt\, e^{-\varepsilon t}\,e^{i\omega t}\,\chi_{\hat{F},\hat{G}}(t)\,, \tag{B.46}$$

where the infinitesimal regularization parameter $\varepsilon\to 0^+$ ensures that at $t = -\infty$ the system is governed by the unperturbed Hamiltonian $\hat{H}$. Written in terms of the perturbation operator and the probed observable, we find then the general expression

$$\chi_{\hat{F},\hat{G}}(\omega) = -\frac{i}{\hbar}\lim_{\varepsilon\to 0^+}\int_0^{\infty} dt\, e^{-\varepsilon t}\,e^{i\omega t}\,\langle[\hat{F}(t),\hat{G}(0)]\rangle\,. \tag{B.47}$$

In the following subsections, we proceed to specialize Eq. (B.47) to the case of the density and current response functions of the two-component BH model under the QGW quantum theory. Although the linear fluctuations around the mean-field Gutzwiller state would provide a similar result at lowest order, the QGW formalism makes the calculation straightforward and particularly transparent. Moreover, the higher-order correlations accessed by the QGW quantum theory are shown to introduce the effect of multi-mode fluctuations into the response functions: in particular, this last feature is at the roots of our accurate predictions for the current response functions and, in turn, of the superfluid components of the system.

## B.2  Density response function

As a standard yet relevant case of study, we are interested in the linear response of the **q**-component of the density operator

$$\hat{\rho}_{i,\mathbf{q}} = \sum_{\mathbf{k}} \hat{a}_{i,\mathbf{k}-\mathbf{q}}^\dagger\,\hat{a}_{i,\mathbf{k}}\,, \tag{B.48}$$

for the $i^{\text{th}}$ species under the same type of density perturbation. Therefore, we set $\hat{F} = \hat{G} = \delta\hat{\rho}_{i,\mathbf{q}}$ with $\delta\hat{\rho}_{i,\mathbf{q}} = \hat{\rho}_{i,\mathbf{q}} - \langle\hat{\rho}_{i,\mathbf{q}}\rangle_{\text{eq}}$, where the equilibrium contribution vanishes in a uniform system unless at $\mathbf{q} = \mathbf{0}$. The density-density response function is given by rephrasing Eq. (B.47) into

$$\chi_{\hat{n}_i}(\mathbf{q},\omega) = -\frac{i}{\hbar} \lim_{\varepsilon\to 0^+} \int_0^\infty dt\, e^{-\varepsilon t}\, e^{i\omega t} \left\langle \left[\delta\hat{\rho}_{i,\mathbf{q}}(t), \delta\hat{\rho}_{i,-\mathbf{q}}(0)\right] \right\rangle. \tag{B.49}$$

In order to evaluate this response function within the QGW formalism, we apply the usual quantization procedure to the expectation value in Eq. (B.49), which at the lowest order in the fluctuations is mapped into

$$\left\langle \left[\delta\hat{\rho}_{i,\mathbf{q}}(t), \delta\hat{\rho}_{i,-\mathbf{q}}(0)\right] \right\rangle \longrightarrow \sum_{\mathbf{r}} e^{-i\mathbf{q}\cdot\mathbf{r}} \left\langle \left[\delta_1\hat{n}_i(\mathbf{r},t), \delta_1\hat{n}_i(\mathbf{0},0)\right] \right\rangle, \tag{B.50}$$

where $\delta_1\hat{n}_i(\mathbf{r},t)$ is the first-order expansion of the QGW density operator defined in Eq. (A.3), whose time-dependence is controlled by the interaction picture of the QGW Hamiltonian (42). Therefore, evaluating the commutator in Eq. (B.50), we readily obtain

$$\chi_{\hat{n}_i}(\mathbf{q},\omega) = \frac{2}{\hbar} \sum_\alpha \frac{N_{i,\alpha,\mathbf{q}}^2\, \omega_{\alpha,\mathbf{q}}}{(\omega + i\,0^+)^2 - \omega_{\alpha,\mathbf{q}}^2}. \tag{B.51}$$

As anticipated, our result for $\chi_{\hat{n}_i}(\mathbf{q},\omega)$ agrees exactly with the linear response result in Ref. [32] – see in particular Eq. (29) and App. A therein – based on the time-dependent Gutzwiller approximation, although our quantum formalism offers a more transparent and straightforward tool for the estimation of response functions, as well as the unexplored possibility of including non-Gaussian fluctuations in the density (spin) channel.

## B.3 Current response functions

### B.3.1 Current operator

As a starting ground, we determine the species-resolved local current operators by resorting to the continuity equation, which is given by

$$\nabla_{\mathbf{r}}\hat{j}_{i,\mathbf{r}} = -\frac{\partial\hat{n}_{i,\mathbf{r}}}{\partial t} = \frac{i}{\hbar}\left[\hat{n}_{i,\mathbf{r}}, \hat{H}\right]. \tag{B.52}$$

where $\nabla_{\mathbf{r}}$ has to be intended according to its definition on a lattice. Inserting the BH Hamiltonian (2) on the right-hand side of Eq. (B.52), one finds

$$\hat{j}_{i,\mathbf{r}} = \frac{iJa}{\hbar} \sum_{j=1}^d \left(\hat{a}_{i,\mathbf{r}+\mathbf{e_j}}^\dagger \hat{a}_{i,\mathbf{r}} - \text{h.c.}\right) \mathbf{e_j}. \tag{B.53}$$

For later convenience, we introduce the current operator acting along a specific direction of the square lattice, for instance the $x$-directed links, for which we have

$$\hat{j}_{i,\mathbf{r}}|_x = \frac{iJa}{\hbar}\left(\hat{a}_{i,\mathbf{r}+\mathbf{e_x}}^\dagger \hat{a}_{i,\mathbf{r}} - \text{h.c.}\right). \tag{B.54}$$

In momentum space, the unidirectional current operator Eq. (B.54) transforms into

$$\hat{j}_{i,\mathbf{q}}|_x = \frac{iJa}{\hbar} \sum_{\mathbf{k}} \left[e^{-ik_x a} - e^{i(k_x+q_x)a}\right] \hat{a}_{i,\mathbf{k}}^\dagger \hat{a}_{i,\mathbf{k}+\mathbf{q}}, \tag{B.55}$$

which in the uniform limit $\mathbf{q} \to \mathbf{0}$ becomes

$$\hat{j}_{i,\mathbf{q}\to\mathbf{0}}|_x = \frac{\hbar}{m\,a} \sum_{\mathbf{k}} \sin(k_x\,a)\,\hat{a}^{\dagger}_{i,\mathbf{k}}\,\hat{a}_{i,\mathbf{k}}\,. \tag{B.56}$$

The uniform current operator derived above is the fundamental object by which we estimate the relevant current response functions in the following subsections.

### B.3.2  Intraspecies response

First, we consider the case where both the probe and the response operators correspond to the same current species $\hat{j}_{i,\mathbf{q}}$. Let us take the momentum vector $\mathbf{q}$ to lie along the $z$ axis, then the longitudinal and transverse components of the current operator are given by $\hat{j}_{i,\mathbf{q}}|_z$ and $\hat{j}_{i,\mathbf{q}}|_x$, respectively. In the following, we consider only the static ($\omega = 0$) and uniform ($\mathbf{q} \to \mathbf{0}$) limits of the transverse response function [87,88], which represents the paramagnetic contribution to the superfluid matrix discussed in Sec. 3.2. The relevant response function is

$$\chi^{T}_{\hat{j}_i,\hat{j}_i}(\mathbf{q}\to\mathbf{0},\omega=0) = -\frac{i}{\hbar}\lim_{\varepsilon\to0^+}\int_0^{\infty} dt\, e^{-\varepsilon\,t}\,\left\langle\left[\hat{j}_i|_x(t),\hat{j}_i|_x(0)\right]\right\rangle. \tag{B.57}$$

Starting from Eq. (B.56) and making use of the mapping (A.10), the QGW quantization of the uniform current operator $\hat{j}_{i,\mathbf{q}\to\mathbf{0}}(t)|_x$ is found to be

$$\hat{\mathcal{J}}_i|_x = \frac{\hbar}{m\,a}\sum_{\alpha,\beta}\sum_{\mathbf{k}}\left[U^{*}_{i,\alpha,\mathbf{k}}\,\hat{b}^{\dagger}_{\alpha,\mathbf{k}} + V_{i,\alpha,\mathbf{k}}\,\hat{b}_{\alpha,-\mathbf{k}}\right]\left[U_{i,\beta,\mathbf{k}}\,\hat{b}_{\beta,\mathbf{k}} + V^{*}_{i,\beta,\mathbf{k}}\,\hat{b}^{\dagger}_{\beta,-\mathbf{k}}\right]\sin(k_x\,a), \tag{B.58}$$

in which the presence of the condensate plays no role as $\sin(k_x\,a)$ vanishes at $\mathbf{k} = \mathbf{0}$. Next, on the right-hand side of Eq. (B.57) we insert a complete basis of excited states as a resolution of the identity operator to obtain

$$\chi^{T}_{\hat{j}_i,\hat{j}_i}(\mathbf{0},0) = -\frac{i}{\hbar}\lim_{\varepsilon\to0^+}\int_0^{\infty} dt\, e^{-\varepsilon\,t}\sum_{N,n>0}\left[\left\langle\psi_0^N\left|\hat{\mathcal{J}}_i|_x(t)\right|\psi_n^N\right\rangle\left\langle\psi_n^N\left|\hat{\mathcal{J}}_i|_x(0)\right|\psi_0^N\right\rangle - \text{c.c.}\right]. \tag{B.59}$$

At zero temperature, we obtain that the first-order term of the QGW current operator (B.58) produces a vanishing contribution to the transverse current response. Therefore, we immediately recognize that only expectation values of the kind $\langle\psi_0^N|\,\hat{b}_{\alpha,\mathbf{k}}\,\hat{b}_{\beta,\mathbf{p}}\,|\psi_n^{N+2}\rangle$ and $\langle\psi_0^N|\,\hat{b}_{\alpha,\mathbf{k}}\,\hat{b}_{\beta,\mathbf{p}}\,|\psi_n^{N+2}\rangle$ provide a finite result. More precisely, the second-order of $\hat{\mathcal{J}}_i(t)|_x$ generating such contributions can be written in a symmetric form as

$$\sum_{\alpha,\beta}\sum_{\mathbf{k}}U_{i,\alpha,\mathbf{k}}V_{i,\beta,\mathbf{k}}\,\sin(k_x\,a)\,\hat{b}_{\alpha,\mathbf{k}}\,\hat{b}_{\beta,\mathbf{k}} = \frac{1}{2}\sum_{\alpha,\beta}\sum_{\mathbf{k}}\left[U_{i,\alpha,\mathbf{k}}V_{i,\alpha,\mathbf{k}} - U_{i,\beta,\mathbf{k}}V_{i,\alpha,\mathbf{k}}\right]\sin(k_x\,a)\,\hat{b}_{\alpha,\mathbf{k}}\,\hat{b}_{\beta,\mathbf{k}}, \tag{B.60}$$

where we have used the fact that the particle (hole) amplitudes $U_{i,\alpha,\mathbf{k}}\,(V_{i,\alpha,\mathbf{k}})$ are even functions of momentum in our system. Plugging the right-hand side of Eq. (B.60) back into the linear response function (B.59) and computing the time integral, we find

$$\chi^{T}_{\hat{j}_i,\hat{j}_i}(\mathbf{0},0) = -\frac{\hbar}{(m\,a)^2}\frac{2^2}{4}\sum_{\alpha,\beta}\sum_{\mathbf{k}}\frac{\left(U_{i,\alpha,\mathbf{k}}V_{i,\beta,\mathbf{k}} - U_{i,\beta,\mathbf{k}}V_{i,\alpha,\mathbf{k}}\right)^2}{\omega_{\alpha,\mathbf{k}} + \omega_{\beta,\mathbf{k}}}\sin^2(k_x\,a), \tag{B.61}$$

where we have neglected the conjugation of the particle (hole) amplitudes $U_{i,\alpha,\mathbf{k}}\,(V_{i,\alpha,\mathbf{k}})$ for simplicity, as they turn out to be always real in the present case. In the above equation, although the numerical factors cancel each other, we have chosen to write them explicitly in order to emphasize their origins. The factor of $1/4$ arises from the symmetrization of the current

operator in Eq. (B.60); one factor of 2 descends from the inner product $\langle \psi_0^N | \hat{b}_{\alpha,\mathbf{k}}\, \hat{b}_{\beta,\mathbf{p}} | \psi_n^{N+2} \rangle$ (appearing twice), while the other factor of 2 comes from the complex conjugate term in Eq. (B.59).

### B.3.3 Interspecies response

Now, we consider the case in which the probe is proportional to the current of one species while we measure the linear response of the current of the other species, in the transverse direction and in the static-uniform limit as before. This corresponds to the off-diagonal response function

$$\chi^T_{\hat{j}_1,\hat{j}_2}(\mathbf{q} \to \mathbf{0}, \omega = 0) = -\frac{i}{\hbar} \lim_{\varepsilon \to 0^+} \int_0^\infty dt\, e^{-\varepsilon t} \left\langle \left[ \hat{j}_1|_x(t), \hat{j}_2|_x(0) \right] \right\rangle. \tag{B.62}$$

The calculation of $\chi^T_{\hat{j}_1,\hat{j}_2}(\mathbf{q} \to \mathbf{0}, \omega = 0)$ within the QGW framework can be performed analogously to the intraspecies case by considering the current operator (B.58) for both the bosonic flavours. The final result is

$$\chi^T_{\hat{j}_1,\hat{j}_2}(\mathbf{0},0) = -\frac{\hbar}{(m a)^2} \frac{2^2}{4} \sum_{\alpha,\beta} \sum_{\mathbf{k}} \frac{\prod_{i=1}^2 \left( U_{i,\alpha,\mathbf{k}} V_{i,\beta,\mathbf{k}} - U_{i,\beta,\mathbf{k}} V_{i,\alpha,\mathbf{k}} \right)}{\omega_{\alpha,\mathbf{k}} + \omega_{\beta,\mathbf{k}}} \sin^2(k_x a), \tag{B.63}$$

where the numerical factors have the same interpretations as for Eq. (B.61).

It is instructive to briefly comment on how Eq. (B.63) fundamentally differs from the equivalent result of Ref. [27] within the Bogoliubov approximation. In the first place, the expression of $\chi^T_{\hat{j}_1,\hat{j}_2}(\mathbf{q} \to \mathbf{0}, \omega = 0)$ in that work contains an additional factor of $1/2$. This is due to the fact that the $U$'s and $V$'s for the spin and density Goldstone modes are each normalized to one in Ref. [27], whereas in the present case the normalization condition for the $U$'s and $V$'s is given by the sum over all the excitation branches (see the relative discussion in App. A.2), which typically takes more than two branches to saturate numerically. Furthermore, the two results differ by a minus sign. When only the first two excitation branches are considered in the QGW method, one finds that the $U$'s and $V$'s for the density Goldstone mode differ by a minus sign between the two species, whereas they are identical for the spin mode. This results in an overall minus sign when only these two branches are considered in Eq. (B.63). However, we notice that the normalization condition is far from being saturated if limited to the Goldstone modes, so a good agreement with the Bogoliubov result is expected only in the weakly-interacting regime. These remarks resolve the apparent discrepancies between the Bogoliubov and QGW predictions.

## B.4 Average kinetic energy

Albeit not being a proper response function, we conclude by calculating the mean value of the average kinetic energy along $x$-directed links of the lattice, $K_i|_x$, which is exactly proportional to the density $n_i$ in the well-known expression for the superfluid matrix on the continuum [27]. The local kinetic energy operator for the $i^{\text{th}}$ species along the $x$ direction is defined as

$$\hat{K}_i|_x(\mathbf{r}) = -J \left( \hat{a}^\dagger_{\mathbf{r}+\mathbf{e}_x,\mathbf{i}} \hat{a}_{\mathbf{r},i} + \text{h.c.} \right). \tag{B.64}$$

Within the mean-field Gutzwiller approximation, one obtains $K_{0,i}|_x(\mathbf{r}) = -2J \left| \psi_{0,i} \right|^2$, which is trivially proportional to the condensate fraction. To improve this physical description of the kinetic energy, we evaluate the average kinetic energy by rewriting the operator (B.64) in terms of the QGW Bose fields (A.7)-(A.8). It follows that, up to the lowest-order in

the quantum fluctuations, the QGW local kinetic energy operator reads

$$
\begin{aligned}
\hat{\mathcal{K}}_i|_x(\mathbf{r}) \approx &-2J\left|\psi_{0,i}(\mathbf{r})\right|^2 - J\left[\psi_{0,i}(\mathbf{r})\,\delta_1\hat{\psi}_i^\dagger(\mathbf{r}+\mathbf{e}_x) + \psi_{0,i}^*(\mathbf{r}+\mathbf{e}_x)\,\delta_1\hat{\psi}_i(\mathbf{r}) + \text{h.c.}\right] \\
&- J\left[\delta_1\hat{\psi}_i^\dagger(\mathbf{r}+\mathbf{e}_x)\,\delta_1\hat{\psi}_i(\mathbf{r}) + \text{h.c.}\right].
\end{aligned}
\tag{B.65}
$$

For a uniform lattice, the average value of $\hat{\mathcal{K}}_i|_x(\mathbf{r})$ is given by

$$
K_i|_x \approx -2J\left|\psi_{0,i}\right|^2 - \frac{2J}{I}\sum_\alpha\sum_{\mathbf{k}}\left|V_{i,\alpha,\mathbf{k}}\right|^2\cos(k_x a),
\tag{B.66}
$$

where we have made use of Eq. (A.9) and neglected finite-temperature contributions. We notice that only the first term on the right-hand side of Eq. (B.66) agrees with the prediction of mean-field theory, while the second term is due to zero-point fluctuations properly accounted by our quantum theory. In this regard, the expression of $K_i|_x$ recalls the average kinetic energy of the one-component system derived in Ref. [36].

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
