# Peer review of "Quantum Gutzwiller approach for the two-component Bose-Hubbard model"

_SciPost Physics, doi:SciPost Phys. 12, 111 (2022)_

## Round 1 · Referee Report · Anonymous (Referee 1) · 2022-1-29

Report

The issues that raised during the first round of refereeing have been satisfactorily addressed. I thus recommend this paper to be accepted for publication.

---

## Round 1 · Referee Report · Anonymous (Referee 2) · 2022-1-29

Report

The authors have replied to my criticism and I think that the paper can be published.

---

## Round 1 · Author Response

Dear Editor,

We thank the referees for their detailed comments and suggestions, which were very helpful in improving our manuscript.

We have revised our manuscript in order to address the comments of the referees. We hope that with these minor revisions our manuscript is ready for publication in SciPost Physics.

Below, we respond to the referee comments, inserted for convenience, and we give more details on the changes in the manuscript resulting from these comments. At the end of this letter, we provide a summary of all the changes. All changes to the manuscript are indicated in blue for convenience.

Sincerely yours, Victor Colussi, Fabio Caleffi, Chiara Menotti, Alessio Recati

Reply to Referee 1

We thank Referee 1 for their comments that have helped us improve the manuscript. We respond to the referee’s critique in detail below.

Sincerely yours, The Authors

Critique #1: However in Figure 9 error bars on the QMC data are neither shown nor mentioned. Can this point be further elaborated?

Response #1: We thank the referee for pointing this discrepancy out. The QMC data of Ref. [25] referenced in Fig. 9 has no error bars, however the arXiv version of this paper (arXiv:1801.03052) does. In writing our manuscript, we have contacted the authors of this work and agreed to compare against their published data, which they subsequently shared with us. We have removed any mention of error bars for the QMC data in Fig. 9 from our manuscript to avoid any confusion. Additionally, we have made explicit in the acknowledgments section that the QMC data in Fig. 9 was provided directly by the authors of Ref. [25] and compared against in our manuscript with their permission.

Critique #2: In section 3.2.2 in the discussion of the Mott Insulator to Superfluid it is stated that the superfluid density is always larger than the condensate fraction |ψ0,i|2|ψ0,i|2. Can the comparison between superfluid density and condensate fraction be more explicitly shown, either in the plot in Figure 10, or by providing some numerical values?

Response #2: We thank the referee for this suggestion and have added this comparison to Figs. 10 - 12 in the updated manuscript.

Minor Critiques

1. In the caption of Fig.1 there is some confusion with the labels (panel d actually refers fo MI/SF transition and panel e to the CFSF/SF one).

  1. Response: We have fixed the labeling of this figure.

2. In Fig.2 I think it should be more explicit that the blue dotted line refers to the non dispersive modes (it is stated in the body of the text, but I think that it would be clearer if it was mentioned in the caption too).

  1. Response: We have added this to the caption per the referee’s suggestion.

  1. Looking at the caption and the labels referring to sound velocities/response functions it looks to me that the upper/lower panels in Figure 8 are swapped.
    1. Response: We thank the referee for catching this and have corrected the figure.

Reply to Referee 2

We thank Referee 2 for their comments and critiques on our manuscript. To the referee’s stated weakness that “the paper is extremely long and detailed and is not easy to read”, we comment that we have chosen to write the manuscript in a longer, more detailed style motivated by the SciPost Physics’ flexible format and content guideline that “the article should however constitute a complete, self-contained unit”. Our motivation for the chosen structure is given already in the introduction where we state that the self-contained manner of the work was chosen to provide a reference and starting point for future studies in the field. We feel that this is appropriate given the relative newness of the quantum Gutzwiller method and the potential for its increased usage in the field in the future. Our efforts to improve the readability of the manuscript are detailed in the responses below, and we hope that the referee finds the new manuscript now suitable to recommend for publication.

Sincerely yours, The Authors

Critique #1: The paper is rather detailed and long and, therefore, not easy to follow. In particular, it looks that the results are listed one after the other, without an explanation on their actual relevance and connection to what has been already seen. For example, I do not understand the actual motivation of the section on the correlation functions, since it does not add much on what we learned from section 3. In this sense, it would be useful to add a motivation to study correlation functions, beyond what has been already done.

Response #1: We thank the referee for their critique, and agree that with such a long paper it is easy for the lines of motivation to be lost amidst lists of formulae and results. This is something that is not always easy to gauge when writing such a long manuscript, and therefore we find the opinion of the referee to be very valuable. To improve the readability, we have added various sentences throughout the work to help boost and maintain the narrative line - that we are ultimately focused on studying the role of quantum fluctuations using a novel method.

We agree with the referee’s point that both sections 3 and 4 deal essentially with correlation functions. Therefore, these sections are now combined into one section on “Correlation Functions”. We clarify that the coherence, density fluctuations, and spin fluctuations correlation functions of the (previous) section 4 were studied simply to provide a better and more complete understanding of the underlying physics. Importantly, the calculation of the local spin/density fluctuations provides an important example of how a squared operator (e.g. squared density) and its expectation value squared (e.g. density squared) are formally distinct after quantization. This highlights a subtle but important point in the usage of the QGW method as discussed in the first paragraph in Sec. 3.4 of the updated manuscript.

Minor Critiques:

  1. In Fig.1, it would be useful to see in panels a) and b) what is the curve mu=mu(U) corresponding to the curves reported in panels c), d) and e).

    a. Response: In fact that the mu=mu(U) curves were already shown in a) and b) in the previous version. We have made these curves now more visible and indicated them explicitly in the caption so their meaning is unambiguous. We have chosen to draw these curves up to the superfluid transition following the convention of a previous work [36].

  2. In Fig.7, the panels could be taken a bit wider than now.
    1. Response: We have made this figure now wider.

  3. In Fig.8, the caption is not compatible with the positions of the panels.
    1. Response: The caption has been fixed per the referee’s suggestion and that of referee 1.

---

## Round 1 · List of Changes

Summary of Additional Changes

  1. The units have been corrected in the captions of Figs. 10 – 12.

---

## Editorial Decision

published